# Transcription–replication conflicts underlie sensitivity to PARP inhibitors

Michalis Petropoulos[1], Angeliki Karamichali[1,4], Giacomo G. Rossetti[2,4], Alena Freudenmann[2,3], Luca G. Iacovino[2], Vasilis S. Dionellis[1], Sotirios K. Sotiriou[2,3] & Thanos D. Halazonetis[1✉]

An important advance in cancer therapy has been the development of poly(ADP-ribose) polymerase (PARP) inhibitors for the treatment of homologous recombination (HR)-deficient cancers[1–6]. PARP inhibitors trap PARPs on DNA. The trapped PARPs are thought to block replisome progression, leading to formation of DNA double-strand breaks that require HR for repair[7]. Here we show that PARP1 functions together with TIMELESS and TIPIN to protect the replisome in early S phase from transcription–replication conflicts. Furthermore, the synthetic lethality of PARP inhibitors with HR deficiency is due to an inability to repair DNA damage caused by transcription–replication conflicts, rather than by trapped PARPs. Along these lines, inhibiting transcription elongation in early S phase rendered HR-deficient cells resistant to PARP inhibitors and depleting PARP1 by small-interfering RNA was synthetic lethal with HR deficiency. Thus, inhibiting PARP1 enzymatic activity may suffice for treatment efficacy in HR-deficient settings.

Homologous recombination (HR) is a main pathway for repair of DNA double-strand breaks (DSBs); yet, a fraction of human cancers, often of ovarian, breast, prostate and pancreas origin, are HR-deficient due to biallelic mutations of genes involved in HR repair[8,9]. Cells defective in HR are very sensitive to PARP inhibitors[5,6] and, as a result, PARP inhibitors have been developed as therapeutic agents[1–4]. The normal cells of these patients remain HR-proficient and, hence, are resistant to PARP inhibitors.

The human PARP family comprises 17 members, of which PARP1 and PARP2 are the only known members that function in DNA repair and are capable of poly(ADP-ribosyl)ation[10]. PARP1 and PARP2 bind to various types of DNA damage lesions, including single-strand nicks, single-strand gaps and DSBs[11,12]. Binding to these lesions involves a conformational switch that traps the PARPs on DNA and enhances their catalytic activity[13–16]. Once activated, PARP1 and PARP2 PARylate various substrates, including histones and themselves[11,12]. AutoPARylation facilitates their release from DNA and switches their conformation back to the catalytically inactive state[13,15,16].

According to the enzymatic cycle described above, inhibiting the catalytic activity of PARPs will prevent autoPARylation and keep these enzymes trapped on DNA; thus, the trapping potential of PARP inhibitors should be proportional to their ability to inhibit PARP catalytic activity[17–20]. However, several studies have reported a poor correlation between the inhibitory and trapping potentials of PARP inhibitors[21–23]; this poor correlation has been attributed to differences in reverse allostery[24]. In the normal catalytic cycle, binding of PARPs to DNA damage sites transmits a conformational switch from the DNA binding domain to the catalytic domain to regulate catalytic activity; in reverse allostery, binding of an inhibitor to the catalytic domain transmits a conformational change from the catalytic domain to the DNA binding domain, thereby affecting trapping. PARP inhibitors may enhance retention of PARPs on DNA, or be neutral, or favour release of PARPs from DNA, depending on how they affect, by means of reverse allostery, the conformation of the DNA binding domain[24].

The mechanism by which PARP inhibitors induce lethality of HR-deficient cancer cells is at present attributed to trapping of PARPs on DNA[7,21–23]. Specifically, it has been proposed that trapped PARPs block progression of the replisome, leading to formation of DNA DSBs that require HR for repair. Here, we describe a new role of PARP1, together with TIMELESS and TIPIN, to prevent transcription–replication conflicts (TRCs) and propose that this function is more relevant for the synthetic lethality of PARP inhibitors with HR deficiency.

## Role of TIMELESS and TIPIN in TRCs

In budding yeast, the proteins Tof1 and Csm3 protect the replisome from conflicts with transcription[25–27]. We examined whether TIMELESS and TIPIN, the mammalian orthologues of Tof1 and Csm3, respectively[28], have a similar function. HeLa human cervical carcinoma cells, transfected with small-interfering RNAs (siRNAs), were synchronized with thymidine at the G1/S boundary and then released into S phase, either in the presence or absence of 5,6-dichloro-1-beta-D-ribofuranosylbenzimidazole (DRB), an inhibitor of transcription elongation[29]; DNA damage was assessed 100 or 200 min later (Fig. 1a and Extended Data Fig. 1a–c). Depletion of TIMELESS or TIPIN led to formation of γH2AX, 53BP1 and RAD51 foci, in a manner dependent on transcription elongation (Fig. 1b,c). Similar results were obtained with synchronized U2OS osteosarcoma and hTERT-RPE1 immortalized retinal pigment epithelial cells (Extended Data Fig. 1d) and with other inhibitors of transcription (Extended Data Fig. 1e–h).

In the above experiments, most cells were in S phase, as they were released from a thymidine-induced G1/S arrest. To determine whether

[1]Department of Molecular and Cellular Biology, University of Geneva, Geneva, Switzerland. [2]FoRx Therapeutics AG, Basel, Switzerland. [3]Present address: Roche Pharma Research and Early Development, Roche Innovation Center Basel, Basel, Switzerland. [4]These authors contributed equally: Angeliki Karamichali, Giacomo G. Rossetti. ✉e-mail: thanos.halazonetis@unige.ch

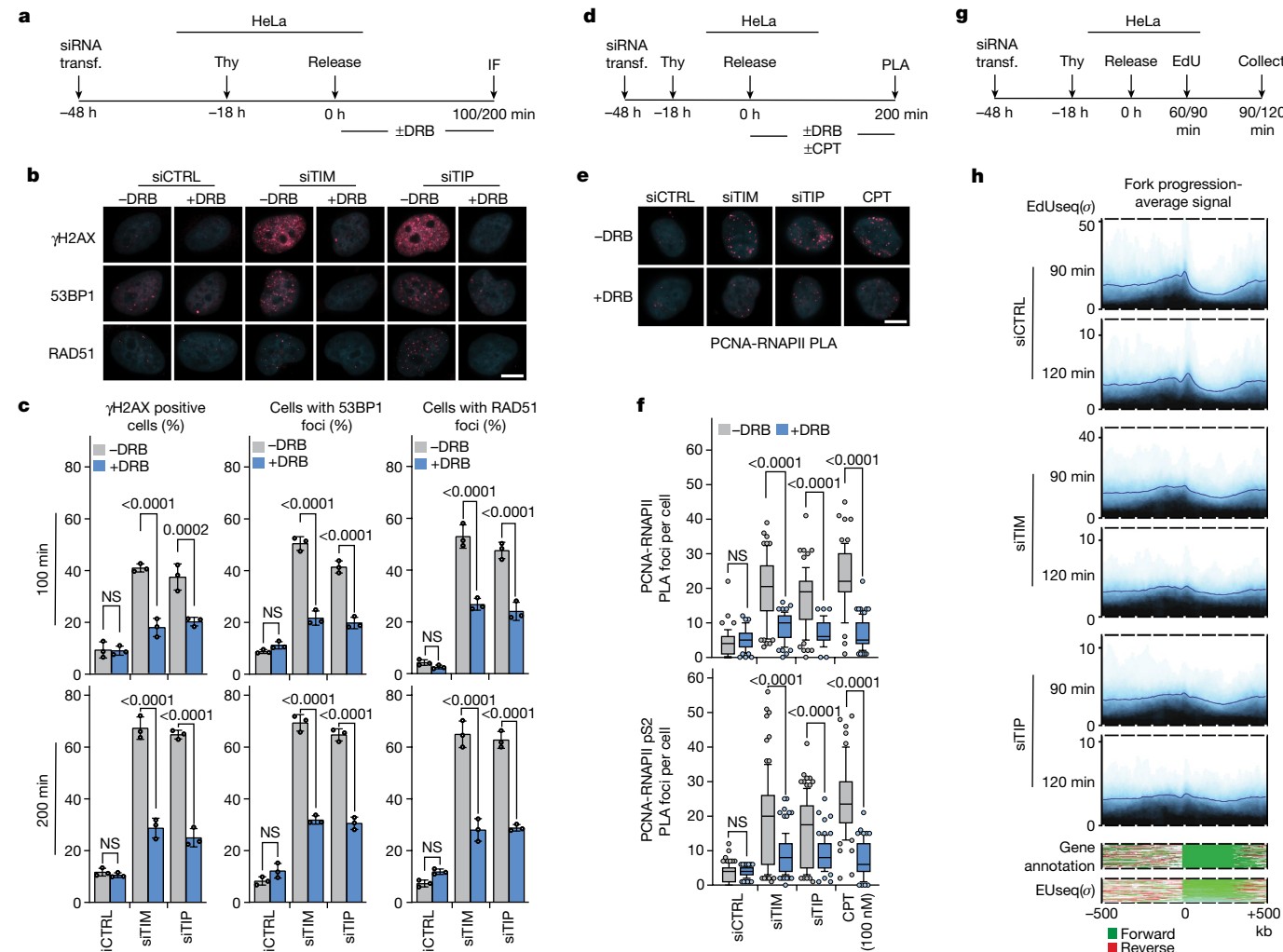

**Fig. 1 | TIMELESS and TIPIN suppress the occurrence of TRCs. a–c**, DNA damage response in HeLa cells depleted for TIMELESS (TIM) or TIPIN (TIP). **a**, Outline of the experiment. **b**, Representative immunofluorescence (IF) images; the nuclei were counterstained with DAPI. **c**, Means ± 1 s.d. of percentage of cells with more than 20 γH2AX, more than 20 53BP1 or more than 10 RAD51 foci; *n* = 3 replicates; more than 259 cells per group (range 259–414); analysis of variance (ANOVA). A residual DNA damage response in cells treated with DRB could reflect TRC-independent mechanisms or transcription elongation complexes that had already escaped the promoter when DRB was added. **d–f**, Detection of TRCs by PLA. **d**, Outline of the experiment with camptothecin (CPT) as positive control. **e**, Immunofluorescence images of PCNA-RNAPII PLA foci; the nuclei were counterstained with DAPI. **f**, Number of PLA foci per cell;

plots show medians and value ranges of 25–75 and 10–90%; filled circles indicate the cells in the top and bottom deciles; *n* = 2 replicates; more than 51 cells per group (range 51–106); ANOVA. **g,h**, Increased fork progression following depletion of TIMELESS or TIPIN. **g**, Outline of the experiment. **h**, Average EdUseq signal over large (more than 300 kb) transcribed genes 90 and 120 min after release into S phase. The genes were aligned by their transcription start site with their 5′–3′ orientation from left to right. Lower panels show the gene annotation and EUseq signal for each genomic locus used to generate the EdUseq plot. Scale bars in microscopy images, 5 μm. Span of genomic regions, 1 Mb. CTRL, control; EU, 5-ethynyl uridine; NS, not significant; σ, sigma value; Thy, thymidine; transf., transfection.

DNA replication was required for induction of the DNA damage response, we repeated the experiment with unsynchronized cells that were treated with 5-ethynyl-2′-deoxyuridine (EdU) to distinguish replicating from non-replicating cells. Depletion of TIMELESS or TIPIN induced a DNA damage response only in the cells that were in S phase and that were not treated with DRB, suggesting the involvement of TRCs (Extended Data Fig. 1i–k).

Several more experiments linked TRCs to the observed DNA damage response. First, a proximity ligation assay (PLA), which monitored physical proximity of Proliferating Cell Nuclear Antigen (PCNA) to RNA-polymerase II (RNAPII), showed an enhanced signal after TIMELESS or TIPIN depletion; moreover, this enhanced signal was strongly attenuated by DRB (Fig. 1d–f). Second, depletion of TIMELESS or TIPIN in cells expressing catalytically inactive RNase H1 led to the emergence of discreet RNase H1 foci indicating the presence of R-loops (Extended

Data Fig. 2a–c). Third, overexpression of wild-type RNase H1, which helps resolve R-loops, attenuated the DNA damage response induced by TIMELESS or TIPIN depletion (Extended Data Fig. 2d–f). We note that R-loops accumulate in response to TRCs and are associated with formation of DNA breaks[30–32].

Previous studies have shown that DNA damage induced by TRCs in early S phase can persist until mitosis, where it is repaired by mitotic DNA synthesis (MiDAS)[33,34]. We wondered whether some of the DNA damage induced by depletion of TIMELESS or TIPIN also remained unrepaired until mitosis. HeLa cells transfected with the appropriate siRNAs were released into S phase in the presence of RO-3306, a Cdk1 inhibitor that prevents mitotic entry; DRB was also optionally administered, but only during the first 200 min of S phase; 11 h after release into S phase, RO-3306 was withdrawn and EdU was added to monitor MiDAS (Extended Data Fig. 2g). Depletion of either TIMELESS or TIPIN

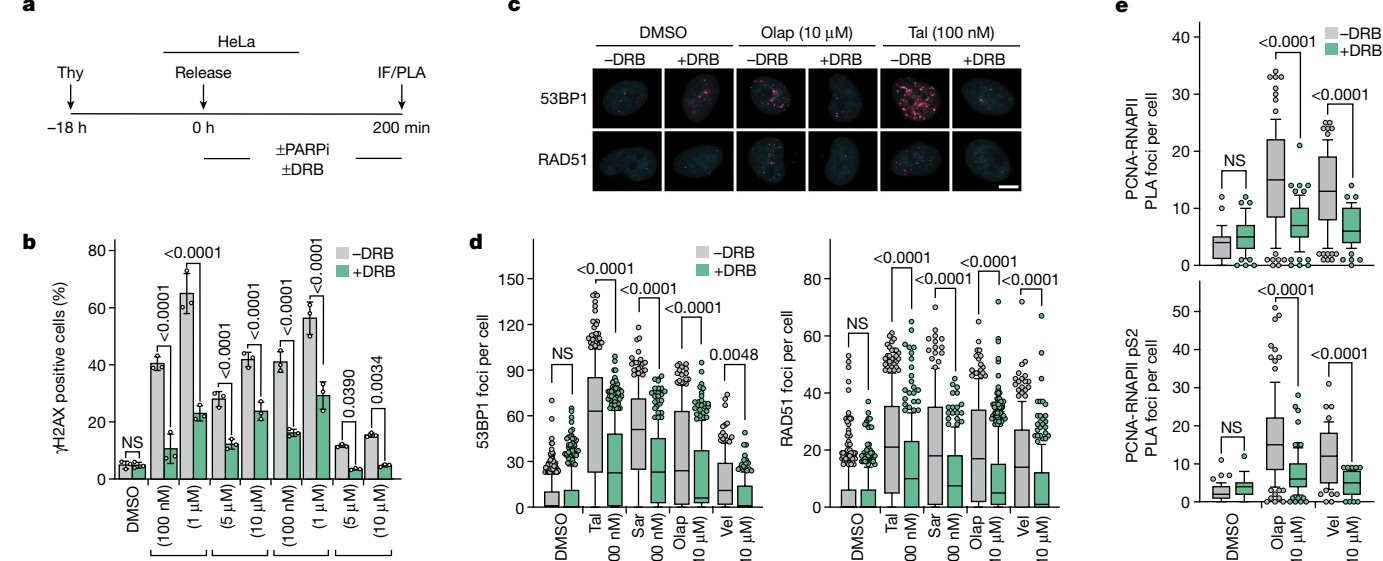

**Fig. 2 | PARP inhibitors induce TRCs. a–e**, DNA damage response and TRCs in HeLa cells treated with PARP inhibitors (PARPi). **a**, Outline of the experiment. **b**, Means ± 1 s.d. of percentage of cells with more than 20 γH2AX foci; n = 3 replicates; more than 232 cells per group (range 232–461); analysis of variance (ANOVA). **c**, Representative immunofluorescence (IF) images; the nuclei were counterstained with DAPI. **d**, Number of RAD51 and 53BP1 foci per cell; plots show medians and value ranges of 25–75 and 10–90%; filled circles indicate the cells in the top decile; n = 3 replicates; more than 142 cells per group (range 142–542); ANOVA. **e**, Number of PLA foci per cell plotted as in **d**; n = 2 replicates; more than 51 cells per group (range 51–99); ANOVA. Scale bar in microscopy images, 5 μm. NS, not significant; Olap, olaparib; Sar, saruparib; Tal, talazoparib; Thy, thymidine; Vel, veliparib.

led to MiDAS, except for the cells that were treated with DRB during the first 200 min of S phase (Extended Data Fig. 2h,i), consistent with DNA damage being induced in early S phase and persisting until mitosis.

In budding yeast, DNA replication forks pause at centromeres and at highly transcribed transfer RNA loci in a Tof1-dependent manner[27]. To determine whether TIMELESS and TIPIN affect fork progression in human cells, we depleted TIMELESS or TIPIN by siRNA and examined the kinetics of DNA replication over large expressed genes. We note that expressed genes lack intragenic origins and are, therefore, replicated by forks originating from upstream and/or downstream intergenic regions[35]. The cells were released into S phase for 90 or 120 min, EdU was added during the last 30 min and DNA synthesis was monitored by the EdUseq method[35] (Fig. 1g). In the control cells, 90 min after release from the thymidine block, the large, transcribed genes had not yet been replicated, whereas at 120 min replication had advanced but was still incomplete. In the cells depleted of TIMELESS or TIPIN, both at 90 and 120 min, replication had advanced further into the gene bodies than in the control cells (Fig. 1h and Extended Data Fig. 2j,k).

## PARP inhibitors induce TRCs in early S phase

PARP1 and TIMELESS physically interact[36,37]. We, therefore, examined whether PARP inhibitors phenocopy depletion of TIMELESS or TIPIN. HeLa cells were arrested at the G1/S boundary and released into S phase with or without PARP inhibitors and with or without DRB; 200 min later, γH2AX levels were monitored (Fig. 2a). Four PARP inhibitors were examined: olaparib, talazoparib, veliparib and saruparib (also known as AZD5305). All induced a DNA damage response, which was suppressed by DRB and other inhibitors of transcription elongation (Fig. 2b–d and Extended Data Fig. 3a,b). Olaparib and veliparib were also examined for their ability to induce physical proximity of PCNA with RNAPII; both PARP inhibitors led to a positive PLA signal, which was dependent on transcription elongation (Fig. 2e).

Next, we monitored whether the timing of exposure of the cells to PARP inhibitors during S phase was important for inducing a DNA damage response. Synchronized cells were treated with PARP inhibitors and,

optionally, with DRB during early S phase (0–3.5 h after release from a thymidine block), mid-S phase (3.5–7 h) or late S phase (7–10.5 h); γH2AX levels were monitored at the end of the PARP inhibitor treatment (Extended Data Fig. 3c). A DNA damage response, suppressible by DRB, was observed in the cells treated with PARP inhibitors in early S phase; the DNA damage response was weaker in the cells treated with PARP inhibitors in mid-S phase and practically absent in the late S phase-treated cells (Extended Data Fig. 3d). The magnitude of the DNA damage response correlated with the number of expressed genes that map to the early, mid- and late S replicating genomic domains, respectively (Extended Data Fig. 3e). Similar results were obtained when we monitored MiDAS; HeLa cells exposed to olaparib or talazoparib during early, but not late, S phase showed MiDAS, which was suppressed by concurrent administration of DRB (Extended Data Fig. 3f–h).

The above experiments indicate that PARP inhibitor treatment, particularly in early S phase, leads to TRCs. In further support of this conclusion, treatment of cells expressing catalytically inactive RNase H1 with PARP inhibitors led to the emergence of discreet RNase H1 foci indicating formation of R-loops (Extended Data Fig. 3i,j) and overexpression of wild-type RNase H1, which disrupts R-loops, attenuated the DNA damage response induced by PARP inhibitors (Extended Data Fig. 3k,l).

## PARP inhibition, not trapping, linked to TRCs

The above experiments do not address whether the induction of TRCs by PARP inhibitors requires trapping of PARPs on DNA or whether it is sufficient to inhibit the enzymatic activity of PARPs. As a first step to address this question, we characterized the PARP inhibitors for inhibition of PARP enzymatic activity, PARP trapping and induction of a TRC-dependent DNA damage response.

In vitro, the enzymatic activity of PARP1 was inhibited almost equipotently by all four PARP inhibitors, whereas PARP2 was inhibited potently by talazoparib, olaparib and veliparib, but less so by saruparib, which is a PARP1-selective inhibitor[38] (Extended Data Fig. 4a). By contrast, in cells, the $IC_{50}$ (half-maximum inhibitory concentration) values for inhibition of PARP enzymatic activity varied by more than 1,000-fold, from

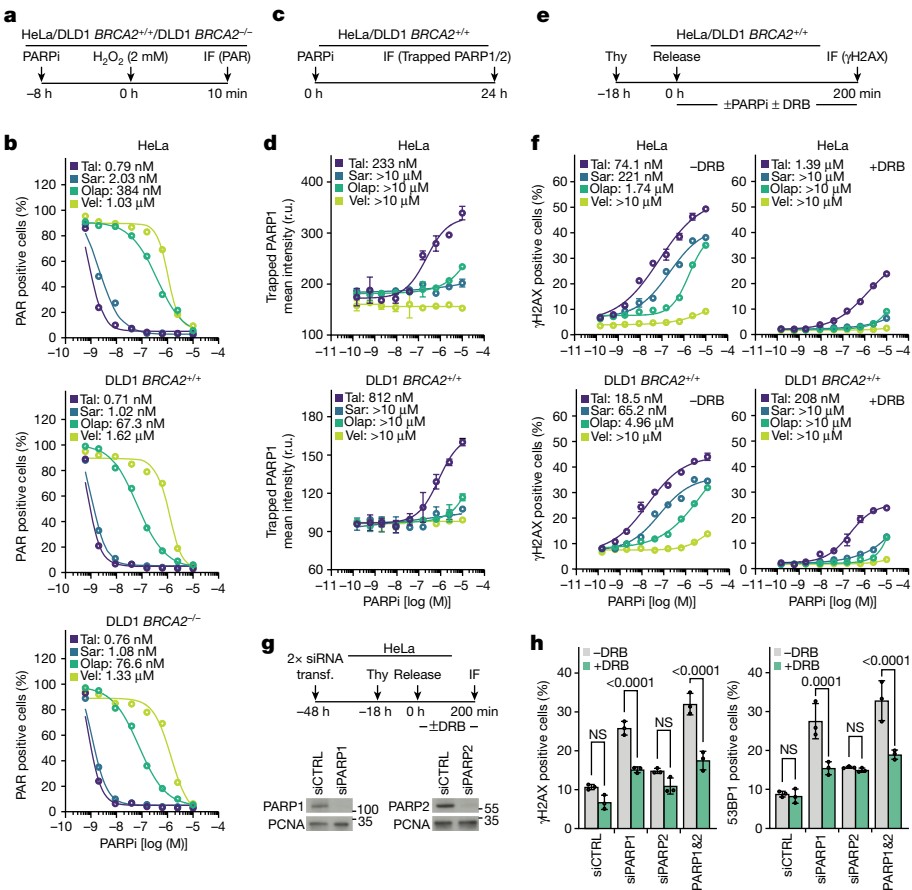

**Fig. 3 | Inhibiting PARP enzymatic activity induces TRCs. a,b,** PARP enzymatic activity in cells treated with PARP inhibitors (PARPi). **a,** Outline of the experiment. **b,** Dose–response curves and calculated IC$_{50}$ values for inhibition of PARP enzymatic activity. One of $n$ = 2 replicates is shown; 100 cells per data point. **c,d,** PARP1 trapping in cells treated with PARP inhibitors. **c,** Outline of the experiment. **d,** Dose–response curves and calculated EC$_{50}$ values for PARP1 trapping; means ± 1 s.d. shown; $n$ = 2 replicates; for HeLa, more than 1,905 (range 1,905–9,368) cells per data point; for DLD1, more than 1,789 (range 1,789–10,815) cells per data point. **e,f,** DNA damage response in cells treated with PARP inhibitors. **e,** Outline of the experiment. **f,** Dose–response curves and calculated EC$_{50}$ values for induction of γH2AX; means ± 1 s.d. shown;

$n$ = 2 replicates; for HeLa, more than 7,597 (range 7,597–10,114) cells per data point; for DLD1, more than 5,512 (range 5,512–9,353) cells per data point. **g,h,** DNA damage response in cells transfected with siRNA targeting *PARP1* and/or *PARP2*. **g,** Outline of the experiment and immunoblotting to monitor PARP1 and PARP2 depletion; PCNA is loading control. **h,** Means ± 1 s.d. of percentage of cells with more than 20 γH2AX or more than 20 53BP1 foci; $n$ = 3 replicates; more than 148 cells per group (range 148–243); analysis of variance (ANOVA). CTRL, control; IF, immunofluorescence; NS, not significant; Olap, olaparib; r.u., relative units; Sar, saruparib; Tal, talazoparib; Thy, thymidine; transf., transfection; Vel, veliparib.

---

about 1 nM for talazoparib and saruparib to about 1 µM for veliparib (Fig. 3a,b and Extended Data Fig. 4b). Trapping of PARP1 and PARP2 on chromatin (Fig. 3c,d and Extended Data Fig. 4c,d) did not correlate well with inhibition of PARP catalytic activity. For example, talazoparib trapped PARP1 much more potently than saruparib, even though both inhibited equally well the enzymatic activity of PARP1 in cells (Fig. 3b). These results are consistent with previous studies showing that PARP inhibitors have different capacities to trap PARPs on DNA due to differences in reverse allostery[24].

The half-maximum effective concentration (EC$_{50}$) values of the four PARP inhibitors for inducing a TRC-dependent DNA damage response in early S phase (Fig. 3e,f and Extended Data Fig. 4e), correlated best with inhibition of PARP enzymatic activity, rather than with trapping of PARP1 or PARP2 on chromatin (Fig. 3a–d and Extended Data Fig. 4d). Talazoparib and saruparib were equipotent in inducing a TRC-dependent DNA damage response, despite having very different PARP trapping activities, and olaparib and veliparib, which were weak inducers of a DNA damage response, were also the weakest inhibitors of PARP enzymatic activity in cells (Fig. 3b,f).

To further explore whether PARP trapping is required for induction of a TRC-dependent DNA damage response, we depleted PARP1

and PARP2 by siRNA and monitored DNA damage markers in early S phase HeLa cells (Fig. 3g). Depletion of PARP1 induced a DNA damage response, in a manner dependent on transcription elongation, whereas depletion of PARP2 had no effect (Fig. 3h). These results support our conclusion that inhibiting PARP enzymatic activity is sufficient to induce TRCs, because depleted PARPs cannot be trapped. Moreover, it seems that only PARP1 prevents TRC-induced DNA damage, even though during mouse embryonic development there is some partial overlap of the functions of PARP1 and PARP2 (ref. 39).

## TIMELESS and PARP1 act through the same pathway

PARP1 and TIMELESS interact with each other[36,37], suggesting that they may act through the same molecular pathway to prevent TRCs. As a first step to explore this hypothesis, we depleted TIMELESS or TIPIN by siRNA in cells treated with PARP inhibitors. The depletion of TIMELESS or TIPIN did not enhance further the DNA damage response induced by olaparib or talazoparib (Extended Data Fig. 5). Next, we examined if PARP1 would be hyper-activated in cells, in which TIMELESS was depleted, the rationale being that if PARP1 signals the presence of TRCs to the replisome by means of TIMELESS, then in the absence

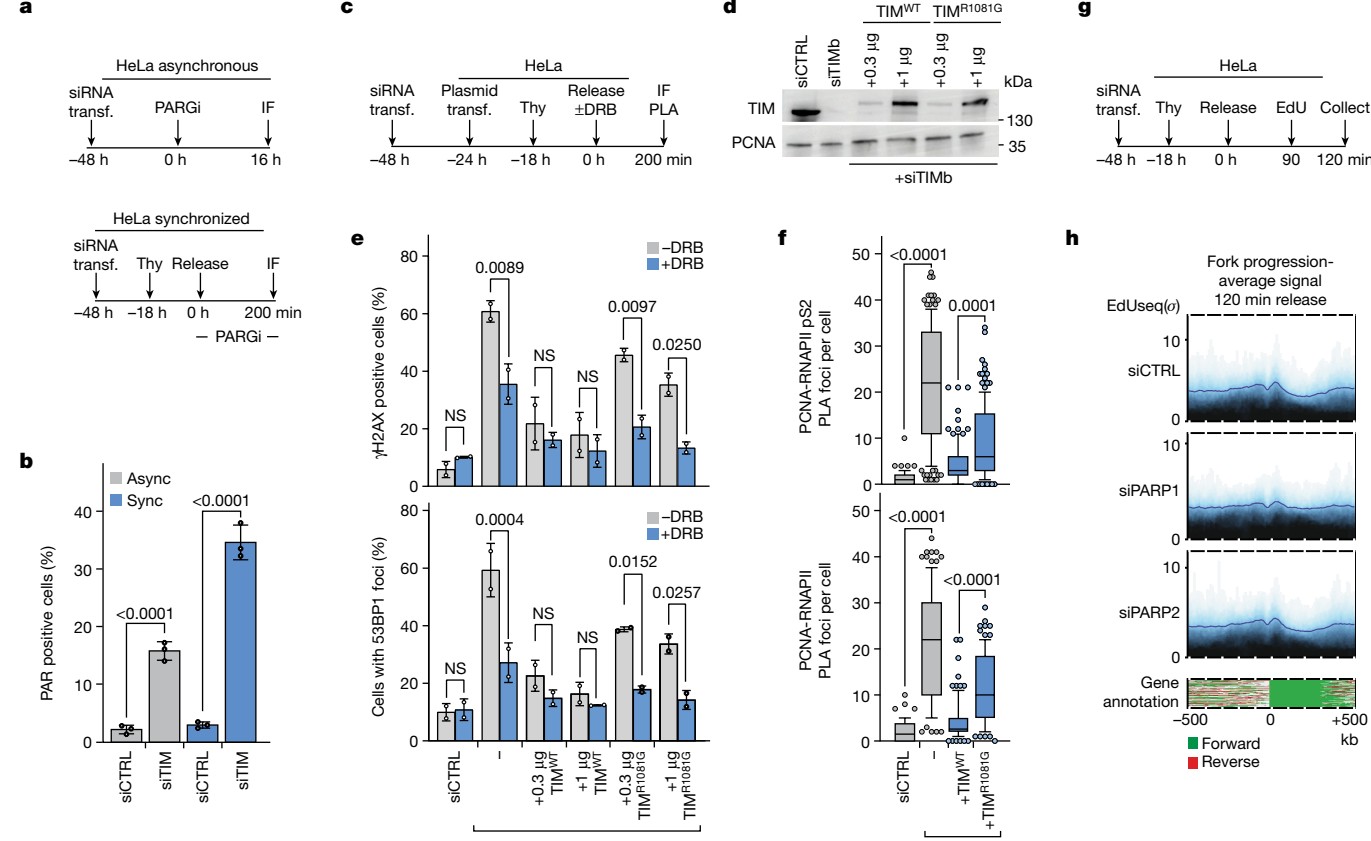

**Fig. 4 | The TIMELESS–PARP1 interaction is required to avert TRCs.**
**a**,**b**, PAR chain formation following depletion of TIMELESS (TIM) by siRNA.
**a**, Outline of the experiment. **b**, Means ± 1 s.d. of percentage of PAR positive
cells; $n = 3$ replicates; more than 512 cells per group (range 512–921); analysis
of variance (ANOVA). **c**–**f**, The TIMELESS–PARP1 interface is important for
averting TRCs. **c**, Outline of the experiment. Transfected plasmids drove
expression of exogenous wild-type (TIM^WT) or mutant (TIM^R1081G) TIMELESS,
while endogenous TIMELESS was depleted by siRNA (TIMb). **d**, Immunoblot
monitoring TIMELESS levels. Ectopic TIMELESS has a higher molecular weight
than endogenous TIMELESS. The TIMb siRNA targets only the endogenous
*TIMELESS* gene. PCNA served as loading control. **e**, Means ± 1 s.d. of percentage
of cells with more than 20 γH2AX or more than 20 53BP1 foci; $n = 3$ replicates;
more than 97 cells per group (range 97–445); ANOVA. **f**, Number of PLA foci per

cell; plots show medians and value ranges of 25–75 and 10–90%; filled circles
indicate the cells in the top and bottom deciles; $n = 2$ replicates; more than
89 cells per group (range 89–138); ANOVA. **g**,**h**, Increased fork progression
following depletion of PARP1, but not PARP2. **g**, Outline of the experiment.
**h**, Average EdUseq signal over large (more than 300 kb) transcribed genes
120 min after release into S phase. Data from three independent replicates have
been merged; the individual replicates are shown in Extended Data Fig. 6. The
genes were aligned by their transcription start site with their 5′–3′ orientation
from left to right. The lower panel shows gene annotation for each genomic
locus used to generate the EdUseq plot. Span of genomic regions, 1 Mb.
Async, asynchronous; CTRL, control; IF, immunofluorescence; NS, not
significant; PARGi, poly(ADP-ribose) glycohydrolase inhibitor; $\sigma$, sigma
value; Sync, synchronized; Thy, thymidine; transf., transfection.

of TIMELESS, TRCs would not be averted and PARP1 activity would be
augmented. Indeed, depletion of TIMELESS enhanced poly(ADP-ribose)
(PAR) chain formation in unsynchronized HeLa cells and, even more
so, in HeLa cells synchronized in early S phase (Fig. 4a,b).

To examine more directly the importance of a physical interac-
tion between TIMELESS and PARP1 in preventing TRCs, the endog-
enous TIMELESS and/or PARP1 proteins were depleted by siRNA and
exogenous wild-type or mutant versions of the above proteins were
expressed. The mutant versions had single amino acid substitutions
targeting the TIMELESS–PARP1 interface[37]. The transfected cells were
synchronized in G1/S, released into S phase, and 200 min later, DNA
damage response markers and TRCs were monitored by immunofluo-
rescence and PLAs, respectively (Fig. 4c). Consistent with the binding of
PARP1 to TIMELESS being functionally important, ectopic expression
of wild-type TIMELESS and PARP1 proteins prevented the induction
of TRCs and TRC-dependent DNA damage, whereas expression of the
mutant TIMELESS or PARP1 proteins did not (Fig. 4d–f and Extended
Data Fig. 5c,d).

Finally, we examined whether depletion of PARP1 or PARP2 affected
fork progression, similar to what we observed in cells depleted of

TIMELESS or TIPIN. We monitored EdU incorporation over large, tran-
scribed genes 120 min after release from a thymidine block (Fig. 4g).
Depletion of PARP1 accelerated fork progression to a similar extent to
the depletion of TIMELESS or TIPIN, whereas depletion of PARP2 had no
obvious effect (Fig. 4h and Extended Data Fig. 6). These results are con-
sistent with PARP1 signalling through TIMELESS the presence of TRCs
to the replisome, whereas PARP2 does not interact with TIMELESS[36,37].

## TRCs mediate the synthetic lethality of PARP inhibitors

The TRC-dependent DNA damage response induced by PARP inhibitors
raised the question of whether TRCs drive the synthetic lethality of
PARP inhibitors with HR deficiency. To help answer this question, DLD1
*BRCA2*^+/+ and *BRCA2*^−/− cells were released from a thymidine block into
S phase and PARP inhibitors were administered with or without DRB
either during early or late S phase (0–3.5 and 7–10.5 h after release from
the thymidine block, respectively). Twenty-four hours after release
from the thymidine block, when the cells should have progressed into
the next cell cycle, we scored for micronuclei, markers of aberrant

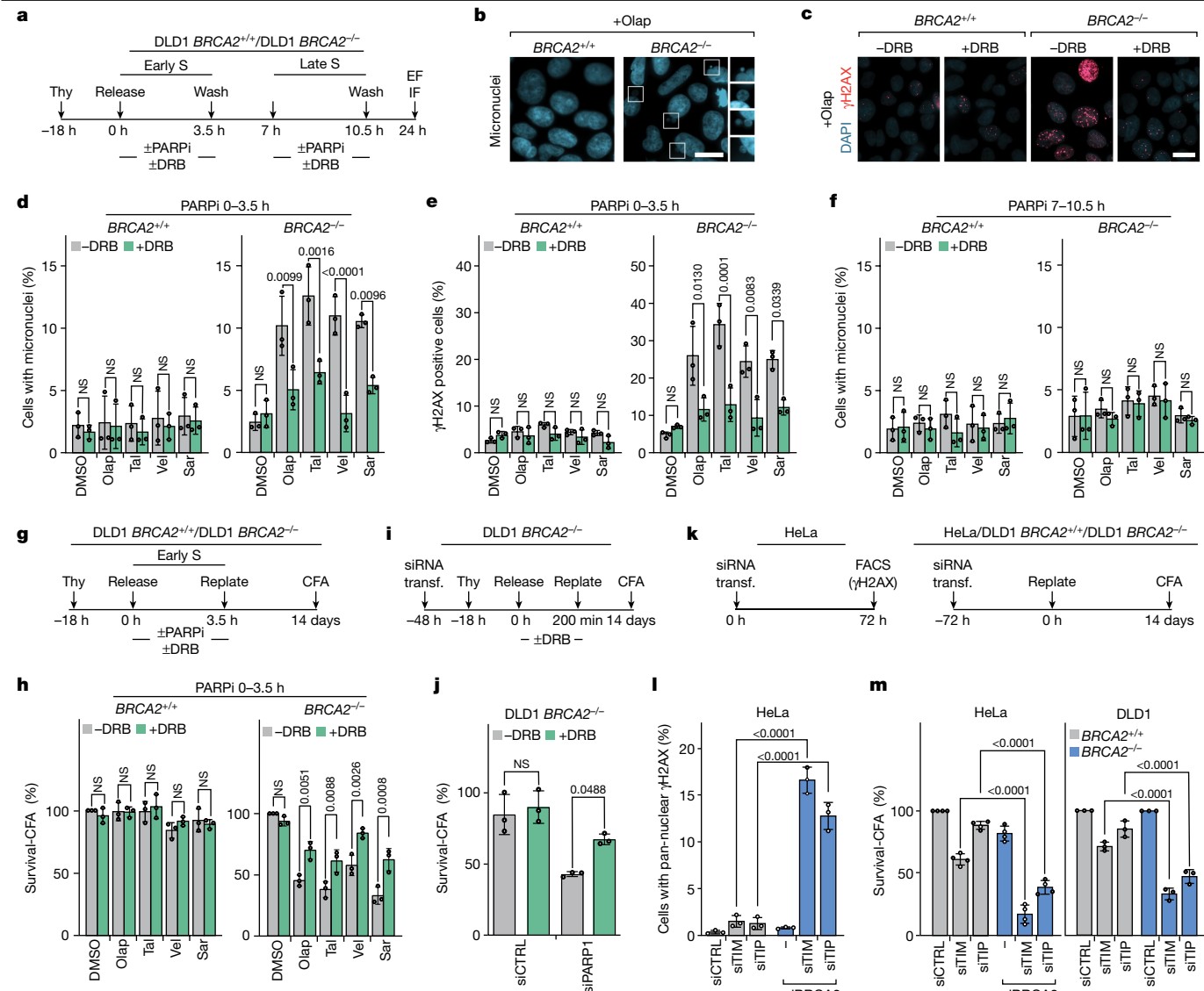

**Fig. 5 | Inhibition of transcription elongation confers resistance to PARP inhibitors. a–f**, DNA damage by PARP inhibitors (PARPi) is suppressed by inhibiting transcription elongation. **a**, Outline of the experiment. PARP inhibitors and, optionally, DRB were administered during early or late S phase (0–3.5 or 7–10.5 h after release from a thymidine (Thy) block, respectively). **b,c**, Representative images of cells treated with olaparib (Olap, 10 μM) during early S phase showing micronuclei (**b**) and γH2AX foci (**c**) 24 h after release from the thymidine block. **d–f**, Means ± 1 s.d. of percentage of cells with micronuclei (**d,f**) or more than 20 γH2AX foci (**e**) after treatment with PARP inhibitors in early (**d,e**) or late (**f**) S phase; $n = 3$ replicates; more than 250 cells per group (range 250–400); analysis of variance (ANOVA). **g,h**, Synthetic lethality induced by PARP inhibitors is alleviated by inhibiting transcription elongation. **g**, Outline of the experiment. **h**, Means ± 1 s.d. of percentage of surviving cells, as assessed by colony formation assay (CFA); DMSO-treated cells serve as reference; $n = 3$ replicates; ANOVA. **i,j**, Synthetic lethality induced by PARP1 depletion is alleviated by inhibiting transcription elongation. **i**, Outline of the experiment. **j**, Means ± 1 s.d. of percentage of surviving cells, as assessed by CFA; non-transfected cells serve as reference; $n = 3$ replicates; ANOVA. **k–m**, Depletion of TIMELESS (TIM) or TIPIN (TIP) are synthetic lethal with HR deficiency. **k**, Outline of the experiments. **l**, Means ± 1 s.d. of percentage of cells with pan-nuclear γH2AX staining; $n = 3$ replicates; ANOVA. **m**, Means ± 1 s.d. of percentage of surviving cells, as assessed by CFA; control-transfected cells serve as reference; $n = 3$ replicates; ANOVA. Scale bars for microscopy images, 10 μm. CTRL, control; EF, epifluorescence; FACS, fluorescence-activated cell sorting; IF, immunofluorescence; NS, not significant; Sar, saruparib (1 μM); Tal, talazoparib (100 nM); transf., transfection; Vel, veliparib (10 μM).

mitoses, and for γH2AX (Fig. 5). A significant fraction of the $BRCA2^{-/-}$ cells treated with PARP inhibitors in early but not late S phase scored positive for both markers; moreover, the emergence of these markers was suppressed by administering DRB together with the PARP inhibitors (Fig. 5b–f and Extended Data Fig. 7a,b). By contrast, the $BRCA2^{+/+}$ cells were generally devoid of micronuclei and γH2AX signal, indicating that any TRC-related DNA damage induced by the PARP inhibitors (Fig. 2) had been repaired after removal of the PARP inhibitors from the tissue culture media. Cell lethality of the $BRCA2^{-/-}$ cells, as determined by a colony formation assay 14 days after treatment with the PARP

inhibitors, paralleled the presence of micronuclei and γH2AX signal and was alleviated by DRB, whereas the $BRCA2^{+/+}$ cells survived well the PARP inhibitor treatment (Fig. 5g,h and Extended Data Fig. 7c,d). Cell lethality induced by PARP inhibitors in PEO1 HR-deficient ovarian cancer cells[40] was also reversed by DRB, whereas HR-proficient PEO4 ovarian cancer cells, derived from the same cancer as the PEO1 cells[40], were resistant to PARP inhibitors (Extended Data Fig. 7e,f). In further support of the notion that TRCs are synthetic lethal with HR deficiency, HR-deficient cells were more sensitive to the topoisomerase I inhibitor camptothecin than HR-proficient cells (Extended Data Fig. 7g,h).

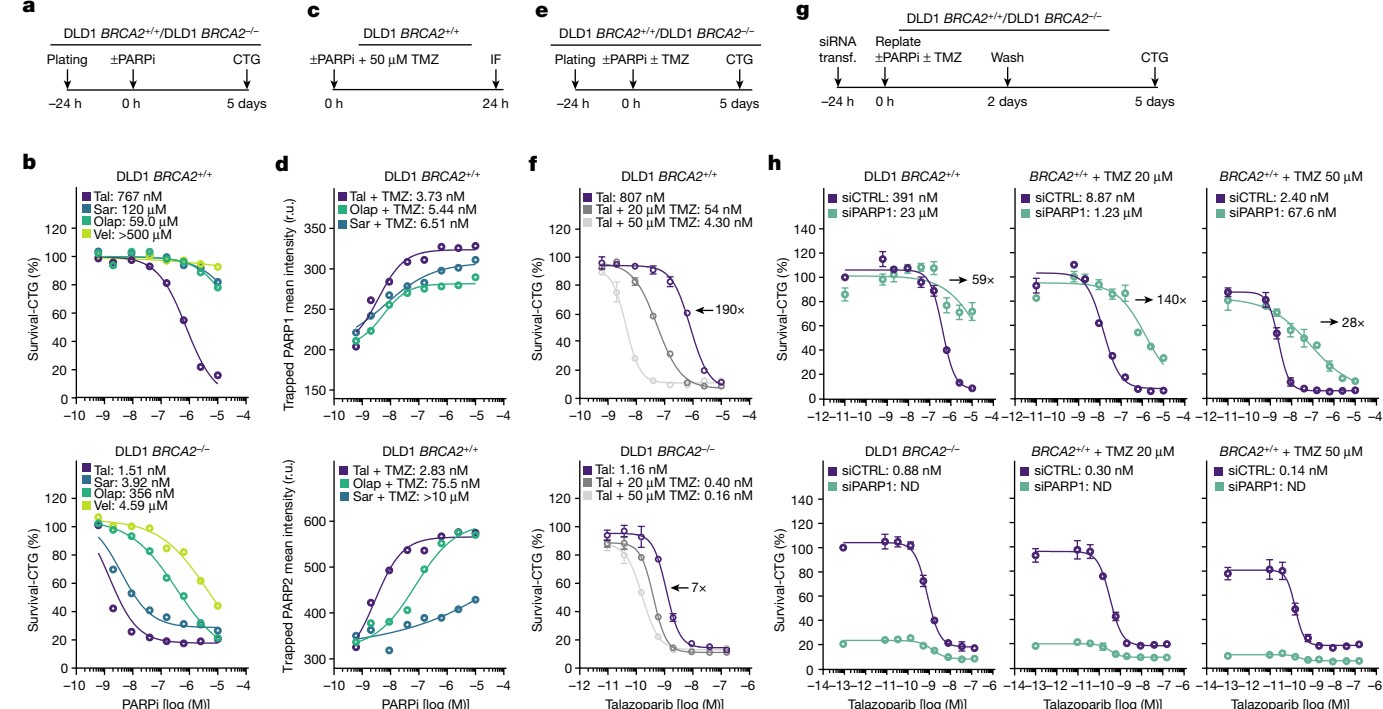

**Fig. 6 | PARP trapping reduces the selectivity of PARP inhibitors for HR deficiency. a,b**, Sensitivity of DLD1 *BRCA2*[+/+] and DLD1 *BRCA2*[−/−] cells to PARP inhibitors (PARPi). **a**, Outline of the experiment. Survival was assessed using the CellTiter-Glo (CTG) assay. **b**, Dose–response survival curves and calculated $EC_{50}$ values for the various PARP inhibitors. Data from one of *n* = 2 replicates. **c,d**, TMZ enhances trapping of PARP1 and PARP2. **c**, Outline of the experiment. **d**, Dose–response curves and calculated $EC_{50}$ values for trapping of PARP1 and PARP2; data from one of *n* = 2 replicates; for PARP1 > 120 (range 120–4,848) cells per data point; for PARP2 > 2,293 (range 2,293–5,377) cells per data point. **e,f**, TMZ reduces the selectivity of talazoparib (Tal) for HR deficiency. **e**, Outline of the experiment. **f**, Dose–response survival curves and calculated $EC_{50}$ values

for talazoparib-mediated lethality of DLD1 *BRCA2*[+/+] and DLD1 *BRCA2*[−/−] cells with and without added TMZ. Horizontal arrows indicate the fold-change in $EC_{50}$ values as a result of administering 50 μM TMZ. Data from one of *n* = 2 replicate experiments. **g,h**, Depletion of PARP1 renders HR-proficient cells resistant to talazoparib. **g**, Outline of the experiment. **h**, Dose–response survival curves and calculated $EC_{50}$ values for induction of lethality of DLD1 *BRCA2*[+/+] and DLD1 *BRCA2*[−/−] cells by talazoparib following depletion of PARP1 in the absence and presence of TMZ. Horizontal arrows indicate the fold-change in $EC_{50}$ values as a result of depleting PARP1; *n* = 2 replicates. CTRL, control; IF, immunofluorescence; ND, $EC_{50}$ values not determined; Olap, olaparib; r.u., relative units; Sar, saruparib; transf., transfection; Vel, veliparib.

Depletion of PARP1 by siRNA induces a TRC-dependent DNA damage response similar to the one induced by PARP inhibitors (Fig. 3g,h). We, therefore, examined whether depletion of PARPs by siRNA induced synthetic lethality with HR deficiency. We studied cancer cell lines that were naturally HR-deficient or in which HR deficiency was induced by targeting the *BRCA2* gene. Depletion of PARP1 or both PARP1 and PARP2 compromised the survival of the HR-deficient cancer cells, mirroring the effect of the PARP inhibitors, whereas depletion of PARP2 had no effect (Extended Data Fig. 8). The synthetic lethality induced by PARP1 depletion could be partially suppressed by treating the siRNA-transfected cells with DRB during early S phase (Fig. 5i,j).

Finally, because PARP1, TIMELESS and TIPIN function in the same pathway to prevent TRCs, we examined whether depletion of TIMELESS or TIPIN, similar to PARP1 depletion, were synthetic lethal with HR deficiency (Fig. 5k). Codepletion of BRCA2 and TIMELESS or BRCA2 and TIPIN led to a strong DNA damage response in HeLa cells, as revealed by pan-nuclear γH2AX staining, and to cell lethality; whereas, depletion of TIMELESS or TIPIN or BRCA2 on their own had a much smaller effect (Fig. 5l). Similarly, depletion of TIMELESS or TIPIN compromised the viability of DLD1 *BRCA2*[−/−] cells, while sparing the HR-proficient DLD1 *BRCA2*[+/+] cells (Fig. 5k,m).

## PARP trapping reduces the selectivity of PARP inhibitors

The experiments presented so far indicate that the synthetic lethality of PARP inhibitors with HR deficiency is due, at least in part, to TRCs and that TRCs can be induced without trapping PARPs on DNA.

To address this further, we determined dose–response curves by which the four PARP inhibitors induced lethality of DLD1 *BRCA2*[−/−] and *BRCA2*[+/+] cells (Fig. 6a,b). For the HR-deficient *BRCA2*[−/−] cells, the dose–response lethality curves (Fig. 6b) matched the curves for inhibition of PARP1 cellular enzymatic activity (Fig. 3b), but not the curves for PARP1 or PARP2 trapping (Fig. 3d and Extended Data Fig. 4d). The inverse was observed when we plotted the dose–response curves for the HR-proficient *BRCA2*[+/+] cells; here, the dose–response curves for induction of lethality (Fig. 6b) matched the curves for PARP1 and PARP2 trapping (Fig. 3d and Extended Data Fig. 4d).

To explore further the role of PARP trapping on survival of HR-deficient and HR-proficient cells, we co-administered PARP inhibitors with temozolomide (TMZ). TMZ, a DNA alkylating prodrug, induces DNA nicks in which PARPs are recruited and, potentially, trapped[17,41,42]. We determined dose–response curves for trapping of PARP1 and PARP2 on chromatin in cells treated with talazoparib, saruparib or olaparib in the presence of 50 μM TMZ. Addition of TMZ enhanced significantly PARP trapping by all PARP inhibitors (Fig. 3d versus Fig. 6c,d).

Next we examined whether the increased PARP trapping affected the dose–response curves for induction of lethality in DLD1 *BRCA2*[−/−] and DLD1 *BRCA2*[+/+] cells treated with PARP inhibitors. TMZ, at 50 μM, enhanced 190-fold the potency by which talazoparib induced lethality of HR-proficient cells, whereas the corresponding effect for HR-deficient cells was only sevenfold, meaning that the selectivity for HR deficiency was significantly reduced (Fig. 6e,f). Similar effects were observed with saruparib and olaparib (Extended Data Fig. 9a,b).

Following on from the above observations, we determined dose–response lethality curves for talazoparib-treated DLD1 *BRCA2*[−/−] and DLD1 *BRCA2*[+/+] cells, in which endogenous PARP1 was depleted by siRNA (Fig. 6g). Depletion of PARP1 by siRNA severely reduced the viability of DLD1 *BRCA2*[−/−] cells (Fig. 6h), suggesting that the loss of PARP1 enzymatic activity is sufficient to induce lethality in HR-deficient cells. By contrast, PARP1-depleted DLD1 *BRCA2*[+/+] cells were highly resistant to talazoparib, in accordance with its on-target inhibition (Fig. 6h). Finally, consistent with lethality being linked to a persistent DNA damage response, depletion of PARP1 suppressed the DNA damage response in talazoparib-treated HR-proficient, but not HR-deficient, cells (Extended Data Fig. 9c–e).

## Discussion

Several mechanisms have been proposed to explain why PARP inhibitors are synthetic lethal with HR deficiency. The now-favoured mechanism posits that PARP inhibitors trap PARPs on DNA; in turn, the trapped PARPs block fork progression, leading to DNA DSBs that require HR for repair[7]. Here, we propose a TRC mechanism; specifically, that PARP1 signals the presence of impending TRCs to TIMELESS and TIPIN, pausing the replisome until the TRCs are resolved. If PARP1 or TIMELESS and TIPIN fail to perform their function, TRCs lead to DNA damage that requires HR for repair. The main difference between the two proposed mechanisms is the nature of the object with which the replisomes collide: trapped PARPs versus transcription elongation complexes. Evidence supporting the revised mechanism is that the synthetic lethality of PARP inhibitors with HR deficiency can be alleviated by inhibiting transcription elongation and that depleting TIMELESS, TIPIN or PARP1 by siRNA is synthetic lethal with HR deficiency.

Recent observations by others are consistent with the TRC mechanism proposed here. Depletion of TIMELESS induces formation of R-loops[43], consistent with induction of TRCs; PARP1 binds to R-loops[44]; PARP inhibitors enhance replication fork speed[45] and HR repairs DNA damage induced by TRCs[33,34].

Our study did not address the mechanism(s) by which PARP1 senses TRCs. One possibility is that PARP1 is recruited by R-loops[44] or other DNA structures generated at sites of impending TRCs. A second, non-mutually exclusive, possibility is that PARP1 is activated by topoisomerase I, which is present at sites of TRCs to resolve DNA supercoiling[46,47]. Supporting the latter hypothesis, PARP1 and topoisomerase I function together to maintain fork stability under conditions of DNA replication stress[48] and, in budding yeast, Tof1, the orthologue of TIMELESS, functions together with topoisomerase I to pause the replisome ahead of replication fork barriers[49]. Yet another possibility is that the TRCs are sensed by TIMELESS, which then recruits PARP1, along the lines proposed for a role of TIMELESS in sensing DSBs[36,37].

An improved understanding of how PARP inhibitors target HR-deficient cells, could help guide their future clinical development, especially as it relates to isoform specificity and trapping potential. All PARP inhibitors, used at present in the clinic, inhibit both PARP1 and PARP2. However, to protect replication forks from TRCs, the critical family member might be PARP1, because TIMELESS binds preferentially to this family member[36,37]. Thus, a PARP1-selective inhibitor, such as saruparib, might suffice for inducing synthetic lethality with HR deficiency in the clinic, as demonstrated here with cell lines.

The TRC mechanism may help inform whether PARP inhibitors should have high or low trapping activity. The four PARP inhibitors we studied here showed a very good correlation between induction of lethality of HR-deficient cells and inhibition of PARP enzymatic activity and no correlation with trapping potential. Moreover, depletion of PARP1 by siRNA induced lethality of HR-deficient cells. Consistent with the model that trapping may not be important for therapeutic efficacy, one of the original reports describing the synthetic lethality between PARP inhibitors and HR deficiency included experiments in which PARP1 was depleted by siRNA[5]. Moreover, the studies demonstrating that PARP1 trapping is required for PARP inhibitors to be cytotoxic examined mostly HR-proficient cells and often stimulated PARP trapping by combining PARP inhibitors with TMZ or other DNA damaging agents[21–23,42,50–52].

Our findings raise the question whether modulating PARP trapping can enhance the therapeutic window of PARP inhibitors in the clinic[3,17,18]. PARP inhibitors differ in their trapping potential, so this is a parameter that can be optimized independently of inhibitory activity. We propose that reducing trapping potential may decrease the toxicity of PARP inhibitors without compromising efficacy.

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

## Methods

### Cell culture and drug treatments

The human cervical cell line HeLa and the human osteosarcoma cell line U2OS were purchased from the American Type Culture Collection (ATCC) (CCL-2 and HTB-96 respectively) and were grown under standard conditions (37 °C and 5% $CO_2$) in Dulbecco's modified Eagle's medium (DMEM) (Invitrogen, catalogue no. 11960) supplemented with 10% fetal bovine serum (FBS) (Invitrogen, catalogue no. 10500) and penicillin–streptomycin–glutamine (Invitrogen, catalogue no. 10378-016). The hTERT-RPE1 retinal pigment epithelial cells (CRL4000, ATCC), were cultured in DMEM and Ham's F-12 (Invitrogen, catalogue no. 12634-010), supplemented with 10% FBS and penicillin–streptomycin–glutamine. DLD1 *BRCA2*[+/+] (ATCC, CCL-221) and DLD1 *BRCA2*[−/−] (Horizon, catalogue no. HD 105-007) were cultured in Roswell Park Memorial Institute (RPMI) 1640 medium (Thermo Scientific, catalogue no. 11875093) supplemented with 10% FBS and penicillin–streptomycin–glutamine. HeLa cells expressing FLAG-tagged RNase H1 in a doxycycline (DOX)-dependent manner[53] were grown in DMEM supplemented with 10% Tet system-approved FBS (Biowest, catalogue no. S181T) and antibiotics. Expression of FLAG-RNase H1 was induced with addition of 2 μg ml$^{-1}$ DOX (Sigma-Aldrich, catalogue no. D9891) in the medium for 18 h. U2OS T-Rex cells expressing catalytically inactive GFP-RNaseH1$^{D201N}$ in a DOX-dependent manner[54] were grown in DMEM supplemented with 10% Tet system-approved FBS, 1 μg ml$^{-1}$ puromycin (Sigma, catalogue no. P8833) and 50 μg ml$^{-1}$ hygromycin B (Thermo Scientific, catalogue no. 10687010). Expression of catalytically inactive GFP-RNaseH1$^{D201N}$ was induced by addition of 1 ng ml$^{-1}$ DOX for 18 h. The human non-small cell lung carcinoma H1299 cells, expressing a short-hairpin RNA against *BRCA2* in a DOX-dependent manner[55], were grown in DMEM supplemented with 10% Tet system-approved FBS and antibiotics. Expression of shBRCA2 was induced by addition of 2 μg ml$^{-1}$ DOX in the medium. The human PEO1 and PEO4 ovarian cancer cell lines[40] (provided by I. Labidi-Galy) were grown in RPMI 1640 supplemented with 2 mM sodium pyruvate and antibiotics; the human colorectal carcinoma HCT116 cells (ATCC, catalogue no. CCL-247) were grown in McCoy's 5A medium (Thermo Scientific, catalogue no. 16600082) containing 10% FBS and antibiotics; the human OVSAHO ovarian cancer cell line was purchased from Sigma-Aldrich (catalogue no. SCC294) and was grown in RPMI 1640 medium supplemented with 10% FBS and antibiotics. All cell lines were routinely tested for the absence of mycoplasma contamination using the MycoGenie Rapid MycoPlasma Detection Kit (AssayGenie, catalogue no. MORV001) and found negative. Drugs and chemical compounds used in this study were purchased from the following sources: thymidine (Sigma-Aldrich catalogue no. T1895), EdU (Thermo Fisher Scientific, catalogue no. A10044), 5-ethynyl uridine (EU) (Jena Biosciences, catalogue no. CLK-N002-10), camptothecin (Sigma-Aldrich, catalogue no. C9911), 5,6-dichlorbenzimidazol 1-β-D-ribofuranosid (DRB; Sigma-Aldrich, catalogue no. D1916), cordycepin (Tocris, catalogue no. 2294), triptolide (Tocris, catalogue no. 3253), olaparib (Selleckchem, catalogue no. S1060), talazoparib (Selleckchem, catalogue no. S7048), veliparib (Selleckchem, catalogue no. S1004), saruparib (Selleckchem, catalogue no. S9875), hydrogen peroxide (Sigma-Aldrich, catalogue no. H3410), RO-3306 (Sigma-Aldrich, catalogue no. SML0569), TMZ (Sigma-Aldrich, catalogue no. T2577), nocodazole (Tocris, catalogue no. 1228) and PARGi (Tocris, catalogue no. 7006). DRB, cordycepin and triptolide were used at concentrations of 75, 50 and 1 μM, respectively.

### PARP1 and PARP2 biochemical assay

PARP1 and PARP2 activity in presence of increasing concentrations of PARP inhibitors was measured using PARP1 and PARP2 colorimetric assay kits (BPS Bioscience, catalogue no. 80580-1) according to the manufacturer's instructions. The assays were performed in triplicate. PARP inhibitors were dispensed with an acoustic liquid dispenser (Gen5-Acoustic Transfer System; EDC Biosystems). The final concentration of the PARP inhibitors ranged from 0.2 to 100 nM using twofold dilution steps. The absorbance at 450 nm was measured using a Spark 10M microplate reader (Tecan).

### siRNA and plasmid transfections

Transfections of siRNAs (at a final concentration of 40 nM) were performed with the cells at 60% confluency using INTERFERin (Polyplus, catalogue no. 409-01) or Lipofectamine RNAiMAX Transfection Reagent (Thermo Scientific, catalogue no. 13778075) according to the manufacturer's instructions. TIMELESS depletion was achieved by transfection of 10 nM siRNA. Medium change was performed 24 h after siRNA transfection. The following siRNAs were used: negative control (AllStars Negative Control siRNA, Qiagen, catalogue no. 1027281), siTIM (TIMELESS; Qiagen, catalogue no. SI04142194), siTIMb (Dharmacon, 5′-GUAGCUUAGUCCUUUCAAATT-3′), siTIP (TIPIN; Invitrogen, catalogue no. S29864), siPARP1 (Qiagen, catalogue nos. SI02662989 and SI02662996) and (Invitrogen, catalogue no. s1097), siPARP1b (Dharmacon, 5′-GGAAAGAUGUUAAGCAUUUTT-3′ and 5′-CAUGGGAGCUCUUGAAAUAUT-3′ and 5′-AGAAAAGGCUGGAGAG AGATT-3′), siPARP2 (Invitrogen, catalogue no. S19504), siBRCA2 (Qiagen, catalogue no. SI02653434). Efficiency of siRNA-mediated depletion was performed 72 h after transfection by western blotting. Empty vector, full length GFP-TIMELESS WT, GFP-TIMELESS R1081G, full length FLAG-PARP1 WT and FLAG-PARP1 D993G plasmids[37] were transfected using the FuGENE HD transfection reagent (Promega, catalogue no. E2311) according to manufacturer's instructions. Efficiency of plasmid transfection was performed by western blot for detection of TIMELESS and PARP1 proteins.

### Immunoblotting

Protein cell extracts were resolved by SDS–PAGE in precast protein gels (4–15% Mini-PROTEAN TGX, Bio-Rad, catalogue no. 4561083, or 3–8% Criterion XT Tris-Acetate Protein Gel, Bio-Rad, catalogue no. 3450129) and transferred onto polyvinylidene fluoride membranes. Membranes were blocked with 5% milk powder diluted in TBS-Tween 20 (0.01%) for 1 h at room temperature. Incubation with primary antibodies in blocking solution was applied for 1 h at room temperature. The following primary antibodies were used for western blot analysis: PCNA mouse monoclonal (1:1,000, clone PC10, Millipore, catalogue no. MABE288); alpha-Tubulin mouse monoclonal (1:1,000, Calbiochem, catalogue no. CP06); GAPDH mouse monoclonal (1:10,000, Abcam, catalogue no. ab8245); TIMELESS rabbit polyclonal (1:1,000, Abcam, catalogue no. ab109512); TIPIN rabbit polyclonal (1:250, Bethyl Laboratories, catalogue no. A301-474A); PARP1 rabbit polyclonal (1:1,000, Abcam, catalogue no. ab32138); PARP2 rabbit polyclonal (1:500, Active Motif, catalogue no. 39743); BRCA2 mouse monoclonal (1:1,000, Calbiochem, catalogue no. OP95); RNase H1 rabbit polyclonal (1:500, ProteinTech, catalogue no. 15606-1-AP); FLAG mouse monoclonal (1:1,000, Sigma-Aldrich, catalogue no. M2 F1804) and GFP rabbit polyclonal (1:500, Abcam, catalogue no. ab290). Following incubation with primary antibodies, three washes with TBS-Tween 20 (0.01%) were performed. Membranes were incubated with secondary horseradish antimouse or antirabbit peroxidase-coupled antibodies IgG (1:2,500, Promega, catalogue nos. W401B and W402B, respectively) for 1 h at room temperature, before detection by ECL-based chemiluminescence. Uncropped western blot images are provided in Supplementary Fig. 1.

### Flow cytometry

Following siRNA transfection or drug treatment and, optionally, as indicated, following pulse-labelling with 10 μM EdU for 30 min, cells were collected by trypsinization and fixed in 90% methanol overnight at −20 °C. EdU detection was performed using the Click-it EdU Alexa Fluor 647 Flow Cytometry Assay Kit (Invitrogen catalogue no. C-10424) according to the manufacturer's instructions. Detection of γH2AX

phosphorylation was performed using the Guava Histone H2AX Phosphorylation Assay Kit (Luminex, catalogue no. FCCS100182) according to the manufacturer's instructions. The genomic DNA was stained by incubating the cells in PBS containing RNase (Roche, catalogue no. 11119915001) and propidium iodide (Sigma-Aldrich catalogue no. 81845). EdU-DNA-γH2AX profiles were acquired by flow cytometry (Gallios, Beckman Coulter); more than 20,000 cells were analysed per sample using Kaluza software (Beckman Coulter). The gating strategy is provided in Supplementary Fig. 2.

## EdUseq
The EdUseq protocol was performed as previously described[35]. Briefly, HeLa cells were transfected with siRNA; 30 h later thymidine (Sigma-Aldrich) at 2 mM final concentration was added for 18 h, at which time the cells had reached 70–80% confluency. The cells were washed four times with warm PBS and released in fresh medium for 90 or 120 min. EdU (25 μM) was added 30 min before the cells were collected, and the cells were then fixed with 90% ice-cold methanol overnight. Cells were stored until processed for isolation of EdU-labelled DNA. Following fixation, the cells were permeabilized with 0.2% triton X in PBS; then, the EdU incorporated into genomic DNA was coupled to a cleavable biotin-azide linker (Azide-PEG(3+3)-S-S-biotin; Jena Biosciences, catalogue no. CLK-A2112-10), using the reagents of the Click-it Kit (Invitrogen, catalogue no. C-10424). Extraction of genomic DNA was performed with phenol-chloroform ethanol precipitation, followed by isolation of EdU-labelled DNA. Briefly, genomic DNA was sonicated to 100–500 bp nucleotide-long fragments using a bioruptor sonicator (Diagenode). EdU-labelled DNA fragments were captured on Dynabeads MyOne streptavidin C1 (Invitrogen, catalogue no. 65001). The beads were washed three times with Binding and Washing Buffer 1× (5 mM Tris-HCl pH 7.5, 0.5 mM EDTA, 1 M NaCL, 0.5% Tween 20) and then were resuspended to twice the original volume with Binding and Washing Buffer 2×, mixed with an equal volume of sonicated EdU-labelled DNA incubated for 15 min on a rotating wheel at room temperature. Following three washes of the beads with Binding and Washing Buffer 1× and once with TE (10 mM Tris-HCl pH 8, 1 mM EDTA), the EdU-labelled DNA was eluted by incubating the streptavidin beads with 2% β-mercaptoethanol (Sigma, catalogue no. M6250) for 1 h at room temperature. The eluted DNA was used for library preparation using the TruSeq ChIP Sample Prep Kit (Illumina, catalogue no. IP-202-1012). High-throughput 100-base-pair single-end sequencing was performed on an Illumina Hi-Seq 4000 sequencer.

## EdUseq data processing
Sequencing reads were aligned on the non-masked human genome assembly (GRCh37/hg19) using the Burrows–Wheeler Aligner software as described previously[35,56]. Only the reads with the highest quality score were retained. Previously described custom Perl scripts were used to assign the aligned reads to 10 kb genomic bins. Sigma ($\sigma$) values were calculated as the normalized number of reads per bin divided by its standard deviation. The data were visualized using previously described scripts[35]. Assignment of replication timing was performed with REPLI-seq data generated previously[35].

## Cell viability and clonogenic assays
Viability assays were performed with DLD1 BRCA2$^{+/+}$ and DLD1 BRCA2$^{-/-}$ cells treated with various inhibitors and DNA damaging agents, and following siRNA transfections, as indicated. In brief, 2,000 cells per well were seeded in Advanced TC 96-well microplates; 24 h later, PARP inhibitors were dispensed using a D300e digital dispenser (Tecan) at final concentrations ranging from 0.6 nM to 10 μM using fourfold dilution steps. The cells were incubated with the compounds for 5 days before adding CellTiter-Glo 2.0 reagent (Promega, catalogue no. G9242) to each well, according to the manufacturer's instructions. Luminescence was measured using a Spark 10M microplate reader (Tecan).

Clonogenic assays were performed with a variety of cell lines following siRNA transfection or drug treatment. Briefly, following the indicated treatments, the cells were replated in triplicate in six- or 12-well plates (500–3,000 cells per well, depending on cell line) and cultured for an additional 14 days (or more, depending on cell line) in fresh medium. The cell culture medium was changed every two days. At the end of the experiment, medium was removed, and cells were rinsed with PBS and stained with 0.5% (w/v) crystal violet (Sigma-Aldrich) in 20% (v/v) methanol for 30 min in the dark. The staining agent was removed and the plates were rinsed three times in ddH$_2$0, air-dried and the cell colonies were counted.

## Immunofluorescence assays
Cells were seeded onto autoclaved 12 mm glass coverslips or multiwell plates (μ-Plate 96 Wells, catalogue no. 89626) at 70–90% confluency. Following any indicated treatment, the cells were pre-extracted for 2 min with ice-cold 1× PBS containing 0.2% (v/v) Triton X-100 and then fixed with 4% formaldehyde for 10 min at room temperature. After three washes with 1× PBS, the cells were permeabilized in 1× PBS containing 0.2% (v/v) Triton X-100 for 15 min at room temperature. For detection of trapped PARP1 and PARP2, the cells were pre-extracted with cold cytoskeleton buffer (0.5% Triton X-100, 10 mM PIPES pH 6.8, 3 mM MgCl$_2$, 200 mM NaCl, 300 mM sucrose) for 10 min at 4 °C, followed by fixation with ice-cold methanol for 15 min at −20 °C. Following three washes with PBS, the cells were blocked with 5% BSA/1× PBS solution for 1 h at room temperature. Then, coverslips or multiwell plates were incubated for 2 h at room temperature with primary antibodies diluted in 5% BSA/1× PBS. Following incubation with primary antibodies, coverslips or multiwell plates were washed three times with 1× PBS and incubated for 1 h at room temperature with secondary antibodies diluted in 5% BSA/1× PBS. After three washes with 1× PBS, incubation with 1 μg ml$^{-1}$ 4,6-diamidino-2-phenylindole (DAPI)/1× PBS for 15 min in dark at room temperature was performed. Then, three washes with 1× PBS were performed and coverslips were mounted on slides using the Fluoromount-G (Thermo Fisher Scientific, catalogue no. 00-4958-02). For multiwell plates, following incubation with DAPI, 1× PBS was added in the wells. The primary antibodies used for the immunofluorescence were: γH2AX (S139) mouse monoclonal (1:1,000, clone JBW301, Millipore, catalogue no. 05-636); RAD51 rabbit polyclonal (1:1,000, Bioacademia, catalogue no. 70-002); 53BP1 rabbit polyclonal (1:1,000, Novus Biologicals, catalogue no. NB100-304); poly (ADP-ribose) mouse monoclonal (1:500, Trevigen, catalogue no. 4335-MC-100 and 1-500, Enzo Life Sciences, catalogue no. ALX-804-220-R100); PARP1 rabbit polyclonal (1:1,000, ProteinTech, catalogue no. 13371–1-AP); PARP2 rabbit polyclonal (1:1,000, Active Motif, catalogue no. 39743). Secondary antibodies used: Alexa Fluor 488 goat-antirabbit IgG (1:500, Invitrogen, catalogue no. A110334); Alexa Fluor 488 goat-antimouse IgG (1:500, Invitrogen, catalogue no. A11001); Alexa Fluor 594 goat-antirabbit IgG (1:500, Invitrogen, catalogue no. A11037); Alexa Fluor 594 goat-antimouse IgG (1:500, Invitrogen, catalogue no. A11005); Alexa Fluor 647 goat-antirabbit IgG (1:500, Invitrogen, catalogue no. A21244); Alexa Fluor 647 goat-antimouse IgG (1:500, Invitrogen, catalogue no. A21235). For micronuclei detection, following any indicated treatment, cells were fixed with 4% paraformaldehyde for 15 min at room temperature and then permeabilized in PBS containing 0.2% Triton X-100 for 10 min at room temperature. Nuclei were countestained with 1 μg ml$^{-1}$ DAPI for 1 min at room temperature in dark, washed three times with PBS and mounted with Fluoromount-G.

## In situ PLA
Following the indicated treatments, cells grown on coverslips were washed twice with 1× PBS and pre-extracted for 10 min with ice-cold 1× PBS containing 0.5% (v/v) Triton X-100 and protease inhibitor cocktail (Complete, EDTA-free; Roche); they were then washed twice

with 1× PBS and fixed with 4% (v/v) formaldehyde for 10 min at room temperature, followed by two washes with 1× PBS. The cells were then incubated with 1× PBS containing 0.2% (v/v) Triton X-100 for 10 min at room temperature, washed again twice with 1× PBS and blocked with 5% BSA/1× PBS solution for 1 h. The coverslips were then incubated O/N at 4 °C with primary antibodies diluted in 5% BSA/1× PBS. Following incubation with primary antibodies, the coverslips were washed twice with 1× PBS and PLA was performed using Duolink PLA technology (Sigma-Aldrich, catalogue no. DUO92008) according to the manufacturer's instructions. Briefly, coverslips were incubated with antirabbit PLUS and antimouse MINUS PLA probes (Sigma-Aldrich, catalogue no. DUO92002 and DUO92004 respectively) for 1 h at 37 °C, followed by two washes in Wash Buffer A (0.01 M Tris, 0.15 M NaCl and 0.05% Tween 20, pH 7.4) for 5 min. Then, PLA probes were ligated for 30 min at 37 °C, followed by two washes for 5 min in Wash Buffer A and amplification using the Duolink In Situ Detection Reagents Red (Sigma-Aldrich, catalogue no. DUO92008), performed at 37 °C for 100 min. After amplification, the coverslips were washed twice in Wash Buffer B (0.2 M Tris and 0.1 M NaCl, pH 7.5) for 10 min and then incubated with 1 mg ml$^{-1}$ DAPI/1× PBS for 15 min in the dark at room temperature. Finally, the coverslips were washed three times with 1× PBS and mounted on slides using Fluoromount-G. Primary antibodies used: RNAPII, H5 (1:500, BioLegend, catalogue no. 920204), RNAPII, CTD4H8 (1:500, Millipore, catalogue no. 05-623) and PCNA (1:500, Abcam, catalogue no. ab18197).

## Quantification of nascent RNA production by EU labelling

Cells grown on multiwell plates were pulse-labelled with 1 mM EU for 30 min, washed twice with 1× PBS and fixed in 4% formaldehyde for 10 min at room temperature. After three washes with 1× PBS, the cells were permeabilized with 1× PBS containing 0.2% (v/v) Triton X-100 for 10 min at room temperature. EU incorporation was detected with Click-iT EU Alexa fluor 488 Imaging Kit (Thermo Fisher Scientific, catalogue no. C10329). Multiwell plates were incubated with Hoechst 33342 for 15 min in the dark at room temperature and subsequently washed three times with 1× PBS.

## Detection of EdU incorporation in mitosis

For detection of mitotic EdU foci, cells, cultured on coverslips, were synchronized at the G1/S transition with 2 mM thymidine (Sigma-Aldrich, catalogue no. T1895) for 18 h, washed three times with 1× PBS and released in fresh medium containing 6 µM RO-3306 (Sigma-Aldrich, catalogue no. SML0569) for 11 h. The cells were then washed three times with warm medium and released in medium containing 100 ng ml$^{-1}$ nocodazole (Tocris, catalogue no. 1228) and 20 µM EdU (Invitrogen, catalogue no. A10044) for 1 h. The cells were then fixed and permeabilized with 4% paraformaldehyde, 20 mM HEPES, 10 mM EGTA, 0.2% Triton X-100, 1 mM MgCl$_2$ for 20 min at room temperature and then washed three times with 1× PBS. EdU incorporation was performed using the Click-it EdU Alexa Fluor 647 Kit (Invitrogen, catalogue no. C10340), after which the cells were washed twice with 1× PBS, incubated with DAPI (0.5 mg ml$^{-1}$, Thermo Fisher, catalogue no. D1306) in 1× PBS for 15 min at room temperature, washed three times with PBS and rinsed in distilled water. The coverslips were then mounted on slides using ProLong Gold Antifade Mountant (Thermo Fisher, catalogue no. P10144). Quantification of EdU foci on metaphase chromosome spreads was performed manually.

## Image acquisition and analysis

Images from coverslips were acquired with a Zeiss Imager M2 AX10 microscope equipped with ApoTome2 and a Plan-APOCHROMAT ×100/1.4 oil immersion objective, using the ZEN3.4 (blue edition) software. Images were analysed with ZEN3.4 (blue edition) or ImageJ/FIJI software (National Healthcare Institute, USA). The threshold to determine whether a cell was positive for γH2AX or 53BP1 foci was set

at 20 foci per nucleus and for RAD51 at ten foci per nucleus. Micronuclei, PLA foci and GFP-RNaseH1$^{D210N}$ foci were quantified manually. Automated, multi-channel image acquisition of multiwell plates was performed in an unbiased fashion with an ImageXpress spinning disc confocal microscope (Molecular Devices), equipped with a sCMOS camera (Andor), and with a Nikon ×20 water 1.20 numerical aperture or ×60 water immersion 1.20 numerical aperture objective. The spinning disc confocal images were analysed using MetaXpress Custom Module Editor. The analysis pipeline started with the detection of the nuclei using the DAPI channel. This mask was then applied to quantify pixel intensities for γH2AX, PARP1/2 and EU incorporation for each individual cell. A light deconvolution was applied on the RAD51 and 53BP1 channel to ease the detection of the small foci. The segmentation of foci was then performed using shape, size and intensity above local background parameters. The masked foci were then attributed to their corresponding nuclei before the quantification of relevant parameters. Quantified values for each cell, were exported and were subsequently used to generate graphs using GraphPad Prism 9 software.

## Statistical analysis

Statistical analysis was performed using GraphPad Prism 9 software (v.9.4.1). Detailed description of means or medians, error bars and the number replicates and/or cells analysed is reported in the figure legends. Statistical differences for grouped analyses were performed using repeated-measures one-way analysis of variance (ANOVA) followed by a Tukey's multiple comparisons test. Statistical test results are provided as $P$ values in the figures. Dose–response curves were plotted using GraphPad Prism using as model the concentration of the inhibitor versus response, variable slope (four parameters). No statistical methods were used to determine the size sample size before conducting experiments. Experiments were not randomized and the investigators were not blinded to allocation. Data were assembled into figures using Adobe Illustrator CS6.

## Reporting summary

Further information on research design is available in the Nature Portfolio Reporting Summary linked to this article.

## Data availability

The fastq sequencing data and associated information described in this study have been deposited in the Sequence Read Archive with Gene Expression Omnibus accession number GSE220223. The EUseq data used in this study were previously published[56]. Unprocessed images of western blots and the gating strategy for the flow cytometry experiments are provided as Supplementary Information. All information supporting the conclusions are provided with the paper.

## Code availability

Computer codes and data files used to process and plot the data are available from our previous publications[35,56].

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

**Acknowledgements** We thank the High-Throughput/Content Screening facility ACCESS and the Flow Cytometry and Genomics Platforms of the University of Geneva. This work was

supported by grants from the European Commission (REPLISTRESS) and the Swiss National Science Foundation (grant nos. 182487 and 186230).

**Author contributions** T.D.H. conceived the study and supervised the project. M.P., G.G.R., S.K.S. and T.D.H. designed the experiments. T.D.H. and M.P. wrote the manuscript. M.P. performed the cell-based experiments with the contribution of A.K. and G.G.R. M.P. and A.K. processed samples for sequencing. G.G.R., A.F. and L.G.I. performed the in vitro biochemical and biophysical experiments. M.P., G.G.R., V.S.D., S.K.S. and T.D.H. analysed the data. T.D.H. and V.S.D. performed the bioinformatic analyses. All authors commented on the manuscript.

**Funding** Open access funding provided by University of Geneva.

**Competing interests** T.D.H. and S.K.S. are founders and stockholders of FoRx Therapeutics. S.K.S., G.G.R., A.F. and L.G.I. are employees of FoRx Therapeutics. The other authors declare no competing interests.

**Additional information**
**Correspondence and requests for materials** should be addressed to Thanos D. Halazonetis.

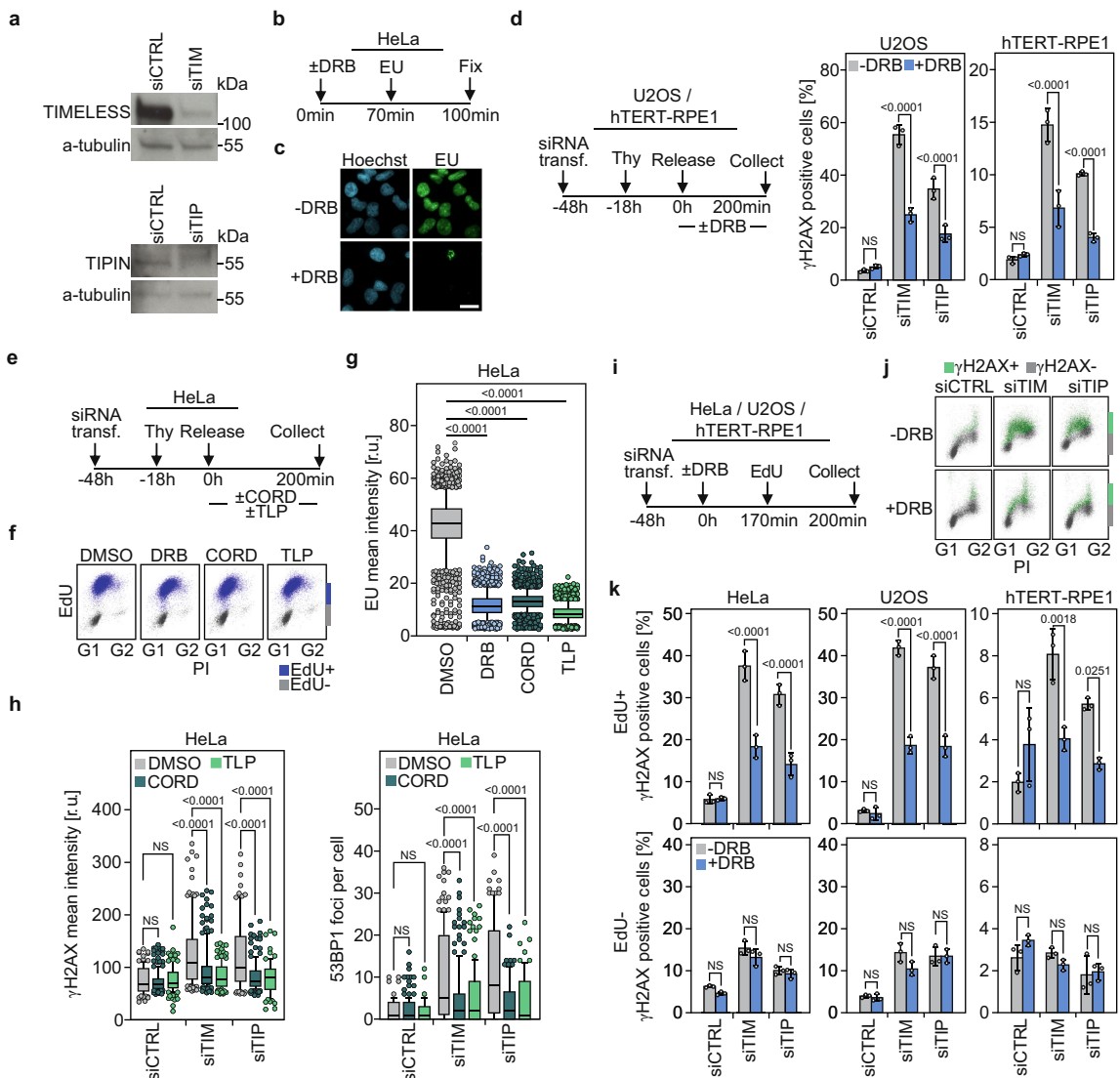

**Extended Data Fig. 1 | Depletion of TIMELESS or TIPIN by siRNAs induces a TRC-dependent DNA damage response in normal and cancer cells.**
**a**, Efficiency of siRNA-mediated depletion of TIMELESS and TIPIN in HeLa cells by immunoblotting. PCNA and α-tubulin served as loading controls.
**b-c**, Inhibition of transcription elongation by DRB. **b**, Outline of the experiment. **c**, Representative images of HeLa cells indicating inhibition of EU incorporation by DRB; the nuclei were counterstained with Hoechst 33342. Scale bar: 10 μm.
**d**, Induction of a DNA damage response in U2OS and hTERT-RPE1 cells transfected with control siRNA or siRNAs targeting *TIMELESS* or *TIPIN*. γH2AX levels were determined by flow cytometry; bars indicate means ± 1 s.d.; *n* = 3 replicates; ANOVA. **e-h**, Transcription inhibitors cordycepin (CORD) and triptolide (TLP) suppress the DNA damage response induced by depletion of TIMELESS or TIPIN. **e**, Outline of the experiment. **f**, Flow cytometry profiles for EdU incorporation and DNA content. **g**, Quantification of EU incorporation;

plots show medians and value ranges of 25-75% and 10-90%, filled circles indicate the individual cells in the top and bottom deciles; *n* = 2 replicates; >2624 cells per group (range: 2624-2899); ANOVA. **h**, Quantification of γH2AX mean intensity and number of 53BP1 foci per cell; plots show medians and value ranges of 25–75% and 10–90%, filled circles indicate the individual cells in the top and bottom deciles; *n* = 2 replicates; >66 cells per group (range: 66–194); ANOVA. **i-k**, Ongoing DNA replication is required for induction of a DNA damage response by depletion of TIMELESS or TIPIN. **i**, Outline of the experiment. **j**, γH2AX levels and DNA content ascertained by flow cytometry. **k**, Quantification of γH2AX positive cells determined separately for the EdU-positive and EdU-negative cells; bars indicate means ± 1 s.d.; *n* = 3 replicates; ANOVA. CTRL, control; TIM, TIMELESS; TIP, TIPIN; transf., transfection; Thy, thymidine; IF, immunofluorescence; PI, propidium iodide; r.u., relative units; NS, not significant.

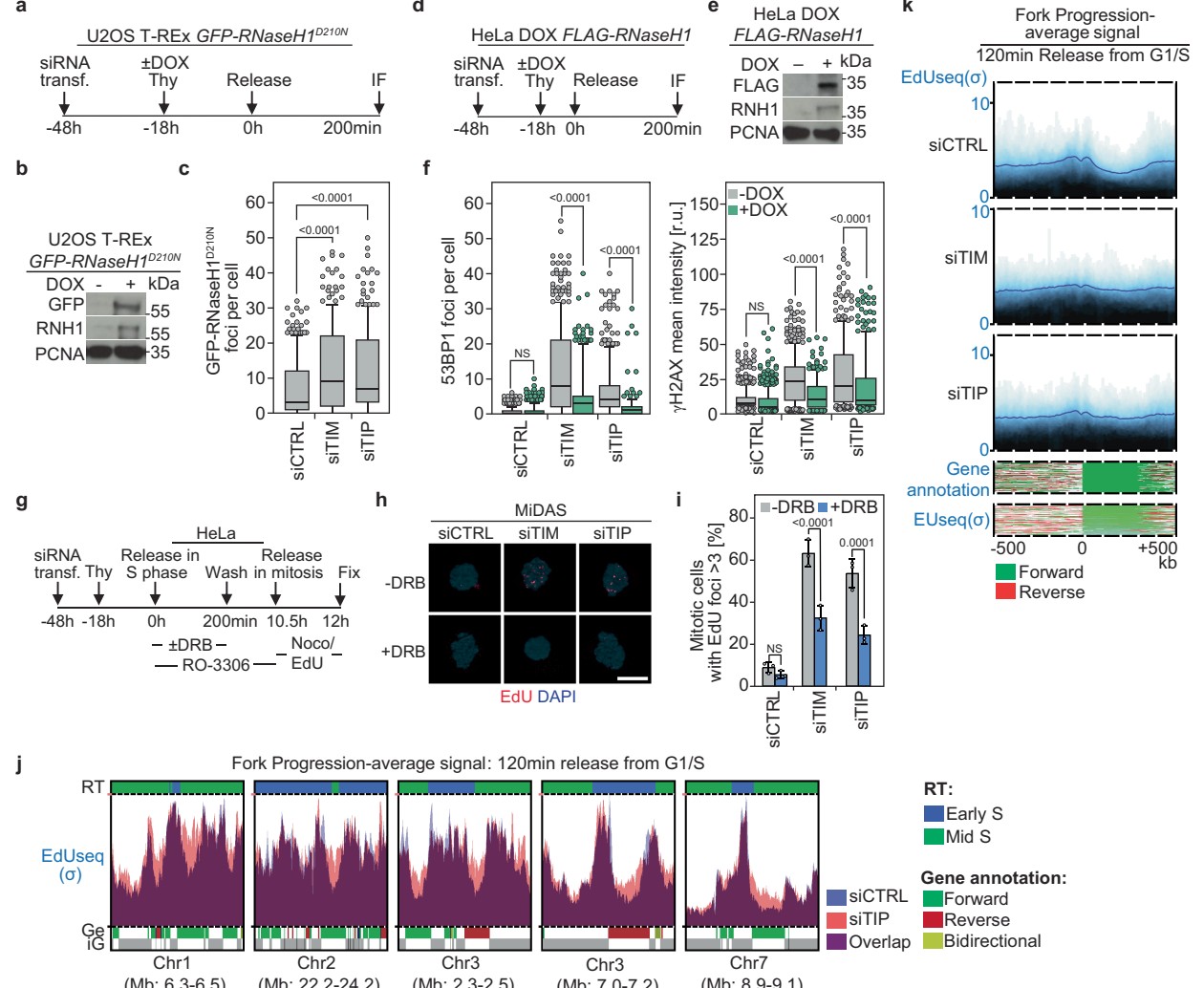

**Extended Data Fig. 2 | Induction of R-loops, MiDAS and increased replication fork speed following depletion of TIMELESS or TIPIN.** **a-c**, Induction of R-loops in cells after depletion of TIMELESS or TIPIN. **a**, Outline of the experiment. The U2OS cells used in this experiment expressed GFP-RNaseH1$^{D210N}$ in a doxycycline (DOX)-dependent manner. **b**, Induction of expression of GFP-RNaseH1$^{D210N}$ by DOX, as monitored by immunoblotting; PCNA served as loading control. **c**, Quantification of the number of GFP-RNaseH1$^{D210N}$ foci following depletion of TIMELESS or TIPIN; plots show medians and value ranges of 25–75% and 10–90%, filled circles indicate the individual cells in the top decile; $n$ = 2 replicates; >212 cells per group (range: 212–221); ANOVA. **d-f**, The DNA damage response induced by depletion of TIMELESS or TIPIN is suppressed by expression of RNase H1. **d**, Outline of the experiment. The HeLa cells used in this experiment expressed FLAG-RNaseH1 in a DOX-dependent manner. **e**, Induction of expression of FLAG-RNaseH1 by DOX, as monitored by immunoblotting. **f**, Quantification of the number of 53BP1 foci per cell and of γH2AX mean intensity; plots show medians and value ranges of 25–75% and 10–90%, filled circles indicate the individual cells in the top and bottom deciles; $n$ = 2 replicates; >199 cells per group (range: 199–557); ANOVA. **g-i**, Induction of MiDAS in prometaphase cells following depletion of TIMELESS or TIPIN. **g**, Outline of the experiment; the Cdk1 inhibitor RO–3306 inhibited entry into

mitosis; nocodazole (Noco) prevented exit from mitosis. **h**, Representative images of prometaphase cells with MiDAS; the DNA was counterstained with DAPI. Scale bar: 5 μm. **i**, Quantification of the percentage of prometaphase cells with >3 EdU foci; bars indicate means ±1 s.d.; $n$ = 3 replicates; >294 prometaphase cells per group (range: 294–315); ANOVA. **j-k**, Increased rates of fork progression over transcribed genes following depletion of TIMELESS or TIPIN in HeLa cells. The outline of the experiment is shown in Fig. 1g. The experiment shown here is a replicate of the experiment shown in Fig. 1h. **j**, EdUseq profiles at five representative genomic regions. Replication timing (RT): blue, early S phase; green, mid S phase. Genes (Ge): green, forward-transcribed genes; red, reverse-transcribed genes; yellow, overlap of forward and reverse-transcribed genes. Intergenic regions (iG): gray. Bin resolution: 10 kb; ruler scale: 100 kb. **k**, Average nascent DNA replication signal (EdUseq) at large (>300 kb) transcribed genes 120 min after release in S phase. The genes are aligned by their transcription start site and all genes are shown with their 5′-3′ orientation from left to right. Lower panels: heatmaps showing gene annotation and EUseq signal for each genomic locus used to generate the average EdUseq signal. Span of genomic regions: 1 Mb. transf., transfection; Thy, thymidine; IF, immunofluorescence; RNH1, RNase H1; r.u., relative units; CTRL, control; TIM, TIMELESS; TIP, TIPIN; σ, sigma value; NS, not significant.

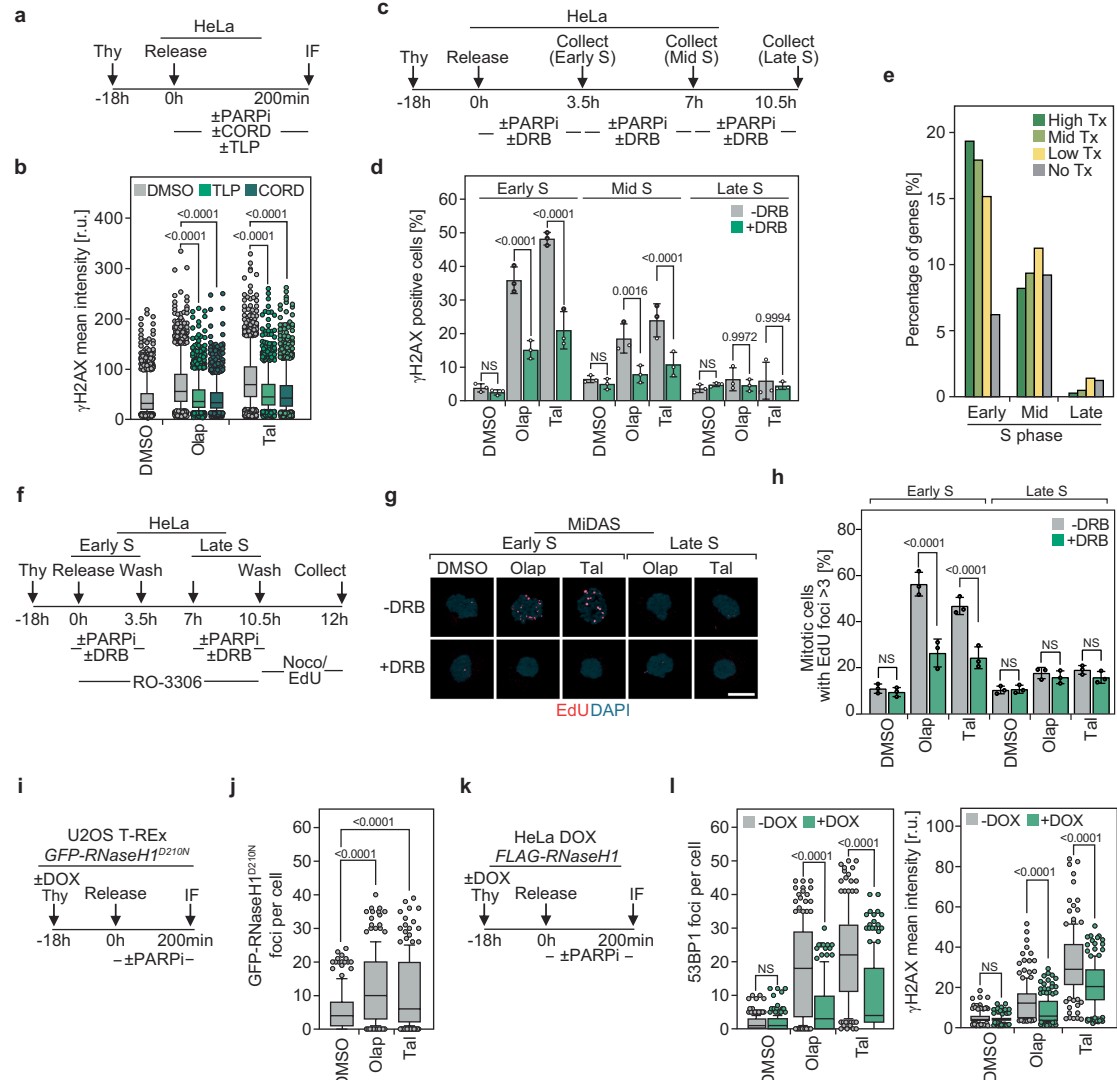

**Extended Data Fig. 3 | PARP inhibitors induce TRC-dependent DNA damage in early S phase and activate MiDAS. a-b**, Transcription inhibitors cordycepin (CORD) and triptolide (TLP) suppress the DNA damage response induced by PARP inhibitors. **a**, Outline of the experiment. **b**, Quantification of γH2AX mean intensity; plots show medians and value ranges of 25–75% and 10–90%, filled circles indicate the individual cells in the top and bottom deciles; $n = 2$ replicates; >1668 cells per group (range: 1668–2282); ANOVA. **c-d**, The induction of a DNA damage response by PARP inhibitors depends on whether cells are exposed to these inhibitors in early, mid or late S phase. **c**, Outline of the experiment. Cells were exposed to PARP inhibitors 0–3.5, 3.5–7 or 7–10.5 h after release from a thymidine block, corresponding to early, mid or late S phase, respectively. **d**, Quantification of the percentage of γH2AX positive cells by flow cytometry; bars indicate means ± 1 s.d.; $n = 3$ replicates; ANOVA. **e**, Distribution of human genes according to replication timing (early, mid or late S phase) and level of nascent transcription (High Tx, upper tertile of all expressed genes; Mid Tx, middle tertile; Low Tx, lower tertile; No Tx, non-expressed genes). Nascent transcription was determined by EUseq analysis of HeLa cells. **f-h**, Induction of MiDAS in prometaphase cells following treatment of cells with PARP inhibitors in early or late S phase (0–3.5 or 7–10.5 h after release from a thymidine block). **f**, Outline of the experiment; the Cdk1 inhibitor RO-3306 inhibited entry into

mitosis; nocodazole (Noco) prevented exit from mitosis. **g**, Representative images of prometaphase cells with MiDAS; the DNA was counterstained with DAPI. Scale bar: 5 μm. **h**, Quantification of the percentage of prometaphase cells with >3 EdU foci; bars indicate means ± 1 s.d.; $n = 3$ replicates; >127 prometaphase cells per group (range: 127–400); ANOVA. **i-j**, Induction of R-loops in cells treated with PARP inhibitors. **i**, Outline of the experiment. The U2OS cells used in this experiment express GFP-RNaseH1^{D210N} in a doxycycline (DOX)-dependent manner. **j**, Quantification of the number of GFP-RNaseH1^{D210N} foci following treatment with PARP inhibitors; plots show medians and value ranges of 25–75% and 10–90%, filled circles indicate the individual cells in the top and bottom deciles; $n = 2$ replicates; >192 cells per group; ANOVA. **k-l**, The DNA damage response induced by PARP inhibitors is suppressed by expression of RNase H1. **k**, Outline of the experiment. The HeLa cells used in this experiment express FLAG-RNaseH1 in a DOX-dependent manner. **l**, Quantification of the number of 53BP1 foci per cell and of γH2AX mean intensity; plots show medians and value ranges of 25–75% and 10–90%, filled circles indicate the individual cells in the top and bottom deciles; $n = 2$ replicates; >141 cells per group (range: 141–206); ANOVA. Thy, thymidine; PARPi, PARP inhibitor; r.u., relative units; Olap, olaparib (10 μM); Tal, talazoparib (100 nM); NS, not significant.

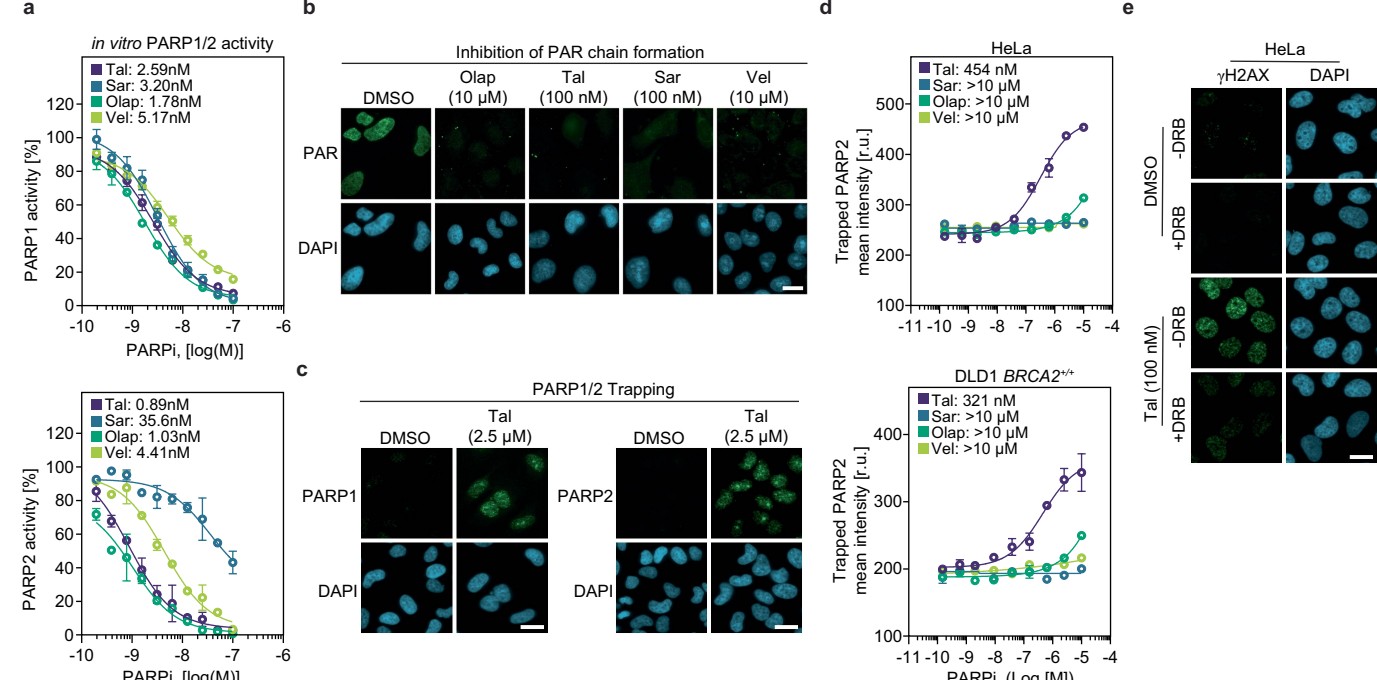

**Extended Data Fig. 4 | Characterization of PARP inhibition and PARP trapping activities of the four PARP inhibitors used in this study.**
**a**, Dose-response curves and calculated IC$_{50}$ values for inhibition of PARP1 and PARP2 enzymatic activities in vitro by the indicated PARP inhibitors. One of $n = 2$ replicates is presented. The IC$_{50}$ values determined by this assay might be inaccurate, due to the assay not being sensitive enough for the most potent inhibitors; these inhibitors might appear less potent than they actually are[20]. **b**, Examples of images of HeLa cells treated with PARP inhibitors and H$_2$O$_2$ that were used to assess inhibition of PARP enzymatic activity in cells. The cells were treated as shown in Fig. 3a and were immunostained for poly(ADP-ribose) (PAR) chains; the nuclei were counterstained with DAPI. Scale bar: 10 μm.

**c**, Representative images of HeLa cells treated with different PARP inhibitors, pre-extracted and immunostained for PARP1 or PARP2; the nuclei were counterstained with DAPI. Scale bar: 10 μm. **d**, Dose-response curves and calculated EC$_{50}$ values for PARP2 trapping in HeLa and DLD1 *BRCA2*$^{+/+}$ cells; means ± 1 s.d.; $n = 2$ replicates; for HeLa >2204 (range: 2204–10543), for DLD1 > 1605 (range: 1605–11936) cells per data point. **e**, Representative images of HeLa cells treated with talazoparib and optionally with DRB and immunostained for γH2AX; the nuclei were counterstained with DAPI. Scale bar: 10 μm. r.u., relative units; Tal, talazoparib; Sar, saruparib; Olap, olaparib; Vel, veliparib.

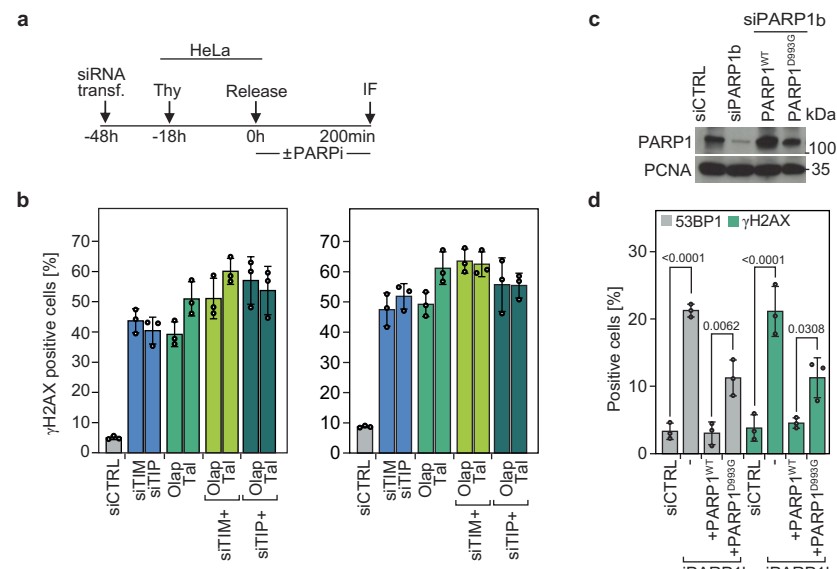

**Extended Data Fig. 5 | PARP inhibitors function in the same pathway as TIMELESS and TIPIN to prevent a TRC-induced DNA damage response.**
**a**, Outline of the experiment. **b**, Quantification of the percentage of cells with γH2AX and 53BP1 foci following depletion of TIMELESS or TIPIN or treatment with PARP inhibitors and/or combinations thereof; γH2AX and 53BP1-positive cells: >20 foci; bars indicate means ± 1 s.d.; $n$ = 3 replicates; >207 cells per group (range: 207–403). Olap, olaparib (10 μM); Tal, talazoparib (100 nM). CTRL, control; TIM, TIMELESS; TIP, TIPIN. **c-d**, Substitutions targeting the TIMELESS-PARP1 interface compromise the function of these proteins in averting TRC-dependent

DNA damage responses. **c**, Levels of endogenous and ectopically-expressed PARP1 proteins in cells transfected with siRNAs (CTRL or PARP1b) and plasmids (PARP1^WT or PARP1^D993G). A representative immunoblot is shown; PCNA served as loading control. siPARP1b, siRNA targeting the endogenous *PARP1* gene, but not the *PARP1* genes expressed by the plasmids; PARP1^WT, wild-type PARP1; PARP1^D993G, D993G single amino acid substitution mutant. **d**, Quantification of the percentage of cells with 53BP1 or γH2AX foci; bars indicate means ± 1 s.d.; $n$ = 3 replicates; for 53BP1 foci >246 (range: 246–657), for γH2AX foci >251 (range: 251–624) cells per group; ANOVA.

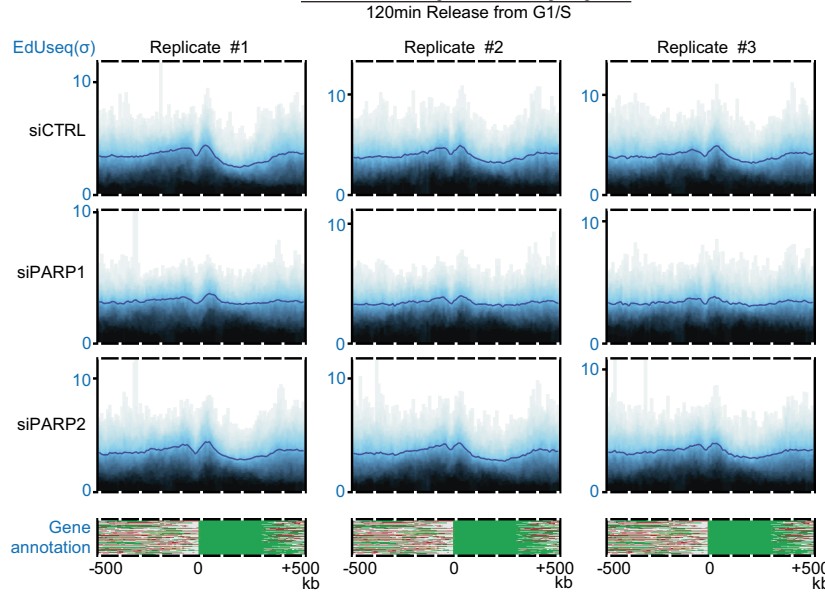

**Extended Data Fig. 6 | Increased replication fork progression rates within large, transcribed genes in HeLa cells depleted of PARP1.** The outline of the experiment, performed in triplicate, is shown in Fig. 4g. Here, the average nascent DNA replication signals over large (>300 kb) transcribed genes 120 min after release in S phase are shown separately for each replicate. The merged averages of all three replicates are shown in Fig. 4h. The genes are aligned by their transcription start site and all genes are shown with their 5′–3′ orientation from left to right. Lower panels: heatmaps showing gene annotation for each genomic locus used to generate the average EdUseq profiles. Span of genomic regions: 1 Mb; σ, sigma value; CTRL, control.

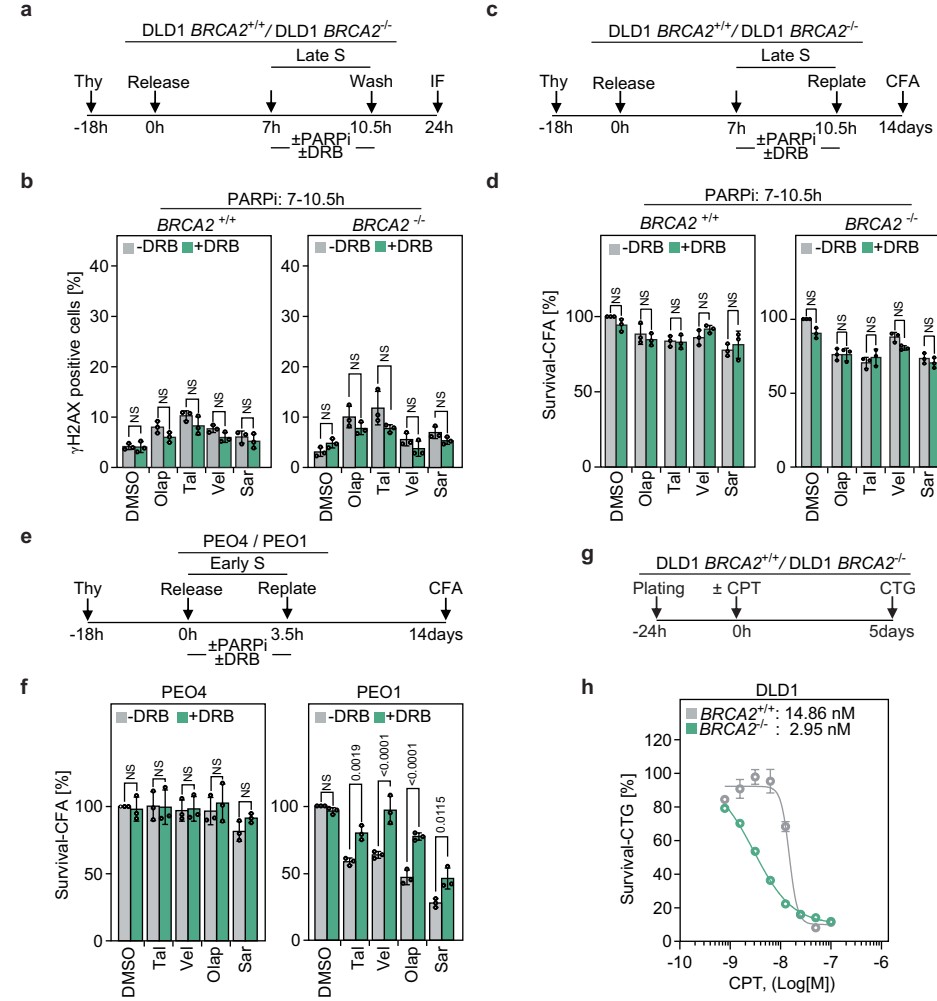

**Extended Data Fig. 7 | The sensitivity of HR-deficient cells to PARP inhibitors is affected by whether the cells are exposed to PARP inhibitors in early or late S phase. a-b**, Exposure of HR-deficient cells to PARP inhibitors in late S phase does not lead to induction of a DNA damage response. **a**, Outline of the experiment. **b**, Quantification of the percentage of cells with more than 20 γH2AX foci per cell. The bars indicate means ± 1 s.d.; $n$ = 3 replicates; >283 cells per group (range: 283–521); ANOVA. **c-d**, Exposure of cells to PARP inhibitors in late S phase does not lead to synthetic lethality with HR deficiency. **c**, Outline of the experiment. **d**, Quantification of cell survival by a colony formation assay (CFA) with the DMSO-treated cells serving as reference. The bars indicate means ± 1 s.d.; $n$ = 3 replicates; ANOVA. **e-f**, The synthetic lethality of HR-deficient

PEO1 cells treated with PARP inhibitors in early S phase is alleviated by inhibiting transcription elongation. **e**, Outline of the experiment. PEO4 cells are HR-proficient revertant cells derived from the same cancer as PEO1 cells. **f**, Quantification of cell survival by CFA; bars indicate means ± 1 s.d.; $n$ = 3 replicates; ANOVA. **g-h**, HR-deficient cells have increased sensitivity to camptothecin (CPT). **g**, Outline of the experiment. **h**, Dose-response survival curves and calculated $EC_{50}$ values for DLD1 *BRCA2*[+/+] and DLD1 *BRCA2*[-/-] cells following treatment with CPT. Olap, olaparib (10 μM); Tal, talazoparib (100 nM); Vel, veliparib (10 μM); Sar, saruparib (1 μM); CTG, CellTiter-Glo Cell Viability Assay; NS, not significant.

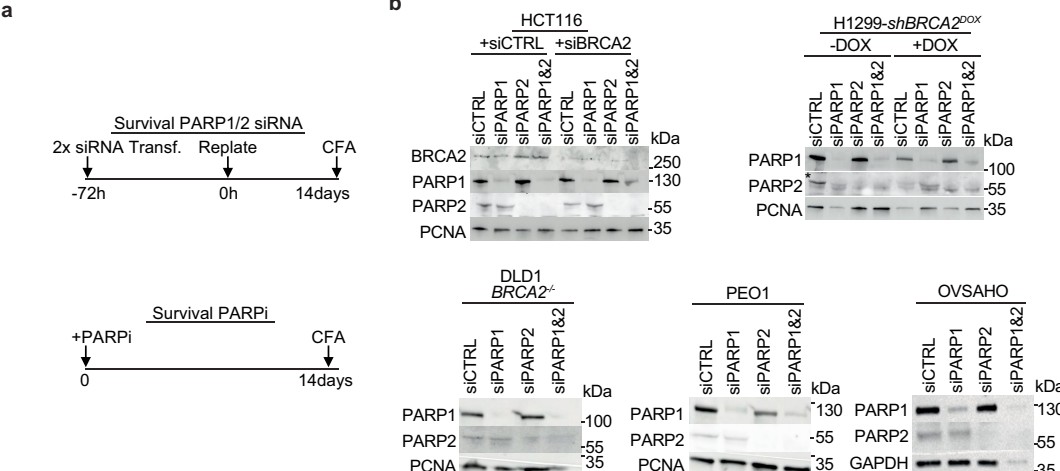

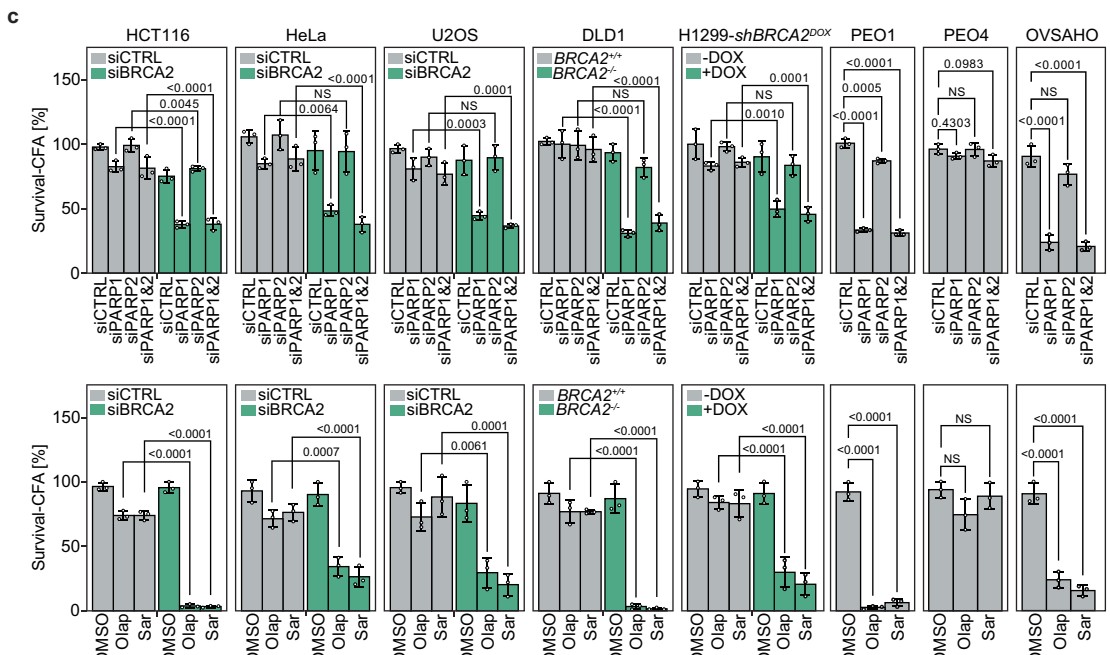

**Extended Data Fig. 8 | Depletion of PARP1 by siRNA is synthetic lethal with HR-deficiency. a**, Outline of the experiment. The cell lines were transfected with siRNA or exposed to PARP inhibitors. Viability was assessed by a colony formation assay (CFA). Note that in the siRNA-transfected cells, PARP1 and/or PARP2 were depleted for only a few days, whereas the PARP inhibitors were present over the entire 14 day-period. **b**, Assessment of the efficacy of depletion of BRCA2, PARP1 and PARP2 by immunoblotting. PCNA and GAPDH served as

loading controls. The H1299-shBRCA2^DOX cells induce expression of shRNA targeting *BRCA2* in a doxycycline (DOX)-dependent manner. **c**, Quantification of cell survival; bars indicate means ± 1 s.d.; *n* = 3 replicates; ANOVA. PARPi, PARP inhibitor; transf., transfection; CTRL, control; Olap, olaparib (1 μM, except for DLD1 and OVSAHO cells: 5 μM); Sar, saruparib (100 nM); NS, not significant.

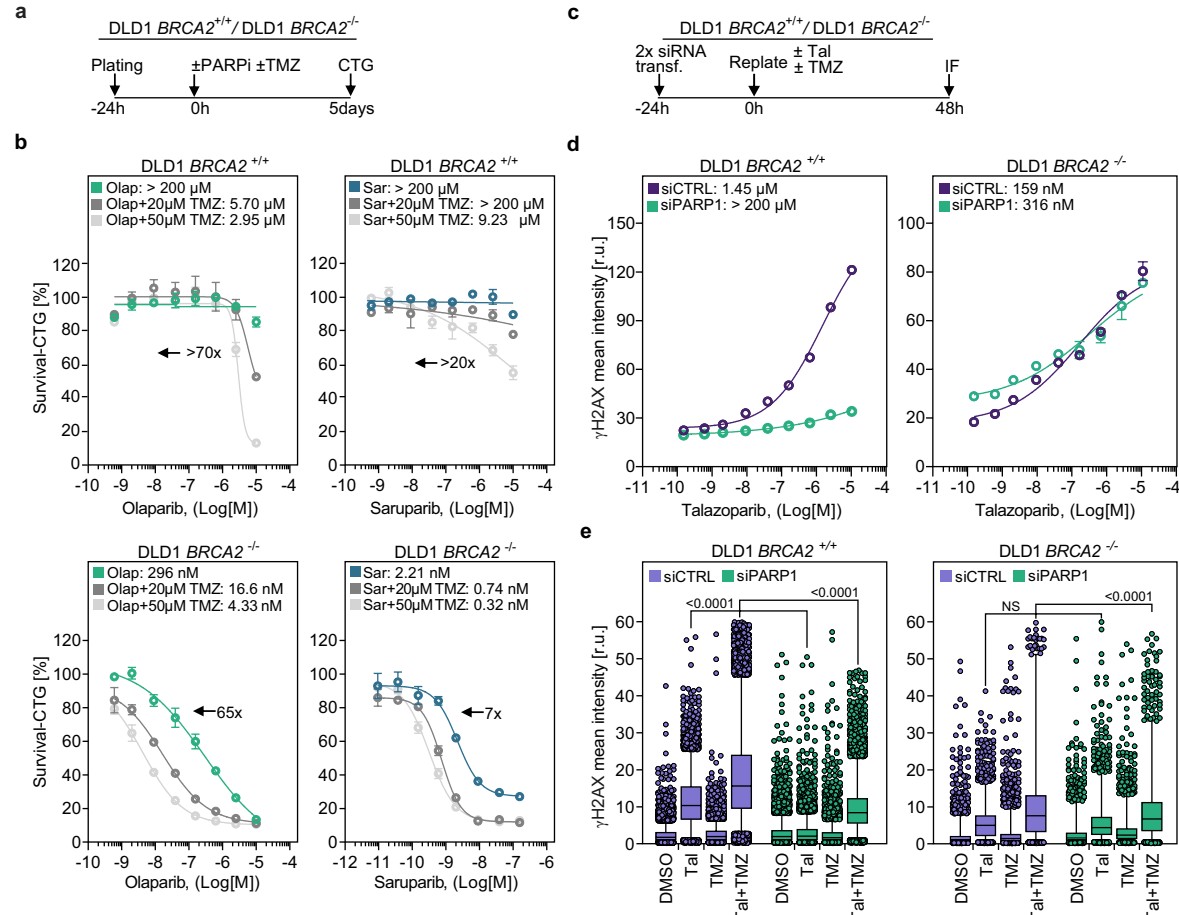

**Extended Data Fig. 9 | PARP trapping is dispensable for the synthetic lethality of PARP inhibitors with HR deficiency, yet toxic for HR-proficient cells. a-b**, Temozolomide (TMZ) reduces the selectivity of olaparib and saruparib for HR-deficient cells. **a**, Outline of the experiment. **b**, Dose-response survival curves and calculated $EC_{50}$ values of olaparib and saruparib-mediated lethality of DLD1 *BRCA2*$^{+/+}$ and DLD1 *BRCA2*$^{-/-}$ cells with and without added TMZ. Horizontal arrows indicate the fold-change in $EC_{50}$ values as a result of administering 50 μM TMZ. Data from one of *n* = 2 replicates. **c-e**, Depletion of PARP1 suppresses the induction of a DNA damage response in HR-proficient, but not in HR-deficient cells, treated with talazoparib. **c**, Outline of the experiment. γH2AX intensity was monitored 48 h after talazoparib was administered. **d**, Dose-response curves and calculated $EC_{50}$ values for induction of γH2AX. Data are from one of *n* = 2 replicates; for DLD1 *BRCA2*$^{+/+}$ > 7732 (range: 7732-21772), for DLD1 *BRCA2*$^{-/-}$ > 3368 (range: 3368–5803) cells per datapoint. **e**, Quantification of γH2AX intensity in DLD1 *BRCA2*$^{+/+}$ and DLD1 *BRCA2*$^{-/-}$ cells transfected with control siRNA or siRNA targeting *PARP1* and treated with talazoparib (1 μM) and optionally with TMZ (50 μM); plots show medians and value ranges of 25–75% and 10–90%, filled circles indicate the individual cells in the top and bottom deciles; *n* = 2 replicates; >17091 cells per group (range: 17091–31634); ANOVA. PARPi, PARP inhibitor; transf., transfection; CTRL, control; CTG, CellTiter-Glo Cell Viability Assay; NS, not significant.

# Reporting Summary

## Statistics

For all statistical analyses, confirm that the following items are present in the figure legend, table legend, main text, or Methods section.

| n/a | Confirmed | |
|---|---|---|
| ☐ | ☒ | The exact sample size (*n*) for each experimental group/condition, given as a discrete number and unit of measurement |
| ☐ | ☒ | A statement on whether measurements were taken from distinct samples or whether the same sample was measured repeatedly |
| ☐ | ☒ | The statistical test(s) used AND whether they are one- or two-sided *Only common tests should be described solely by name; describe more complex techniques in the Methods section.* |
| ☒ | ☐ | A description of all covariates tested |
| ☐ | ☒ | A description of any assumptions or corrections, such as tests of normality and adjustment for multiple comparisons |
| ☐ | ☒ | A full description of the statistical parameters including central tendency (e.g. means) or other basic estimates (e.g. regression coefficient) AND variation (e.g. standard deviation) or associated estimates of uncertainty (e.g. confidence intervals) |
| ☐ | ☒ | For null hypothesis testing, the test statistic (e.g. *F*, *t*, *r*) with confidence intervals, effect sizes, degrees of freedom and *P* value noted *Give P values as exact values whenever suitable.* |
| ☒ | ☐ | For Bayesian analysis, information on the choice of priors and Markov chain Monte Carlo settings |
| ☒ | ☐ | For hierarchical and complex designs, identification of the appropriate level for tests and full reporting of outcomes |
| ☒ | ☐ | Estimates of effect sizes (e.g. Cohen's *d*, Pearson's *r*), indicating how they were calculated |

*Our web collection on statistics for biologists contains articles on many of the points above.*

## Software and code

Policy information about availability of computer code

| Data collection | Genomic DNA was sonicated using a Bioruptor sonicator (Diagenode). Library preparation was performed using the TruSeq ChIP Sample Prep Kit (Illumina, Cat. No. IP-202-1012). High-throughput 100-base-pair single-end sequencing was performed on an Illumina Hi-Seq 4000 sequencer. Microscopy images were acquired using a Zeiss Imager M2 AX10 with the ZEN3.4 (blue edition) software or an ImageXpress spinning disc confocal microscope (Molecular devices) with Metaepxress software. Luminescence for viability experiments was measured using a Spark 10 M microplate reader (Tecan). |
|---|---|
| Data analysis | Sequencing reads were aligned on the non-masked human genome assembly (GRCh37/hg19) using the Burrows-Wheeler Aligner software as described previously (Macheret & Halazonetis, Nature 2018; Macheret et al., Cell Research, 2020). Previously described custom Perl scripts were used to assign the aligned reads to 10 kb genomic bins. Sigma (σ) values were calculated as the normalized number of reads per bin divided by its standard deviation. The data were visualized using previously described scripts (Macheret & Halazonetis, Nature 2018). Assignment of replication timing was performed with REPLI-seq data generated previously (Macheret & Halazonetis, Nature 2018). ImajeJ version 1.8.0 and the MetaXpress Custom Module Editor was used for image analysis. Kaluza v2.1 was used for flow cytometry analysis. GraphPad Prism v9.4.1 was used for statistical analysis and graphing. Figures were assembled with Adobe Illustrator CS6. |

For manuscripts utilizing custom algorithms or software that are central to the research but not yet described in published literature, software must be made available to editors and reviewers. We strongly encourage code deposition in a community repository (e.g. GitHub). See the Nature Portfolio guidelines for submitting code & software for further information.

## Data

Policy information about availability of data

All manuscripts must include a data availability statement. This statement should provide the following information, where applicable:
- Accession codes, unique identifiers, or web links for publicly available datasets
- A description of any restrictions on data availability
- For clinical datasets or third party data, please ensure that the statement adheres to our policy

The fastq sequencing data and associated information described in this study have been deposited in the Sequence Read Archive (SRA) with GEO Accession Number GSE220223. The EUseq data used in this study were previously published 53. Unprocessed images of western blots and the gating strategy for the flow cytometry experiments are provided as Supplementary Information. All information supporting the conclusions are provided with the paper.

## Human research participants

Policy information about studies involving human research participants and Sex and Gender in Research.

| | |
|---|---|
| Reporting on sex and gender | N/A |
| Population characteristics | N/A |
| Recruitment | N/A |
| Ethics oversight | N/A |

Note that full information on the approval of the study protocol must also be provided in the manuscript.

# Field-specific reporting

Please select the one below that is the best fit for your research. If you are not sure, read the appropriate sections before making your selection.

☒ Life sciences          ☐ Behavioural & social sciences          ☐ Ecological, evolutionary & environmental sciences

For a reference copy of the document with all sections, see nature.com/documents/nr-reporting-summary-flat.pdf

# Life sciences study design

All studies must disclose on these points even when the disclosure is negative.

| | |
|---|---|
| Sample size | No statistical methods were used to determine the sample size. All experiments were performed in triplicate (independent biological triplicates) with few exceptions of experiments that were performed in duplicates (this is mentioned in the figure legends). The specific number of cells analysed for each experiment is reported in the figure legends for main and Exteded Data Figures. The experiments were performed in several different cell lines (three independent biological replicates per cell line) to determine consistency of the results across cell lines. |
| Data exclusions | No data were excluded. |
| Replication | For most of the experiments, at least three independent biological experiments were performed. For each experiment, detailed description of number of replicates, sample size and statistics is provided in figure legend. Similar parameters were evaluated by multiple methods. For example, we counted gH2AX foci by automated microscopy; and monitored total nuclear gH2AX levels by microscopy and flow cytometry, having similar results with all methods. |
| Randomization | Cell lines were split into different plates/wells and all control and experimental treatments were randomly assigned to the plates/wells. |
| Blinding | The investigator was not blinded. However, the counting of the variables was in most cases automated (eg. counting of foci, counting of immunofluorescence and flow cytometry signal intensity), so it is not possible to introduce bias by the investigator. |

# Reporting for specific materials, systems and methods

We require information from authors about some types of materials, experimental systems and methods used in many studies. Here, indicate whether each material, system or method listed is relevant to your study. If you are not sure if a list item applies to your research, read the appropriate section before selecting a response.

## Materials & experimental systems

| n/a | Involved in the study |
|---|---|
| ☐ | ☒ Antibodies |
| ☐ | ☒ Eukaryotic cell lines |
| ☒ | ☐ Palaeontology and archaeology |
| ☒ | ☐ Animals and other organisms |
| ☒ | ☐ Clinical data |
| ☒ | ☐ Dual use research of concern |

## Methods

| n/a | Involved in the study |
|---|---|
| ☐ | ☒ ChIP-seq |
| ☐ | ☒ Flow cytometry |
| ☒ | ☐ MRI-based neuroimaging |

# Antibodies

| Antibodies used | Primary antibodies [Immunofluorescence]: |
|---|---|
| | γH2AX (S139) mouse monoclonal (1:1000, clone JBW301, Millipore, Cat. No. 05-636) |

Primary antibodies [Immunofluorescence]:
γH2AX (S139) mouse monoclonal (1:1000, clone JBW301, Millipore, Cat. No. 05-636)
RAD51 rabbit polyclonal (1:1000, Bioacademia, Cat. No. 70-002)
53BP1 rabbit polyclonal (1:1000, Novus Biologicals, Cat. No. NB100-304)
poly (ADP-ribose) mouse monoclonal (1:500, Clone 10HA, Trevigen, Cat. No. 4335-MC-100 & 1-500, Enzo Life Sciences, Cat. No. ALX-804-220-R100)
PARP1 rabbit polyclonal (1:1000, ProteinTech, Cat. No. 13371–1-AP)
PARP2 rabbit polyclonal (1:1000, Active Motif, Cat. No. 39743)

Primary antibodies [Western Blot]:
PCNA mouse monoclonal (1:1000, clone PC10, Millipore, Cat. No. MABE288)
alpha-Tubulin mouse monoclonal (1:1000, clone DM1A, Calbiochem, Cat. No. CP06)
GAPDH mouse monoclonal (1:10000, clone 6C5, Abcam, Cat. No. ab8245)
TIMELESS rabbit polyclonal (1:1000, Abcam, Cat. No. ab109512)
TIPIN rabbit polyclonal (1:250, Bethyl Laboratories Cat. No. A301-474A)
PARP1 rabbit polyclonal (1:1000, Abcam, Cat. No. ab32138)
PARP2 rabbit polyclonal (1:500, Active Motif, Cat. No 39743)
BRCA2  mouse monoclonal (1:1000, clone 2B, Calbiochem, Cat. No. OP95)
Actinin mouse monoclonal (1:1000, clone AT6/172, Sigma-Aldrich, Cat. No. 05-384)
RNase H1 rabbit polyclonal (1:500, ProteinTech, Cat. No. 15606-1-AP)
FLAG mouse monoclonal (1:1000, clone M2, Sigma Aldrich, Cat. No. F1804)
GFP rabbit polyclonal (1:500, Abcam, Cat. No. ab290).

Secondary Antibodies [Immunofluorescence]:
Alexa Fluor 488 Goat-Anti Rabbit IgG (1:500, Invitrogen, Cat. No., A110334)
Alexa Fluor 488 Goat-Anti Mouse IgG (1:500, Invitrogen, Cat. No. A11001)
Alexa Fluor 594 Goat-Anti Rabbit IgG (1:500, Invitrogen, Cat. No. A11037)
Alexa Fluor 594 Goat-Anti Mouse IgG (1:500, Invitrogen, Cat. No, A11005)
Alexa Fluor 647 Goat-Anti Rabbit IgG (1:500, Invitrogen, Cat. No. A21244)
Alexa Fluor 647 Goat-Anti Mouse IgG (1:500, Invitrogen, Cat. No. A21235)

Secondary Antibodies [Western Blot]:
Anti-Mouse HRP IgG (1:2500, Promega, Cat. No. W401B)
Anti-Rabbit HRP IgG (1:2500, Promega, Cat. No. W402B)

Validation

For antibodies used to monitor DNA damage, we examined cells treated with or without DNA damaging agents. For antibodies used to monitor protein levels by western blot, we validated loss of the protein band in cells transfected with the appropriate siRNA. Specificities of the antibodies were validated by the manufacturer and are listed below:

γH2AX (S139) mouse (Millipore, Cat. No. 05-636): https://www.merckmillipore.com/CH/de/product/Anti-phospho-Histone-H2A.X-Ser139-Antibody-clone-JBW301,MM_NF-05-636
RAD51 rabbit (Bioacademia, Cat. No. 70-002): https://www.bioacademia.co.jp/en/products/list?
53BP1 rabbit (Novus Biologicals, Cat. No. NB100-304): https://www.novusbio.com/products/53bp1-antibody_nb100-304
poly (ADP-ribose) mouse (Trevigen, Cat. No. 4335-MC-100): https://www.rndsystems.com/products/par-padpr-antibody-10ha_4335-mc-100#product-citations
poly (ADP-ribose) mouse (Enzo Life Sciences, Cat. No. ALX-804-220-R100): https://www.enzolifesciences.com/ALX-804-220/poly-adp-ribose-monoclonal-antibody-10h/
PARP1 rabbit  (ProteinTech, Cat. No. 13371–1-AP): https://www.ptglab.com/products/PARP1-Antibody-13371-1-AP.htm
PARP2 rabbit  (Active Motif, Cat. No. 39743): https://www.activemotif.com/catalog/details/39743/parp-2-antibody-pab
PCNA mouse (Millipore, Cat. No. MABE288): https://www.merckmillipore.com/CH/de/product/Anti-PCNA-Antibody-clone-PC10,MM_NF-MABE288
alpha-Tubulin mouse (Calbiochem, Cat. No. CP06): https://www.merckmillipore.com/CH/de/product/Anti-Tubulin-Mouse-mAb-DM1A,EMD_BIO-CP06
GAPDH mouse (Abcam, Cat. No. ab8245): https://www.abcam.com/products/primary-antibodies/gapdh-antibody-6c5-loading-control-ab8245.html
TIMELESS rabbit (Abcam, Cat. No. ab109512): https://www.abcam.com/products/primary-antibodies/timeless-antibody-epr5275-ab109512.html
TIPIN rabbit (Bethyl Laboratories Cat. No. A301-474A): https://www.thermofisher.com/antibody/product/TIPIN-Antibody-Polyclonal/A301-474A

PARP1 rabbit (Abcam, Cat. No. ab32138): https://www.abcam.com/products/primary-antibodies/parp1-antibody-e102-ab32138.html
BRCA2 mouse (Calbiochem, Cat. No. OP95): https://www.merckmillipore.com/CH/de/product/Anti-BRCA2-Ab-1-Mouse-mAb-2B,EMD_BIO-OP95
Actinin mouse (Sigma-Aldrich, Cat. No. 05-384):https://www.merckmillipore.com/CH/de/product/Anti-Actinin-Antibody-clone-AT6-172,MM_NF-05-384
RNase H1 rabbit (ProteinTech, Cat. No. 15606-1-AP): https://www.ptglab.com/products/RNASEH1-Antibody-15606-1-AP.htm
FLAG mouse (Sigma Aldrich, Cat. No. M2 F1804): https://www.sigmaaldrich.com/CH/de/product/sigma/f1804
GFP rabbit (1:500, Abcam, Cat. No. ab290): https://www.abcam.com/en-at/products/primary-antibodies/gfp-antibody-ab290

# Eukaryotic cell lines

Policy information about cell lines and Sex and Gender in Research

| Cell line source(s) | HeLa (ATCC, Cat. No.CCL-2)<br>U2OS (ATCC, Cat. No. HTB-96)<br>hTERT-RPE1 (ATCC, Cat. No. CRL4000)<br>DLD1 (ATCC, Cat. No. CCL-221)<br>DLD1 BRCA2 KO (Horizon, Cat. No. HD 105-007)<br>PEO1 and PEO4 from Prof. Labidi-Galy (Hospital of the University of Geneva); PEO1 (Sigma-Adrich, Cat. No. 10032308), PEO4 (Sigma-Aldrich, Cat. No. 10032309)<br>OVSAHO (Sigma-Aldrich, Cat. No. SCC294)<br>HCT116 (ATCC, Cat. No., CCL-247)<br>HeLa+RNaseH1-FLAG-DOX from Prof Tarsounas (doi.org/10.1016/j.molcel.2015.12.004)<br>U2OS T-REx GFP-RNaseH1(D210N)-DOX from Prof Janscak (doi.org/10.1016/j.molcel.2018.11.036)<br>H1299-shBRCA2-DOX from Prof Tarsounas (https://doi.org/10.1016/j.molcel.2015.12.004) |
|---|---|
| Authentication | Cell line identity verified by karyotyping. |
| Mycoplasma contamination | All cell lines regularly tested and found to be negative. |
| Commonly misidentified lines<br>(See ICLAC register) | None. |

# ChIP-seq

## Data deposition

☒ Confirm that both raw and final processed data have been deposited in a public database such as GEO.

☒ Confirm that you have deposited or provided access to graph files (e.g. BED files) for the called peaks.

| Data access links<br>*May remain private before publication.* | GEO Accession Number GSE220223 |
|---|---|
| Files in database submission | EdUseq_Exp5_HeLa_ThymRel_090_siCTRL.fastq.gz, EdUseq_Exp5_HeLa_ThymRel_090_siTIME.fastq.gz, EdUseq_Exp5_HeLa_ThymRel_090_siTIPI.fastq.gz, EdUseq_Exp5_HeLa_ThymRel_120_siCTRL.fastq.gz, EdUseq_Exp5_HeLa_ThymRel_120_siTIME.fastq.gz ,EdUseq_Exp5_HeLa_ThymRel_120_siTIPI.fastq.gz, EdUseq_Exp6_HeLa_ThymRel_120_siCTRL.fastq.gz, EdUseq_Exp6_HeLa_ThymRel_120_siPARP1.fastq.gz, EdUseq_Exp6_HeLa_ThymRel_120_siPARP2.fastq.gz, EdUseq_Exp7_HeLa_ThymRel_120_siCTRL_1.fastq.gz, EdUseq_Exp7_HeLa_ThymRel_120_siCTRL_2.fastq.gz, EdUseq_Exp7_HeLa_ThymRel_120_siPARP1_1.fastq.gz, EdUseq_Exp7_HeLa_ThymRel_120_siPARP1_2.fastq.gz, EdUseq_Exp7_HeLa_ThymRel_120_siPARP2_1.fastq.gz, EdUseq_Exp7_HeLa_ThymRel_120_siPARP2_2.fastq.gz, HeLa_siCTRL_R120_ThyRel_1_nm, HeLa_siCTRL_R120_ThyRel_2_nm, HeLa_siTIME_R120_ThyRel_2_nm, HeLa_siTIPI_R120_ThyRel_2_nm<br>Processed data of above files have also been submitted as csv and bigwig files |
| Genome browser session<br>(e.g. UCSC) | https://www.ncbi.nlm.nih.gov/geo/query/acc.cgi?acc=GSE220223<br>https://genome.ucsc.edu/ |

## Methodology

| Replicates | Control Samples: 7 replicates<br>TIMELESS depleted samples: 3 replicates<br>TIPIN depleted samples: 3 replicates<br>PARP1 depleted samples: 3 replicates<br>PARP2 depleted samples: 3 replicates |
|---|---|
| Sequencing depth | #Total reads<br>EdUseq_Exp5_HeLa_ThymRel_090_siCTRL.fastq.gz: 48200032<br>EdUseq_Exp5_HeLa_ThymRel_090_siTIME.fastq.gz: 33217880<br>EdUseq_Exp5_HeLa_ThymRel_090_siTIPI.fastq.gz: 40521243<br>EdUseq_Exp5_HeLa_ThymRel_120_siCTRL.fastq.gz: 41511519<br>EdUseq_Exp5_HeLa_ThymRel_120_siTIME.fastq.gz: 36394129<br>EdUseq_Exp5_HeLa_ThymRel_120_siTIPI.fastq.gz: 38526266<br>EdUseq_Exp6_HeLa_ThymRel_120_siCTRL.fastq.gz: 24739139 |

EdUseq_Exp6_HeLa_ThymRel_120_siPARP1.fastq.gz: 20417997
EdUseq_Exp6_HeLa_ThymRel_120_siPARP2.fastq.gz: 21675687
EdUseq_Exp7_HeLa_ThymRel_120_siCTRL_1.fastq.gz: 13091412
EdUseq_Exp7_HeLa_ThymRel_120_siCTRL_2.fastq.gz: 14785436
EdUseq_Exp7_HeLa_ThymRel_120_siPARP1_1.fastq.gz: 14918795
EdUseq_Exp7_HeLa_ThymRel_120_siPARP1_2.fastq.gz: 13095859
EdUseq_Exp7_HeLa_ThymRel_120_siPARP2_1.fastq.gz:14695337
EdUseq_Exp7_HeLa_ThymRel_120_siPARP2_2.fastq.gz: 11962343
HeLa_siCTRL_R120_ThyRel_1_nm: 27940030
HeLa_siCTRL_R120_ThyRel_2_nm: 25907850
HeLa_siTIME_R120_ThyRel_2_nm: 24287805
HeLa_siTIPI_R120_ThyRel_2_nm: 22040707

#Uniquely mapped reads
EdUseq_Exp5_HeLa_ThymRel_090_siCTRL.fastq.gz: 34067038
EdUseq_Exp5_HeLa_ThymRel_090_siTIME.fastq.gz: 25563815
EdUseq_Exp5_HeLa_ThymRel_090_siTIPI.fastq.gz: 30140809
EdUseq_Exp5_HeLa_ThymRel_120_siCTRL.fastq.gz: 31173814
EdUseq_Exp5_HeLa_ThymRel_120_siTIME.fastq.gz: 27136579
EdUseq_Exp5_HeLa_ThymRel_120_siTIPI.fastq.gz: 28916857
EdUseq_Exp6_HeLa_ThymRel_120_siCTRL.fastq.gz: 13159181
EdUseq_Exp6_HeLa_ThymRel_120_siPARP1.fastq.gz: 12501152
EdUseq_Exp6_HeLa_ThymRel_120_siPARP2.fastq.gz: 11817676
EdUseq_Exp7_HeLa_ThymRel_120_siCTRL_1.fastq.gz: 7058178
EdUseq_Exp7_HeLa_ThymRel_120_siCTRL_2.fastq.gz: 8226376
EdUseq_Exp7_HeLa_ThymRel_120_siPARP1_1.fastq.gz: 8116157
EdUseq_Exp7_HeLa_ThymRel_120_siPARP1_2.fastq.gz: 7276690
EdUseq_Exp7_HeLa_ThymRel_120_siPARP2_1.fastq.gz: 7968444
EdUseq_Exp7_HeLa_ThymRel_120_siPARP2_2.fastq.gz: 6690834
HeLa_siCTRL_R120_ThyRel_1_nm: 21631573
HeLa_siCTRL_R120_ThyRel_2_nm: 21266081
HeLa_siTIME_R120_ThyRel_2_nm: 19973301
HeLa_siTIPI_R120_ThyRel_2_nm: 18200703

Read length: 100 bp
Single-end reads.

**Antibodies**
We did not use antibodies. Nascent DNA was labeled with EdU and the EdU-labeled DNA was then linked to biotin using Click-iT Chemistry.

**Peak calling parameters**
We did not call peaks. We used gene annotation data (refseq from NCBI) to align the transcription start sites of large genes and then monitor their replication in early S phase.

**Data quality**
We did not call peaks.

**Software**
The data were visualized using custom scripts that have been submitted in the supplementary data section of Macheret and Halazonetis, Nature 2018

# Flow Cytometry

## Plots

Confirm that:

☒ The axis labels state the marker and fluorochrome used (e.g. CD4-FITC).

☒ The axis scales are clearly visible. Include numbers along axes only for bottom left plot of group (a 'group' is an analysis of identical markers).

☒ All plots are contour plots with outliers or pseudocolor plots.

☒ A numerical value for number of cells or percentage (with statistics) is provided.

## Methodology

**Sample preparation**
Cells were harvested by trypsinization and fixed in 90% methanol overnight at -20° C. EdU detection was performed using the Click-it EdU Alexa Fluor 647 Flow Cytometry Assay Kit (Invitrogen Cat. No. C-10424) according to the manufacturer's instructions. Detection of γH2AX phosphorylation was performed using the Guava Histone H2AX Phosphorylation Assay Kit (Luminex, FCCS100182) according to the manufacturer's instructions. The genomic DNA was stained by incubating the cells in PBS containing RNase (Roche, Cat. No. 11119915001) and propidium iodide (PI) (Sigma-Aldrich Cat. No. 81845).

**Instrument**
Gallios, Model 2L/8C, Beckman Coulter

**Software**
Kaluza, version 2.1, Beckman Coulter

| Cell population abundance | At least 20,000 cells were evaluated per sample. |
| --- | --- |
| Gating strategy | Cells were gated by FSC/SSC, then by PI peak area height/PI peak height to eliminate clumped cells, then by EdU and by gH2AX signals. An example showing one sample is attached. This example is provided as Supplementary Figure 2. |

☒ Tick this box to confirm that a figure exemplifying the gating strategy is provided in the Supplementary Information.

