## [Peer Review File · Nature]

Manuscript Title: Transcription-replication conflicts underlie sensitivity to PARP inhibitors

Reviewer Comments & Author Rebuttals

Reviewer Reports on the Initial Version:

Referees' comments:

Referee #1:

Petropoulos et al. examined the connection between transcription-replication conflicts (TRC), depletion of S-phase checkpoint proteins TIMELESS or TIPIN, and PARP1/2 depletion or inhibition in HRD-deficient (HRD) cells. They found that depletion of TIMELESS and TIPIN causes faster fork progression and DNA damage in S-phase cells in a transcription-dependent fashion and that this damage can be repaired in mitosis by MiDAS (mitotic DNA synthesis). PARP inhibitors phenocopy TIMELESS or TIPIN depletion and induce highest levels of DNA damage when applied in early S-phase when the number of expressed genes is the highest. Combined PARP1/2 siRNA-mediated depletion, PARPi and TIMELESS or TIPIN siRNA-mediated depletion have the same effect on transcription-dependent DNA damage in S-phase, suggesting that they are epistatic. TIMELESS was previously shown to interact with PARP1 independent of the PARP1 catalytic activity and TIMELESS PARP1-binding mutant cannot rescue the effects of TIMELESS depletion on TRC-dependent DNA damage. Depletion of PARP1 and PARP2 or TIMELESS/TIPIN is synthetic lethal with HRD caused by BRCA mutations. Inhibition of transcription elongation abrogates PARPi-induced synthetic lethality with BRCA deficiency, but only if PARPi is administered already during early S-phase. Collectively, these results show that TRCs caused by PARP1/2 depletion, PARPi or TIMELESS/TIPIN depletion result in DNA damage that cannot be repaired in HRD cells, and that TRCs contribute to synthetic lethality between PARPi and HRD. Given that this study focuses on TRCs as a mechanism of PARP1- and PARPi-induced synthetic lethality with HRD, a few key points need to be addressed with regard to the measurement of TRCs, PARP1 connection with TRCs, and synthetic lethality induced by PARP1 depletion vs PARP1 inhibition.

1) Synthetic lethality between PARPi and HRD was previously shown to be caused by PARP trapping and not PARP depletion (PMID: 23118055). In the Discussion the authors provide possible explanations for these conflicting results. However, this needs to be properly tested using multiple cell lines WT vs BRCA-mutated from different types of cancer (e.g., breast, ovarian, pancreatic) to show whether indeed PARP1 depletion shows synthetic lethality with BRCA mutations in cancer cells.

2) If PARP-related DNA damage phenotypes are due to TRCs, then the authors should test whether resolution of TRCs with RNase H overexpression can rescue the phenotypes.

3) If TRCs are responsible for PARPi + HRD synthetic lethality, then induction of TRCs through

camptothecin (Topo I inhibitor) should also show synthetic lethality with HRD.

4) DRB induces Pol II pausing, which was shown to induce DNA-RNA hybrid formation at transcription start sites where Pol II is paused (PMID: 29104020). In DRB-treated cells the levels of DNA-RNA hybrids and therefore TRCs, which can result in DNA damage, is expected to be higher. Therefore, it is essential to use another method, in addition to the proximity ligation assay, to examine DNA-RNA hybrids, such as RNase H hybrid-binding domain (HBD) or catalytic mutant.

5) In relation to point 4, transcription can be inhibited with alpha-amanitin, which does not affect Pol II pausing and DNA-RNA hybrid formation.

6) If PARP1/2 'signal the presence of TRCs to TIMELESS and TIPIN' then PARP1/2 are expected to somehow recognize TRCs, i.e., DNA-RNA hybrids, which can be tested in vitro.

7) BRCA1 and BRCA2 depletion was shown to increase DNA-RNA hybrid levels (PMID: 25699710, PMID: 24896180). What are the levels of DNA-RNA hybrids in BRCA1/2-deficient cells +/- PARP or TIMELESS siRNA?

8) The presentation of EdU-seq results does not allow proper conclusions. In Fig. 1h the gene scale is 50 kb. Considering a fork speed of 2 kb/min, within 120 min of release from the thymidine block 60 kb genes would be already replicated. EdU incorporation from 90-120 min would mean that in WT cells EdU signal would be concentrated at the gene ends (after 40 kb) so the observed dip makes sense. However, in siTIMELESS and siTIPIN cells the EdU signal is evenly distributed throughout the transcribed gene, which can be interpreted as:

i) replication fork progression is slower – not faster as the authors interpret, OR

ii) replication forks are faster and move into downstream genes – as the authors interpret.

To distinguish between the two scenarios, the authors should show a longer scale (e.g., 100 kb).

Referee #2:

The MS "Transcription-replication conflicts sensitize BRCA2-deficient cells to PARP inhibitors" addresses the highly topical and incompletely understood issue of the clinically-relevant synthetic lethality afforded by PARP inhibitors in homologous recombination (HR)-deficient cells. On the one hand, the MS convincingly relates transcription-dependent S-phase DNA damage to PARP inhibition as well as PARP1/2 loss, identifying a potential source of synthetic lethality in transcription-replication conflicts (TRCs). In a slightly more preliminary line of investigation, evidence is sought for a proposed novel mechanism by which PARP1/2 might act with TIMELESS-TIPIN to stabilize replisomes at TRCs in order to protect cell viability in HR-deficient cells. According to this TRC-centric view and the proposed joint

replisome-stabilizing role of TIMELESS-PARP1/2, it is suggested that PARP trapping on DNA might not be the only route to effective clinical synthetic lethality with HR deficiency. The MS is clearly written and presented, original, and the methodology is state-of-the-art and appropriate to address the questions asked. Related work is referenced for context.

The current MS extends on recent work (ref. 32) by the authors and others showing that TRCs in HR-deficient cells contribute to DNA damage, and that at least some of this damage can be repaired after cells enter mitosis (upon release of cells from Cdk1 inhibitor RO-3306, dubbed MiDAS). In ref. 32, sites of MiDAS in HR-deficient cells were associated with early-replicating regions.

Here, the replication-pausing complex factors TIMELESS and TIPIN are implicated in TRC damage by showing that their depletion in HeLa cells (Fig. 1) results in elevated damage markers in S phase, which is partially suppressed by inhibiting transcriptional elongation (DRB treatment). A similar effect is observed in HeLa cells treated with PARP inhibitors +/-DRB (Fig. 2). DNA damage, which can be partially suppressed by DRB, is also observed when PARP1/2 are simultaneously depleted from HeLa cells (Fig. 3). These findings are interesting, albeit perhaps not entirely surprising given that TRCs and PARPi are well-established proponents of replication stress, and PARPs as well as TIMELESS-TIPIN are required for proper DNA repair and replication progression. However, the authors then propose the attractive hypothesis that PARP1/2 may serve as sensors of TRCs, "communicating these signals to the replisome via TIMELESS".

TIMELESS-TIPIN and TIMELESS-PARP1 have previously been reported as mutually exclusive complexes; the TIMELESS-PARP1 interaction has been linked to TIMELESS recruitment to DNA double-strand breaks in a role distinct from replication fork pausing, instead relating to timely break repair (refs. 36, 37). Depletion-of-TIMELESS-induced DNA damage is shown herein to be suppressed to a greater extent by re-expression of WT TIMELESS as compared to TIMELESS R1081G (a mutant shown in ref. 37 to compromise PARP1 interactions). Interestingly, depletion of TIMELESS or TIPIN or PARP1 (as previously also shown in ref. 5) or PARP2 (and to slightly higher degree PARP1 and 2) are shown to be synthetic sick/lethal with BRCA2/HR-deficiency (Fig. 5) by colony formation assays. Finally, mitotic DNA damage and synthetic lethality of PARP inhibitors with HR-deficiency is shown to be partially suppressed by DRB, indicating at least a partial dependence on transcription elongation (Fig. 6).

The authors interpret their findings such that a failure to resolve TRCs (and the resulting DNA damage) significantly underlies the synthetic lethality of PARP inhibitors in HR-deficient cells. In a departure from conventional models that suggest this well-established synthetic lethality is caused predominantly by perturbed PARP-release from DNA lesions in the presence of PARPi, leading to collisions with DNA replication and DNA damage requiring HR repair, the authors propose that the replication fork pausing complex components TIMELESS-TIPIN interact with PARP1/2 at TRCs to stabilize replisomes and protect TRC intermediates from damage, which otherwise persists into mitosis where the damage gives rise to mitotic DNA synthesis (MiDAS) and may ultimately be responsible for synthetic lethality (although a contribution of transcription elongation to cell death upon PARP1/2 depletion is not directly demonstrated). This would suggest that HR-deficient cancer cells could potentially be susceptible to

targeted PARP1/2 degradation as a possible alternative to the use of PARP-trapping inhibitors currently used in the clinic. The MS thus proposes an interesting concept with potentially important biomedical implications.

The links uncovered in this work between TRCs and DNA damage in scenarios of PARP inhibition as well as PARP depletion are convincing and will be of interest to many in the field. However, the notion that “we describe here a novel role of PARP1 and 2, together with the proteins TIMELESS and TIPIN, in protecting the replisome from conflicts with the transcription machinery”, which constitutes a significant part of the MS/novelty, is insufficiently supported by the data presented. “In support of this mechanism, depletion of TIMELESS or TIPIN or of both PARP1 and 2 led to synthetic lethality with HR deficiency”. This correlation in survival phenotypes shown in HeLa and DLD1 (colorectal cancer) cells could also be consistent with independent roles of the replication pausing complex and PARP1/2 in promoting DNA repair/replication progression, generally, or in these specific cellular backgrounds. Additional evidence and clarifications would be needed to demonstrate more clearly a molecular mechanism of joint action by TIMELESS-TIPIN and PARP1/2 in protecting replisomes and cells from TRC damage as the basis for PARPi synthetic lethality with HR-deficiency. Another limitation pointed out on p. 8 is that the “mechanism by which PARP1 and 2 sense the presence of TRCs and how they transmit this information to the replisome” has not been addressed. It would not seem too difficult to strengthen the case by investigating the invoked functional interaction of PARP1 with topoisomerase I in this context.

Specific points:

- A key question is whether the synthetic lethality upon PARP1&2 depletion in BRCA2-deficient backgrounds as shown in Fig. 5 is dependent on transcription elongation. Perhaps this could be addressed by testing the impact of a temporary DRB step or similar, but may require more acute PARP loss such as by using a degron tag on endogenous PARPs?

- DNA damage upon TIMELESS or TIPIN depletion is observed in about 50% of cells in Ext. Data Fig. 5d,e. This is not further increased upon depletion of PARP1/2 (perhaps slightly with PARPi). The apparent epistasis is taken as support for the suggestion on p. 6 that “PARP1 and 2 functioning in the same molecular pathway as TIMELESS/TIPIN”. Clearly, under the conditions employed the overriding factor for the observed phenotype is depletion of TIMELESS or TIPIN. How effectively is PARP1 and 2 depleted in conjunction with TIMELESS or TIPIN in these cells by WB? How do the cell cycle profiles compare between the various cell cultures depleted for one or more factor(s)? How do individual and combined depletion affect the number of gH2AX and 53BP1 foci per cells (damage levels could change without affecting the reported percentage of focus-positive cells)?

- Besides phenotypic correlations, it is Fig. 4 that addresses a potential mechanistic link between TIMELESS and PARP1 directly. Re-expression of a TIMELESS R1081 PARP1 interaction mutant (ref. 37) is less effective than TIMELESS WT in suppressing S-phase damage upon endogenous TIMELESS depletion. Does a quantification of re-expression levels (WB) suggest similar amounts of WT/mutant TIMELESS?

Would a similar effect be achieved by siPARP and re-expression of the complementary PARP1 D993G mutant defective in TIMELESS binding? (Does the over-expressed TIMELESS R1081 fail to co-IP PARP1 as compared to re-expressed WT as expected?)

- In a related point, TIMELESS has been reported to specifically interact with PARP1. If TRC-related DNA damage is a reflection of loss of a joint activity of TIMELESS with PARP1, how can it be explained that the transcription-dependent damage phenotypes are virtually identical upon depletion of PARP1 or PARP2 (where TIMELESS-PARP1 actions should be unperturbed) (Fig. 3j)?

- In Fig. 1h, data should be presented to the same scale for siTIM as for siCTRL/siTIP. Could apparent enhanced progression of replication into gene bodies for siTIM/siTIP relate to overall lower DNA replication signals elsewhere, as compared to control?

- It would be helpful if cell-cycle profiles could be provided for the various cell cycle-critical experiments shown throughout the MS.

- Additional information regarding the numbers of cells analyzed and statistical tests used should be added.

Referee #3:

In this work Petropoulos and colleagues present data consistent with the idea that PARP trapping is not the main mechanism for PARPi sensitivity of HR-deficient cells, but that it is (mostly) the result of inhibition of PARP1/2 activity and the impact of this on avoiding DNA damage resulting from transcription-replication conflicts. PARP1/2 are suggested to act together with the fork protection complex components TIMELESS and TIPIN, which are known to pause and stabilise the replisome in difficult to replicate regions, including regions of high transcriptional activity (evidence to support this in human cells is presented by the authors in Fig. 1h). Knockdown of TIM, TIP, PARP1/2 as well as the use of PARPis was shown to lead to DSBs (mostly in early S-phase, during which most regions containing highly expressed genes are replicated), whereas the use of the transcription elongation inhibitor DRB largely prevented this damage. Synthetic lethality of TIM, TIP and/or PARP1/2 with BRCA2 knockdown indicated that the combined action of the FPC and PARPs is necessary for survival in the absence of functional HR. Synthetic lethality between loss of PARP1/2 and HR deficiency is the key evidence presented that PARP trapping is not the main driver of DNA damage and cell death upon PARP inhibition. Finally, DRB substantially reduced DNA damage and cell death of BRCA2^{-/-} cells exposed to PARPi, consistent with a key role for TRCs.

This is a well written paper with high quality data and an important message with both fundamental and translational/clinical implications of interest to a broad audience.

Based on the evidence presented I agree that a role for TRCs seems likely. However, I feel this could be

further strengthened.

1. One concern I have is how heavily the conclusion of the importance of TRCs is dependent on the use of a single transcription elongation inhibitor, DRB.

- To ensure that there are no confounding effects and to strengthen the conclusion that TRCs are key, can the authors exclude any (indirect) impact of DRB on replication fork progression or DNA damage signalling?

- Do alternative transcription inhibitors with different mechanisms (alpha-amanitin, triptolide) give similar results? Showing this for a few key experiments would be beneficial.

- Showing the opposite effect of increased transcription would greatly strengthen the conclusion. This could for example be achieved by overexpression of TBP (Kotsanis et al., 2016, doi:10.1038/ncomms13087), which among other things would be predicted to increase PARPi sensitivity (of BRCA2^{-/-} cells in particular) and might even be synthetic lethal with HR deficiency.

2. Rescue with DRB is incomplete. Although this may well be the result of incomplete inhibition of transcription. Can the authors exclude some additional contribution of PARP trapping? Also because the effect of PARPi appears to be significantly stronger than that of PARP1/2 knockdown (ED Fig. 4d,e). Perhaps the conclusion could be moderated slightly: TRCs may well play a key role, but trapping may still contribute.

3. In the discussion, the authors acknowledge that it is not yet clear how PARP1/2 may detect TRCs. The data presented in Fig. 4 are said to be consistent with “with PARP1 & 2 serving as sensors of TRCs and communicating these signals to the replisome via TIMELESS”. Although this is true in principle, initial detection might equally involve the FPC, with TIMELESS recruiting PARPs when encountering TRCs. Data from Young et al. (2015, doi:10.1016/j.celrep.2015.09.017) are consistent with TIMELESS recruiting PARP1: “PARP1 binding to certain substrates and their recruitment to DNA damage lesions is impaired by TIMELESS knockdown”. In this case increased PARylation upon siTIM could be the general consequence of the observed increase in DNA damage, and increased DNA damage for the R1081G mutant due to failure to recruit PARPs. PARP activity at a paused fork might then reinforce fork stalling, prevent DNA damage and recruit factors to resolve the TRC.

Unless the authors have good arguments to discount this possibility or can think of experiments to distinguish between these possibilities, they should at least briefly discuss alternative models to the one presented.

Other comments:

- Please clearly define “gH2AX positive cells” and “cells with 53BP1/RAD51 foci”. Was a single focus sufficient? For gH2AX this is particularly unclear because a number of different methods appear to have been used to detect it without indicating what method is used where.

- Fig. 1b: gH2AX +DRB images for siCTRL and siTIM are identical. Presumably a copy/paste error?

- After Fig. 2b the concentrations of PARPis are no longer indicated. Please provide concentrations for all experiments.

- ED Fig. 2: How representative are these? Would a statistical test for genome-wide impacts be possible? Error for Chr2 ruler scale (2 Mb)? Aren't the ruler scales 200kb? Are the bins really 10 kb (resolution seems higher than that)?

- In a few places the text does not accurately describe the data. The authors may wish to more carefully phrase the following:

p 3, "TIMELESS or TIPIN induced γ H2AX only in the EdU-positive cells". It was indeed a lot less in EdU -ve cells, but not completely absent.

p 4, "The four PARP inhibitors also induced..." For Fig. 2e only two inhibitors were used. This also made me wonder whether the other two did not give expected results or whether they were not tested. Please clarify or include data for all.

p 6, "...which was fully rescued by the ectopic..." This is not quite correct.

p 7, "...nor did they compromise cell viability..." Impact was small, but not absent.

- Please give additional details in Methods for
- Empty vector, GFP-TIM and GFP-TIMR1081G plasmids
- Dilutions for all antibodies
- EdU-seq data processing. Please define "the highest quality score".

Author Rebuttals to Initial Comments:

Response to Referees' Comments

We thank all the Referees for their constructive comments. We have tried to address all the comments that were raised and we believe that all the main issues have been addressed. We also provide more evidence that the synthetic lethality of PARP inhibitors with HR deficiency does not require trapping of PARPs on DNA. Moreover that the ability of PARP inhibitors to induce lethality of HR proficient cells is dependent on trapping. Thus, trapping narrows the therapeutic window of PARP inhibitors. We believe that this is an important point to make, as it may lead to the development of less toxic PARP inhibitors. Indeed, PARP inhibitor toxicity is a serious problem in the clinic; at the latest Annual ASCO Meeting there was one full session dedicated to this issue.

The current version of the manuscript represents a major improvement over the original version. Figs 1h; 3c,e; 4g,h,j; 6d,f,h and Extended Data Figs 1f,g,h,i; 3b; 4d,e; 5; 6a,f,h; 7b,d,e show new experiments that mostly focus on addressing the role of PARP trapping on lethality of HR-deficient and HR-proficient cells.

Overall, we hope that the Referees will appreciate the large number of new experiments in this revised version of the manuscript, guided, of course, by their comments.

Referee #1:

“Petroopoulos et al. examined the connection between transcription-replication conflicts (TRC), depletion of S-phase checkpoint proteins TIMELESS or TIPIN, and PARP1/2 depletion or inhibition in HR-deficient (HRD) cells. They found that depletion of TIMELESS and TIPIN causes faster fork progression and DNA damage in S-phase cells in a transcription-dependent fashion and that this damage can be repaired in mitosis by MiDAS (mitotic DNA synthesis). PARP inhibitors phenocopy TIMELESS or TIPIN depletion and induce highest levels of DNA damage when applied in early S-phase when the number of expressed genes is the highest. Combined PARP1/2 siRNA-mediated depletion, PARPi and TIMELESS or TIPIN siRNA-mediated depletion have the same effect on transcription-dependent DNA damage in S-phase, suggesting that they are epistatic. TIMELESS was previously shown to interact with PARP1 independent of the PARP1 catalytic activity and TIMELESS PARP1-binding mutant cannot rescue the effects of TIMELESS depletion on TRC-dependent DNA damage. Depletion of PARP1 and PARP2 or TIMELESS/TIPIN is synthetic lethal with HRD caused by BRCA mutations. Inhibition of transcription elongation abrogates PARPi-induced synthetic lethality with BRCA deficiency, but only if PARPi is administered already during early S-phase. Collectively, these results show that TRCs caused by PARP1/2 depletion, PARPi or TIMELESS/TIPIN depletion result in DNA damage that cannot be repaired in HRD cells, and that TRCs contribute to synthetic lethality between PARPi and HRD. Given that this study focuses on TRCs as a mechanism of PARP1- and PARPi-induced synthetic lethality with HRD, a few key points need to be addressed with regard to the measurement of TRCs, PARP1 connection with TRCs, and synthetic lethality induced by PARP1 depletion vs PARP1 inhibition.”

We thank the Referee for the comprehensive review of our findings.

1) Synthetic lethality between PARPi and HRD was previously shown to be caused by PARP trapping and not PARP depletion (PMID: 23118055). In the Discussion the authors provide possible explanations for these conflicting results. However, this needs to be properly tested using multiple cell lines WT vs BRCA-mutated from different types of cancer (e.g., breast, ovarian, pancreatic) to show whether indeed PARP1 depletion shows synthetic lethality with BRCA mutations in cancer cells.

In PMID 23118055 (Murai et al., 2012), the authors show that olaparib is less effective in inducing death of HR-proficient cells, after knocking out the PARP1 gene, suggesting that death is mediated by PARP trapping (Fig. 1D). However, the authors did not perform a similar experiment with HR-deficient cells. In this revised version of our manuscript we show that PARP trapping mediates death of HR-proficient cells, similar to Muraj et al., but is not required for PARP inhibitors to mediate death of HR-deficient cells (Fig. 6b,i; Ext Data Fig. 7b,d,e). We also demonstrated lethality of two additional HR-deficient cell lines, PEO1 and OVSAHO, after depleting PARP1 by siRNA (Ext Data Fig. 6e,h).

2) If PARP-related DNA damage phenotypes are due to TRCs, then the authors should test whether resolution of TRCs with RNase H overexpression can rescue the phenotypes.

Referee Fig.1

We performed this experiment. RNase H overexpression suppressed DNA damage induced by depletion of TIMELESS/TIPIN (Referee Fig. 1a) and by PARP inhibitors (Referee Fig. 1b). We would like to keep these data for another manuscript, so we hope that the Referee will be satisfied by us showing the data here. Of course, if the Referee wants us to include these data in this manuscript, then we will do so.

3) If TRCs are responsible for PARPi + HRD synthetic lethality, then induction of TRCs through camptothecin (Topo I inhibitor) should also show synthetic lethality with HRD.

Referee Fig.2

We did this experiment and demonstrated that camptothecin (CPT) has a stronger effect on HR-deficient, than HR-proficient cells (Referee Fig. 2). However, CPT affects replication and

transcription more broadly and not only in the context of transcription-replication conflicts. Hence, it was not as selective as PARP inhibitors.

4) DRB induces Pol II pausing, which was shown to induce DNA-RNA hybrid formation at transcription start sites where Pol II is paused (PMID: 29104020). In DRB-treated cells the levels of DNA-RNA hybrids and therefore TRCs, which can result in DNA damage, is expected to be higher. Therefore, it is essential to use another method, in addition to the proximity ligation assay, to examine DNA-RNA hybrids, such as RNase H hybrid-binding domain (HBD) or catalytic mutant.

Referee Fig. 3

We used cells expressing an RNase H catalytic mutant and show the presence of R-loops both after depletion of TIMELESS/TIPIN (Referee Fig. 3a) and by PARP inhibitors (Referee Fig. 3b). We would like to keep these data for another manuscript, so we hope that the Referee will be satisfied by us showing the data here. Of course, if the Referee wants us to include these data in this manuscript, then we will do so.

5) In relation to point 4, transcription can be inhibited with alpha-amanitin, which does not affect Pol II pausing and DNA-RNA hybrid formation.

Inhibition of transcription by alpha-amanitin is a slow, irreversible process requiring a few hours; in contrast, inhibition of transcription elongation by DRB is fast and reversible (PMID: 8760875). Because of the slow kinetics, alpha-amanitin was therefore not suitable to study TRCs in early S phase. To address the comment of the Referee we studied two additional inhibitors of transcription elongation: cordycepin (CORD) and triptolide (TPL). First, we documented that they inhibited transcription in our cells (Extended Data Fig. 1g), similar to what is reported in the literature, and then showed that, similar to DRB, they suppressed the induction of DNA damage by TIMELESS/TIPIN depletion (Extended Data Fig. 1h) and by PARP inhibitors (Extended Data Fig. 3d).

6) If PARP1/2 'signal the presence of TRCs to TIMELESS and TIPIN' then PARP1/2 are expected to somehow recognize TRCs, i.e., DNA-RNA hybrids, which can be tested in vitro.

Indeed, PARP1 was recently shown to bind to R-loops in vitro; we now cite this study (PMID: 36794853). In addition, PARP1/2 may recognize topoisomerase I, which is recruited at sites of supercoiled DNA, associated with TRCs (PMID: 11602253; PMID: 33981998).

7) BRCA1 and BRCA2 depletion was shown to increase DNA-RNA hybrid levels (PMID: 25699710, PMID: 24896180). What are the levels of DNA-RNA hybrids in BRCA1/2-deficient cells +/- PARP or TIMELESS siRNA?

We regret that we did not do this experiment. It is part of the experiments we are planning for our next manuscript on this topic. We hope that the Referee agrees that this is not the most critical experiment and that our conclusions are well supported by the data we present,

8) The presentation of EdU-seq results does not allow proper conclusions. In Fig. 1h the gene scale is 50 kb. Considering a fork speed of 2 kb/min, within 120 min of release from the thymidine block 60 kb genes would be already replicated. EdU incorporation from 90-120 min would mean that in WT cells EdU signal would be concentrated at the gene ends (after 40 kb) so the observed dip makes sense. However, in siTIMELESS and siTIPIN cells the EdU signal is evenly distributed throughout the transcribed gene, which can be interpreted as:
i) replication fork progression is slower – not faster as the authors interpret, OR
ii) replication forks are faster and move into downstream genes – as the authors interpret. To distinguish between the two scenarios, the authors should show a longer scale (e.g., 100 kb).

We regret a mistake in the scale shown in Fig 1h. It is not 50 kb, as we had marked, but 500 kb. Therefore, the only correct interpretation is that replication of transcribed genes is faster when TIMELESS/TIPIN are depleted. We repeated this experiment to monitor fork progression at both 90 and 120 min after release from the thymidine block (new Fig. 1h). Comparison of the 90 and 120 min data shows that we are monitoring fork progression; the genomic domains shown correspond to large (greater than 300 kb), transcribed genes (indicated by the green color in the gene annotation panel). The “old” Fig. 1h is now shown as Extended Data Fig. 2b. Please note that we also did similar experiments after depleting PARP1 and PARP2, in triplicate. The data show similar effects of depleting PARP1, as for depletion of TIMELESS/TIPIN (Fig. 4j; Extended Data Fig. 5).

Referee #2:

The MS “Transcription-replication conflicts sensitize BRCA2-deficient cells to PARP inhibitors” addresses the highly topical and incompletely understood issue of the clinically-relevant synthetic lethality afforded by PARP inhibitors in homologous recombination (HR)-deficient cells. On the one hand, the MS convincingly relates transcription-dependent S-phase DNA damage to PARP inhibition as well as PARP1/2 loss, identifying a potential source of synthetic lethality in transcription-replication conflicts (TRCs). In a slightly more preliminary line of investigation, evidence is sought for a proposed novel mechanism by which PARP1/2 might act with TIMELESS-TIPIN to stabilize replisomes at TRCs in order to protect cell viability in HR-deficient cells. According to this TRC-centric view and the proposed joint replisome-stabilizing role of TIMELESS-PARP1/2, it is suggested that PARP trapping on DNA might not be the only route to effective clinical synthetic lethality with HR deficiency. The MS is clearly written and presented, original, and the methodology is state-of-the-art and appropriate to address the questions asked. Related work is referenced for context.

We thank the Referee for this summary of our findings.

The current MS extends on recent work (ref. 32) by the authors and others showing that TRCs in HR-deficient cells contribute to DNA damage, and that at least some of this damage can be repaired after cells enter mitosis (upon release of cells from Cdk1 inhibitor RO-3306, dubbed MiDAS). In ref. 32, sites of MiDAS in HR-deficient cells were associated with early-replicating regions.

Here, the replication-pausing complex factors TIMELESS and TIPIN are implicated in TRC damage by showing that their depletion in HeLa cells (Fig. 1) results in elevated damage markers in S phase, which is partially suppressed by inhibiting transcriptional elongation (DRB treatment). A similar effect is observed in HeLa cells treated with PARP inhibitors +/-DRB (Fig. 2). DNA damage, which can be partially suppressed by DRB, is also observed when PARP1/2 are simultaneously depleted from HeLa cells (Fig. 3). These findings are interesting, albeit perhaps not entirely surprising given that TRCs and PARPi are well-established proponents of replication stress, and PARPs as well as TIMELESS-TIPIN are required for proper DNA repair and replication progression. However, the authors then propose the attractive hypothesis that PARP1/2 may serve as sensors of TRCs, “communicating these signals to the replisome via TIMELESS”.

We thank the Referee for these comments.

TIMELESS-TIPIN and TIMELESS-PARP1 have previously been reported as mutually exclusive complexes; the TIMELESS-PARP1 interaction has been linked to TIMELESS recruitment to DNA double-strand breaks in a role distinct from replication fork pausing, instead relating to timely break repair (refs. 36, 37). Depletion-of-TIMELESS-induced DNA damage is shown herein to be suppressed to a greater extent by re-expression of WT TIMELESS as compared to TIMELESS R1081G (a mutant shown in ref. 37 to compromise PARP1 interactions). Interestingly, depletion of TIMELESS or TIPIN or PARP1 (as previously also shown in ref. 5) or PARP2 (and to slightly higher degree PARP1 and 2) are shown to be synthetic sick/lethal with BRCA2/HR-deficiency (Fig. 5) by colony formation assays. Finally, mitotic DNA damage and synthetic lethality of PARP inhibitors with HR-deficiency is shown to be partially suppressed by DRB, indicating at least a partial dependence on transcription elongation (Fig. 6).

Indeed, the two publications showing a physical interaction between TIMELESS and PARP1 (refs 36 and 37 in our original manuscript: Young et al, 2015, PMID: 26456830 and Xie et al, 2015, PMID: 26344098) studied the role of TIMELESS in break repair. Since they did not study the role of TIMELESS and PARP1 in DNA replication stress, their findings neither contradict, nor support our findings. We further note that the cryo-EM structure of the human replisome shows residues 7-803 of TIMELESS well-ordered and in contact with TIPIN, whereas the C-terminal 400 amino acids of TIMELESS that contain the PARP-binding domain are not ordered in the structure (Jones et al., 2021; PMID: 34694004). Therefore, the structure suggests that PARP1 and TIPIN can bind to TIMELESS at the same time, i.e. in a non-mutually exclusive manner, since their binding sites on TIMELESS are distinct.

The authors interpret their findings such that a failure to resolve TRCs (and the resulting DNA damage) significantly underlies the synthetic lethality of PARP inhibitors in HR-deficient cells. In a departure from conventional models that suggest this well-established synthetic lethality is caused predominantly by perturbed PARP-release from DNA lesions in the presence of PARPi, leading to collisions with DNA replication and DNA damage requiring HR repair, the authors propose that the replication fork pausing complex components TIMELESS-TIPIN interact with PARP1/2 at TRCs to stabilize replisomes and protect TRC intermediates from damage, which otherwise persists into mitosis where the damage gives rise to mitotic DNA synthesis (MiDAS) and may ultimately be responsible for synthetic lethality (although a contribution of transcription elongation to cell death upon PARP1/2 depletion is not directly demonstrated).

We showed that inhibiting transcription elongation suppresses the synthetic lethality induced by TIMELESS/TIPIN depletion and by PARP inhibitors. For depletion of PARP1 and PARP2 we show that inhibiting transcription elongation suppresses the induction of DNA damage (Fig. 3i; Extended Data Fig. 4c). If requested, we are happy to do the experiment showing that inhibiting transcription elongation also suppresses the synthetic lethality induced by PARP1 and PARP2 depletion. The revised manuscript includes many new experiments addressing the

role of PARP trapping on the synthetic lethality with HR deficiency (Fig. 6 and Extended Data Fig. 7 are entirely new and devoted to this question).

This would suggest that HR-deficient cancer cells could potentially be susceptible to targeted PARP1/2 degradation as a possible alternative to the use of PARP-trapping inhibitors currently used in the clinic. The MS thus proposes an interesting concept with potentially important biomedical implications.

Referee Fig. 4

We agree with the Referee that this could be a way forward in the clinic. We did some experiments along these lines. We asked a company to synthesize compound SK-575, described by Cao et al, 2020 (PMID: 32924477) for us. SK-575 is a derivative of olaparib that targets PARP1 for ubiquitin-mediated degradation. Unfortunately, SK-575 did not degrade PARP1 very efficiently and at high concentrations did not degrade PARP1 at all (Referee Fig. 4). These results are similar to those obtained by Cao et al. Moreover, PARP2 was inhibited, but not degraded. In biochemical and cell-based assays, SK-575 behaved almost identical to olaparib. For these reasons, we do not discuss our experiments with SK-575 in our manuscript.

The links uncovered in this work between TRCs and DNA damage in scenarios of PARP inhibition as well as PARP depletion are convincing and will be of interest to many in the field. However, the notion that “we describe here a novel role of PARP1 and 2, together with the proteins TIMELESS and TIPIN, in protecting the replisome from conflicts with the transcription machinery”, which constitutes a significant part of the MS/novelty, is insufficiently supported by the data presented. “In support of this mechanism, depletion of TIMELESS or TIPIN or of both PARP1 and 2 led to synthetic lethality with HR deficiency”. This correlation in survival phenotypes shown in HeLa and DLD1 (colorectal cancer) cells could also be consistent with independent roles of the replication pausing complex and PARP1/2 in promoting DNA repair/replication progression, generally, or in these specific cellular backgrounds. Additional evidence and clarifications would be needed to demonstrate more clearly a molecular mechanism of joint action by TIMELESS-TIPIN and PARP1/2 in protecting replisomes and cells from TRC damage as the basis for PARPi synthetic lethality with HR-deficiency.

We agree with the Referee that similar phenotypes of TIMELESS/TIPIN and PARP1 inhibition/depletion do not constitute evidence of joint action. However, we present experiments that indicate joint action. The crystal structure of the TIMELESS-PARP1 complex (Xie et al, 2015, PMID: 26344098) identified residues R1081 of TIMELESS and D993 of PARP1 as critical for this interaction. In Fig. 4e,f, we show that a TIMELESS R1081G mutant is defective in protecting replication forks from TRCs. In Fig. 4h, we show that a PARP1 D993G mutant is also defective in protecting replication forks from TRCs. These results suggest that the physical interaction between TIMELESS and PARP1 is needed for the functions of both TIMELESS and PARP1 to protect the replisome from TRCs and, therefore, indicates joint action.

Another limitation pointed out on p. 8 is that the “mechanism by which PARP1 and 2 sense the presence of TRCs and how they transmit this information to the replisome” has not been

addressed. It would not seem too difficult to strengthen the case by investigating the invoked functional interaction of PARP1 with topoisomerase I in this context.

We respectfully would like to suggest that the mechanism by which PARP1 and 2 sense the presence of TRCs and how they transmit this information to the replisome should be the topic of a follow-up study. There is already evidence in the literature that PARP1 and TIMELESS function together in protecting replication forks (Chaudhuri et al, 2012; PMID: 22388737) and a very recent study reported that PARP1 binds in vitro to R-loops (Laspata et al, 2023; PMID: 36794853). We discuss these studies in the revised version of our manuscript. In regard to how PARP1 transmits the information about TRCs to the replisome, we note that our study provides some preliminary insights by showing that the direct interaction of PARP1 with TIMELESS is functionally important.

Specific points:

- A key question is whether the synthetic lethality upon PARP1&2 depletion in BRCA2-deficient backgrounds as shown in Fig. 5 is dependent on transcription elongation. Perhaps this could be addressed by testing the impact of a temporary DRB step or similar, but may require more acute PARP loss such as by using a degron tag on endogenous PARPs?

The Referee raised this point also above. We showed that inhibiting transcription elongation suppresses the synthetic lethality induced by TIMELESS/TIPIN depletion and by PARP inhibitors. For depletion of PARP1 and PARP2 we show that inhibiting transcription elongation suppresses the induction of DNA damage (Fig. 3i; Extended Data Fig. 4c). If requested, we are happy to do the experiment showing that inhibiting transcription elongation also suppresses the synthetic lethality induced by PARP1 and PARP2 depletion. The revised manuscript includes many new experiments addressing the role of PARP trapping on the synthetic lethality with HR deficiency (Fig. 6 and Extended Data Fig. 7 are entirely new and devoted to this question).

- DNA damage upon TIMELESS or TIPIN depletion is observed in about 50% of cells in Ext. Data Fig. 5d,e. This is not further increased upon depletion of PARP1/2 (perhaps slightly with PARPi). The apparent epistasis is taken as support for the suggestion on p. 6 that "PARP1 and 2 functioning in the same molecular pathway as TIMELESS/TIPIN". Clearly, under the conditions employed the overriding factor for the observed phenotype is depletion of TIMELESS or TIPIN. How effectively is PARP1 and 2 depleted in conjunction with TIMELESS or TIPIN in these cells by WB? How do the cell cycle profiles compare between the various cell cultures depleted for one or more factor(s)? How do individual and combined depletion affect the number of γ H2AX and 53BP1 foci per cells (damage levels could change without affecting the reported percentage of focus-positive cells)?

The most critical experiments to show that TIMELESS/TIPIN and PARP1 and PARP2 are in the same pathway are the experiments described in Fig. 4e,f,h, expressing mutant TIMELESS and PARP1 proteins targeting the residues at the TIMELESS-PARP1 interface. In the experiment shown in Extended Data Fig. 5 (Extended Data Fig. 4e in the revised manuscript), we combined TIMELESS/TIPIN depletion with PARP inhibitors or with PARP1 and PARP2 depletion. Western blots for specific siRNA combinations are shown in Extended Data Fig. 4d; cell cycle profiles are shown in Extended Data Fig. 4f; the results do not raise any concerns and we understand that it is very legitimate for the Referee to raise these points.

We used a threshold to determine the percentage of γ H2AX and 53BP1-positive cells, as is customary in the field. Given that we consider this to not be the most critical experiment to show that TIMELESS/TIPIN and PARP1/PARP2 function together, we did not count the number of foci per cell, but inspection of the microscopy images did not reveal an obvious difference in number of foci per cell as a result of combining treatments.

- Besides phenotypic correlations, it is Fig. 4 that addresses a potential mechanistic link between TIMELESS and PARP1 directly. Re-expression of a TIMELESS R1081 PARP1 interaction mutant (ref. 37) is less effective than TIMELESS WT in suppressing S-phase damage upon endogenous TIMELESS depletion. Does a quantification of re-expression levels (WB) suggest similar amounts of WT/mutant TIMELESS? Would a similar effect be achieved by siPARP and re-expression of the complementary PARP1 D993G mutant defective in TIMELESS binding? (Does the over-expressed TIMELESS R1081 fail to co-IP PARP1 as compared to re-expressed WT as expected?)

We performed the experiment suggested by the Referee with the PARP1 D993G mutant and observed decreased capacity to rescue TRC-induced DNA damage, similar to what we observed with the TIMELESS R1081G mutant (Fig. 4h).

- In a related point, TIMELESS has been reported to specifically interact with PARP1. If TRC-related DNA damage is a reflection of loss of a joint activity of TIMELESS with PARP1, how can it be explained that the transcription-dependent damage phenotypes are virtually identical upon depletion of PARP1 or PARP2 (where TIMELESS-PARP1 actions should be unperturbed) (Fig. 3j)?

Depletion of PARP1 and PARP2 gave similar results in experiments monitoring TRC-induced DNA damage by γ H2AX and 53BP1 foci. But in a new experiment that we report in the revised manuscript and is more TRC-specific, PARP1 is seen to be more important than PARP2. Specifically, depletion of PARP1, but not PARP2, enhances fork progression over transcribed genes, as revealed by high throughput sequencing (Fig. 4j and Extended Data Fig. 5). This is consistent with PARP1, but not PARP2, interacting with TIMELESS, which is the protein that slows fork progression. We reconcile these findings, by considering that more than one mechanisms may be used to sense TRCs. For example, both PARP1 and PARP2 may bind R-loops and both may interact with topoisomerase I. PARP1 would be more efficient to transmit the signal to the replisome, since it can interact with TIMELESS, but PARP2 may also be able to transmit the signal. As we understand how PARP1 and PARP2 communicate with the replisome (for example, do they PARylate a subunit of the replisome?), we will be able to better understand the roles of PARP1 and PARP2 in TRCs.

- In Fig. 1h, data should be presented to the same scale for siTIM as for siCTRL/siTIP. Could apparent enhanced progression of replication into gene bodies for siTIM/siTIP relate to overall lower DNA replication signals elsewhere, as compared to control?

The sequencing data are set to the same scale as far as the average replication in the early replicating part of the genome is concerned. The sigma value on the y axis is determined by the background signal at the non-replicating parts of the genome. So the samples with higher signal values had less noise at the non-replicating parts of the genome.

- It would be helpful if cell-cycle profiles could be provided for the various cell cycle-critical experiments shown throughout the MS.

We provide cell cycle profile data in Extended Data Figs 1f, 4f and 6a.

- Additional information regarding the numbers of cells analyzed and statistical tests used should be added.

We added this information in the Methods section.

Referee #3:

In this work Petropoulos and colleagues present data consistent with the idea that PARP trapping is not the main mechanism for PARPi sensitivity of HR-deficient cells, but that it is (mostly) the result of inhibition of PARP1/2 activity and the impact of this on avoiding DNA damage resulting from transcription-replication conflicts. PARP1/2 are suggested to act together with the fork protection complex components TIMELESS and TIPIN, which are known to pause and stabilise the replisome in difficult to replicate regions, including regions of high transcriptional activity (evidence to support this in human cells is presented by the authors in Fig. 1h). Knockdown of TIM, TIP, PARP1/2 as well as the use of PARPis was shown to lead to DSBs (mostly in early S-phase, during which most regions containing highly expressed genes are replicated), whereas the use of the transcription elongation inhibitor DRB largely prevented this damage. Synthetic lethality of TIM, TIP and/or PARP1/2 with BRCA2 knockdown indicated that the combined action of the FPC and PARPs is necessary for survival in the absence of functional HR. Synthetic lethality between loss of PARP1/2 and HR deficiency is the key evidence presented that PARP trapping is not the main driver of DNA damage and cell death upon PARP inhibition. Finally, DRB substantially reduced DNA damage and cell death of BRCA2-/- cells exposed to PARPi, consistent with a key role for TRCs.

This is a well written paper with high quality data and an important message with both fundamental and translational/clinical implications of interest to a broad audience.

We thank the Referee for the above comments.

Based on the evidence presented I agree that a role for TRCs seems likely. However, I feel this could be further strengthened.

1. One concern I have is how heavily the conclusion of the importance of TRCs is dependent on the use of a single transcription elongation inhibitor, DRB.

- To ensure that there are no confounding effects and to strengthen the conclusion that TRCs are key, can the authors exclude any (indirect) impact of DRB on replication fork progression or DNA damage signalling?

In Extended Data Fig. 1f, we show that DRB and two other transcription elongation inhibitors (see next comment) do not affect cell cycle progression. In Extended Data Fig. 1i, we show that DRB on its own (siCTRL +DRB) does not induce γ H2AX expression.

- Do alternative transcription inhibitors with different mechanisms (alpha-amanitin, triptolide) give similar results? Showing this for a few key experiments would be beneficial.

We used cordycepin and triptolide. We documented that they inhibited transcription in our cells (Extended Data Fig. 1g), similar to what is reported in the literature, and then showed that, similar to DRB, they suppressed the induction of DNA damage by TIMELESS/TIPIN depletion (Extended Data Fig. 1h) and by PARP inhibitors (Extended Data Fig. 3d).

- Showing the opposite effect of increased transcription would greatly strengthen the conclusion. This could for example be achieved by overexpression of TBP (Kotsanis et al., 2016, doi:10.1038/ncomms13087), which among other things would be predicted to increase PARPi sensitivity (of BRCA2-/- cells in particular) and might even be synthetic lethal with HR deficiency.

Referee Fig. 5

We employed the system of TBP overexpression described by Kotsanis et al. TBP overexpression (OE) induced a DNA damage response in early S phase, as compared to normal expression (NE; i.e. no induction of expression of ectopic TBP) (Referee Fig. 5a,b). However, the level of induction was less than the one observed with PARP inhibitors (see for example, Extended Data Fig. 7e, which uses the same assay format). Overexpression of TBP induced more lethality in *BRCA2*^{-/-} cells, than in *BRCA2*^{+/+} cells (Referee Fig. 5c,d); the difference between *BRCA2*^{-/-} and *BRCA2*^{+/+} cells was statistically significant, but the magnitude of the effect was not impressive, which fits with the lower levels of induction of DNA damage response.

2. Rescue with DRB is incomplete. Although this may well be the result of incomplete inhibition of transcription. Can the authors exclude some additional contribution of PARP trapping? Also because the effect of PARPi appears to be significantly stronger than that of PARP1/2 knockdown (ED Fig. 4d,e). Perhaps the conclusion could be moderated slightly: TRCs may well play a key role, but trapping may still contribute.

We studied more extensively the role of PARP trapping in this revised version of the manuscript (Fig. 6 and Extended Data Fig. 7). PARP trapping induces lethality in HR-proficient cells. In HR-deficient cells, inhibition of PARP activity induces very strong lethality and the contribution of lethality by PARP trapping is much less. However, PARP trapping can be important for killing HR-deficient cells in the context of PARP inhibitors that are not very potent in inhibiting the enzymatic activity of PARPs (for example, Fig. 6f, right-bottom panel: TMZ potentiates PARP trapping by olaparib and decreases 65-fold the IC50 value of killing of *BRCA2*^{-/-} cells by olaparib).

3. In the discussion, the authors acknowledge that it is not yet clear how PARP1/2 may detect TRCs. The data presented in Fig. 4 are said to be consistent with “with PARP1 & 2 serving as sensors of TRCs and communicating these signals to the replisome via TIMELESS”. Although this is true in principle, initial detection might equally involve the FPC, with TIMELESS recruiting PARPs when encountering TRCs. Data from Young et al. (2015, doi:10.1016/j.celrep.2015.09.017) are consistent with TIMELESS recruiting PARP1: “PARP1 binding to certain substrates and their recruitment to DNA damage lesions is impaired by TIMELESS knockdown”. In this case increased PARylation upon siTIM could be the general consequence of the observed increase in DNA damage, and increased DNA damage for the R1081G mutant due to failure to recruit PARPs. PARP activity at a paused fork might then reinforce fork stalling, prevent DNA damage and recruit factors to resolve the TRC.

Unless the authors have good arguments to discount this possibility or can think of experiments to distinguish between these possibilities, they should at least briefly discuss alternative models to the one presented.

We agree with the Referee that we cannot exclude the above possibility and, therefore, we mention it in the Discussion of the revised manuscript.

Other comments:

- Please clearly define “gH2AX positive cells” and “cells with 53BP1/RAD51 foci”. Was a single focus sufficient? For gH2AX this is particularly unclear because a number of different methods appear to have been used to detect it without indicating what method is used where.

For cells positive for γ H2AX, 53BP1 and RAD51, the thresholds were 20, 20 and 10 foci per cell, respectively (mentioned in the Methods section). For γ H2AX, in each graph, we indicate if number of foci (γ H2AX positive cells) or mean signal intensity over the entire nucleus were measured. These points are now more clearly explained in the Methods section.

- Fig. 1b: gH2AX +DRB images for siCTRL and siTIM are identical. Presumably a copy/paste error?

Thank you very much for noticing this! We fixed this error.

- After Fig. 2b the concentrations of PARPis are no longer indicated. Please provide concentrations for all experiments.

The concentrations of the PARP inhibitors in Fig. 2d,e are now mentioned in the figure legends.

- ED Fig. 2: How representative are these? Would a statistical test for genome-wide impacts be possible? Error for Chr2 ruler scale (2 Mb)? Aren't the ruler scales 200kb? Are the bins really 10 kb (resolution seems higher than that)?

Extended Data Fig. 2a: The bins are 10 kb, the ruler scale is 100 kb, each panel is 2 Mb. It would be very hard to do statistics for this. All the data are like this, we did not select “nice-looking” regions. Both the original fastq data and the processed data have been deposited on the SRA database.

- In a few places the text does not accurately describe the data. The authors may wish to more carefully phrase the following:

p 3, “TIMELESS or TIPIN induced γ H2AX only in the EdU-positive cells”. It was indeed a lot less in EdU -ve cells, but not completely absent.

We fixed the text; thank you.

p 4, “The four PARP inhibitors also induced...” For Fig. 2e only two inhibitors were used. This also made me wonder whether the other two did not give expected results or whether they were not tested. Please clarify or include data for all.

Indeed for Fig. 2e we only tested two inhibitors. The other two were not tested.

p 6, “...which was fully rescued by the ectopic...” This is not quite correct.

We fixed the text; thank you.

p 7, “...nor did they compromise cell viability...” Impact was small, but not absent.

We fixed the text; thank you.

- Please give additional details in Methods for
- Empty vector, GFP-TIM and GFP-TIMR1081G plasmids

These plasmids were previously described by Xie et al, 2015; PMID 26344089.

- Dilutions for all antibodies

These are now listed in the Methods section.

- EdU-seq data processing. Please define “the highest quality score”.

This is the highest score of the bwa alignment algorithm; it has a value of 60 and we use this highest score, so that we only consider sequence reads that are unambiguously and uniquely mapped on the human genome.

Reviewer Reports on the First Revision:

Referees' comments:

Referee #1:

I would like to thank the authors for addressing the points raised by the reviewers. However, the authors substantially changed the manuscript without highlighting the changes in the manuscript text and figures. I cannot find data from old Fig. 5, on which my first comment was based.

1) Plots provided in new Fig. 6 and Ext. Data Fig. 7 do not address my point regarding PARP depletion vs PARP inhibition causing synthetic lethality with HRD. To address this, the authors should compare viability of BRCA+ vs BRCA- cell lines treated with 1) PARPi (at IC50 concentration), 2) siPARP1, 3) siPARP2, and 4) siPARP1 + siPARP2 (similar to old Figure 5 in HeLa and DLD1). I requested for this experiment to be done in additional BRCA+/- cell lines to further support their results. It is not sufficient to use BRCA-deficient OVSAHO and PEO1 ovarian cancer cells without their matched BRCA+ counterparts, and colony formation results should be quantified (pictures of the dishes provided in Ext. Data Fig. 6h are not sufficient). From the plots in Fig. 6, it seems that PARP1/2 depletion (without PARPi) does not cause lethality but actually increases survival in most cases.

Instead, the authors examined the effect of combined PARP depletion and PARP inhibition. What is the rationale for this experiment? Data in Fig. 6h using siPARP1 addresses on- and off-targets of PARPi. If PARPi inhibits only PARP1, then PARP1 depletion should abrogate the lethality caused by PARPi. In BRCA-deficient cells talazoparib seems to exert its lethality by inhibiting PARP2, whereas in BRCA-proficient cells it also acts through PARP1 inhibition. AZD5305 is expected to be a selective PARP1 inhibitor. In BRCA-deficient cells siPARP1 cells are more sensitive to AZD5305, suggesting that it has off-target effects. Like talazoparib, in BRCA-deficient cells olaparib exerts its lethality by inhibiting PARP2. Based on data in Ext. Data Fig. 7, it seems that talazoparib has off-target effects, as combined siPARP1/2 has the same effect as siControl in BRCA-deficient cells treated with talazoparib. Unless the authors have an explanation for these results, I suggest removing them altogether from the manuscript (new Fig. 6 and related text) as they only distract from the main message and do not provide any solid conclusions.

Furthermore, the authors included analysis of PARP trapping without explaining how it was done in the Methods and without providing raw data. Results in Fig. 3e are surprising as PARP trapping requires induction of DNA damage (with TMZ, MMS, IR...). Based on extensive biochemical analysis, Rudolph and Luger (PMID: 33531508, 35259019, 37531469) conclude that PARPi differ in catalytic inhibition potencies, with talazoparib being the most potent inhibitor and that PARP trapping in cells correlates with the inhibition of PARP enzymatic activity. Your data point to differential PARPi potency in inhibiting catalytic activity, which is responsible for differential response of HR-proficient and HR-deficient cells to PARPi. I would revert to the old version of the manuscript with old Fig. 3 and related text.

2) An RNH1 rescue experiment is an essential control and must be provided in the same manuscript.

3) The experiment with CPT should be included in this manuscript.

4) Experiments with the RNH1 catalytic mutant should be included in this manuscript.

Referee #2:

I would like to thank the authors for considering the points raised. The revised version is improved and better incorporates/discusses the limitations of the study, although I may not have caught all the revision changes in the absence of clear pointers/highlights in the revised MS pdf. Regarding specific points raised in round 1, and the responses given by the authors in the rebuttal letter, can the authors please address the following open questions:

(1) From authors' reply: "If requested, we are happy to do the experiment showing that inhibiting transcription elongation also suppresses the synthetic lethality induced by PARP1 and PARP2 depletion." Please include this experiment, as this critical point remains unresolved, and please relate the outcomes to the model presented.

(2) From authors' reply: "We agree with the Referee that similar phenotypes of TIMELESS/TIPIN and PARP1 inhibition/depletion do not constitute evidence of joint action. However, we present experiments that indicate joint action. The crystal structure of the TIMELESS-PARP1 complex (Xie et al., 2015, PMID: 26344098) identified residues R1081 of TIMELESS and D993 of PARP1 as critical for this interaction. In Fig. 4e,f, we show that a TIMELESS R1081G mutant is defective in protecting replication forks from TRCs. In Fig. 4h, we show that a PARP1 D993G mutant is also defective in protecting replication forks from TRCs. These results suggest that the physical interaction between TIMELESS and PARP1 is needed for the functions of both TIMELESS and PARP1 to protect the replisome from TRCs and, therefore, indicates joint action."

The experiments showing that TIMELESS-PARP1 interaction mutations are not as effective as WT proteins in preventing DNA damage (gH2AX and 53BP1) lend support to the notion of a meaningful protein-protein interaction in the context of DNA replication. Can this be related to TRCs as proposed in the MS? In this case, the residual damage observed upon complementation with the interaction mutations should be suppressed by inhibiting transcription as cells enter S phase. Is this the case?

Textual issues:

(1) The authors might want to review the MS regarding referring to TRCs and TRC-dependent DNA damage. In the absence of assays showing such intermediates in the revised MS and given that TRCs are inferred only from rather indirect evidence, it would be more accurate to speak of transcription-dependent DNA damage.

(2) The notion that MiDAS is a bona fide readout of increased S-phase DNA damage from unresolved

TRCs in the cellular assays in the MS is perhaps inaccurate. Since RO-3306 pleiotropically interferes with DNA synthesis (Cell Rep., 39, 110701 (2022)), may the observed EdU incorporation simply reflect replication stress and late replication, not necessarily DNA double-strand break-associated DNA synthesis?

(3) The notion that yeast “Tof1 and Csm3 slow fork progression at highly transcribed loci (ref. 27)” serves as a blueprint for the TIMELESS-TIPIN functions suggested in the current MS reads like a bit of an overstatement. Ref. 27 describes accidental fork pausing mediated by Tof1 at sites where proteins are tightly bound to the DNA template. While this includes tRNA promoters, potentially in a transcription-independent manner, many other sites such as centromeres are also affected. (Similarly, refs. 25 and 26, which are referred to in the same context, do not actually deal with transcription-replication conflicts, but mechanisms to avoid transcription-replications collisions from occurring). The authors might consider rephrasing to better reflect the differences between yeast and the model proposed in the current MS.

Referee #3:

In this revised manuscript Petropoulos and colleagues make a strong case for the importance of TRCs as the source of DNA damage and synthetic lethality induced by PARP inhibition by PARPis in HR-deficient cells, with TIMELESS/TIPIN together with PARP1/2 acting to protect the replisome from TRCs. In contrast, PARP trapping seems to be less important for the efficiency of HRD cell killing by PARPis, but instead may play a significant role in clinical PARPi toxicity due to the impact it appears to have on HR-proficient cells.

This is a well-written paper, with an important message of interest to a broad audience.

The authors have addressed all concerns I raised in my original review.

I just have just on minor comment relating to the revised manuscript: The data presented in Fig. 3a-g is an important part of the argument that PARP inhibition is more important than PARP trapping. This would be further strengthened if panels e (PARP trapping) and g (DNA damage) showed data for the same cell line, i.e. panel e for HeLa and/or panel g for DLD BRCA2+/+.

Author Rebuttals to First Revision:

Revision of Manuscript: 2022-12-19944A
December 6, 2023

Response to Referees' Comments

We would like to thank the Referees for the careful review of our manuscript.

We have performed many more experiments to improve the manuscript based on the recommendations. Specifically, the following figure panels show entirely new experiments that either replace previous experiments or introduce new results:

- Fig 3d and Ext Data Fig 4d -- PARP1 and PARP2 trapping in cells: All new data; these figures replace Fig 3e (now all 4 PARP inhibitors have been tested for PARP1 and PARP2 trapping in DLD1 and HeLa cells - previously 3 inhibitors had been tested in DLD1 cells only).

- Fig 3f -- γ H2AX in early S phase: All new data; these figures replace Fig 3g (now all 4 PARP inhibitors tested for induction of γ H2AX in DLD1 and HeLa cells treated with and without DRB - previously the experiment was done only in DLD1 cells).

- Fig 3h -- induction of DNA damage by depletion of PARP1, PARP2 and PARP1&PARP2: All new data; this figure replaces Fig 3h (the experiment was redone to achieve better suppression of PARP1 and PARP2 protein levels).

- Fig 5j -- lethality of HR-deficient cells induced by PARP1 depletion can be partially suppressed by DRB: New data requested by Referee 2.

- Fig 6h -- survival of HR-proficient and HR-deficient cells treated with talazoparib and optionally with TMZ and transfected with siRNA targeting PARP1 and/or PARP2: All new data; this figure replaces the left panel of Fig 6h and Ext Data Fig 7b (the experiment was redone to achieve better PARP1 and PARP2 depletion).

- Ext Data Fig 2a,b -- induction of R-loops after depleting TIMELESS or TIPIN: New data requested by Referee 1.

- Ext Data Fig 2c-e -- overexpression of RNase H1 suppresses the DNA damage response induced by depleting TIMELESS or TIPIN: New data requested by Referee 1.

- Ext Data Fig 3i,j -- induction of R-loops after treatment of cells with PARP inhibitors: New data requested by Referee 1.

- Ext Data Fig 3k,l -- overexpression of RNase H1 suppresses the DNA damage response induced by PARP inhibitors: New data requested by Referee 1.

- Ext Data Fig 4c -- images from cells to show trapping of PARP1 and PARP2 on chromatin: Images requested by Referee 1.

- Ext Data Fig 4e -- images from cells to show induction of a DNA damage response: Images requested by Referee 1.

- Ext Data Fig 8 -- effect of depleting PARP1 and/or PARP2 and effect of PARP inhibitors on viability of multiple cell lines, with or without BRCA2 function: All new data; this figure replaces Ext Data Fig 6h, which examined much fewer cell lines and where the data had not been quantitated.

- Ext Data Fig 9f -- effect of depleting PARP1 on induction of γ H2AX in HR-proficient and HR-deficient cells treated with talazoparib: New data; this figure replaces Ext Data Fig 7d, which studied fewer data points.

- Ext Data Fig 9g -- effect of depleting PARP1 on induction of γ H2AX in HR-proficient and HR-deficient cells treated with talazoparib and TMZ: All new data; this figure replaces Ext Data Fig 7e, to achieve better PARP1 depletion.

Significant changes to the text and sentences that refer to the above figures are marked by blue-colored text. There are many stylistic changes throughout the text; these make the manuscript easier to read, but are not highlighted; otherwise, most of the text would have been colored blue.

Access to the high throughput DNA sequencing data can be found at:

<https://www.ncbi.nlm.nih.gov/geo/query/acc.cgi?acc=GSE220223>

Token: grkbuaculrkpvap

In what follows, we address the comments of the Referees, whom we thank for their support and critical input that has allowed us to improve the manuscript.

Referee #1:

"I would like to thank the authors for addressing the points raised by the reviewers. However, the authors substantially changed the manuscript without highlighting the changes in the manuscript text and figures. I cannot find data from old Fig. 5, on which my first comment was based."

We apologize for not highlighting the changes in the previous revision. We had made many changes and replaced many figure panels. In the current version, the text with major changes is highlighted in blue. For clarity, minor stylistic changes are not marked.

"1) Plots provided in new Fig. 6 and Ext. Data Fig. 7 do not address my point regarding PARP depletion vs PARP inhibition causing synthetic lethality with HRD. To address this, the authors should compare viability of BRCA+ vs BRCA- cell lines treated with 1) PARPi (at IC50 concentration), 2) siPARP1, 3) siPARP2, and 4) siPARP1 + siPARP2 (similar to old Figure 5 in HeLa and DLD1). I requested for this experiment to be done in additional BRCA+/- cell lines to further support their results. It is not sufficient to use BRCA-deficient OVSAHO and PEO1 ovarian cancer cells without their matched BRCA+ counterparts, and colony formation results should be quantified (pictures of the dishes provided in Ext. Data Fig. 6h are not sufficient). From the plots in Fig. 6, it seems that PARP1/2 depletion (without PARPi) does not cause lethality but actually increases survival in most cases."

We performed this experiment using cell lines in which *BRCA2* function was wild-type or suppressed (by siRNA or shRNA or by gene targeting). Five paired cell lines were examined (Ext Data Fig 8c). In addition, we examined the PEO1 and PEO4 pair, which are derived at different times from the same cancer, and are *BRCA2* mutant and wild-type, respectively, and the OVSAHO cell line, which has a copy number deletion of the *BRCA2* gene and a mutation in the *FANCD2* gene. We depleted PARP1 or PARP2 or both PARP1 and PARP2 in all these cells lines and in addition treated them with two PARP inhibitors (Ext Data Fig 8C). The data, which are now provided in a quantitative manner and are derived from experiments performed in triplicate, are very consistent and show that PARP1 depletion on its own or PARP1+PARP2 depletion, but not PARP2 depletion on its own, compromise the viability of the HR-deficient

cells, without affecting the HR-proficient cells. Similar results were observed with the PARP inhibitors.

"Instead, the authors examined the effect of combined PARP depletion and PARP inhibition. What is the rationale for this experiment? Data in Fig. 6h using siPARP1 addresses on- and off-targets of PARPi. If PARPi inhibits only PARP1, then PARP1 depletion should abrogate the lethality caused by PARPi. In BRCA-deficient cells talazoparib seems to exert its lethality by inhibiting PARP2, whereas in BRCA-proficient cells it also acts through PARP1 inhibition. AZD5305 is expected to be a selective PARP1 inhibitor. In BRCA-deficient cells siPARP1 cells are more sensitive to AZD5305, suggesting that it has off-target effects. Like talazoparib, in BRCA-deficient cells olaparib exerts its lethality by inhibiting PARP2. Based on data in Ext. Data Fig. 7, it seems that talazoparib has off-target effects, as combined siPARP1/2 has the same effect as siControl in BRCA-deficient cells treated with talazoparib. Unless the authors have an explanation for these results, I suggest removing them altogether from the manuscript (new Fig. 6 and related text) as they only distract from the main message and do not provide any solid conclusions."

In the current revision, the talazoparib data of Fig 6h of the prior revision have been repeated in their entirety and are shown in a new Fig 6h. The saruparib (aka AZD5305) and olaparib data are the same as in the previous version and are shown in Ext Data Fig 9d.

We think that neither PARP2, nor off-target effects explain our results. The most simple interpretation of our data is that in HR-deficient cells, lethality is mostly due to PARP1 inhibition, whereas in HR-proficient cells, lethality is mostly due to PARP1 trapping. We note that PARP trapping, in the absence of TMZ, requires higher doses of PARP inhibitors than the doses required to inhibit PARP enzymatic activity. Thus, low doses of PARP inhibitors kill HR-deficient cells due to inhibition of PARP1 enzymatic activity, but do not kill HR-proficient cells, since the latter are not killed when PARP1 enzymatic activity is inhibited. However, the HR-proficient cells are killed by high doses of PARP inhibitors due to PARP trapping. Depleting PARP1 by siRNA suppresses trapping and, therefore, makes HR-proficient cells even more resistant to PARP inhibitors. In HR-deficient cells, depleting PARP1 does not restore PARP1 enzymatic activity and, hence, these cells remain sensitive to PARP inhibitors.

We believe that the data in Fig 6h support the above interpretation.

Specifically, Fig 6h (lower left panel): EC50 for killing of BRCA2^{-/-} cells by talazoparib is 0.88 nM; when PARP1 is depleted it stays about the same (1.38 nM), depletion of PARP2 or PARP1+PARP2 also did not change the EC50s (0.66 and 0.82 nM, respectively). If trapping of PARP1 or PARP2 were important for killing, then the cells would have become resistant to talazoparib by the siRNA treatments.

Fig 6h (upper left panel): EC50 for killing of BRCA2^{+/+} cells by talazoparib is 386 nM; when PARP1 was depleted the cells became 50 times more resistant to talazoparib (EC50: 19,100 nM). PARP2 depletion had no effect (EC50: 542 nM) and depletion of PARP1+PARP2 was similar to depletion of PARP1 alone (EC50: 13,400 nM). The increased resistance to talazoparib when PARP1 was depleted, is consistent with the BRCA2^{+/+} cells being killed due to PARP1 trapping.

To further support the above model: administering 50 μ M TMZ with talazoparib, enhanced PARP1 trapping by talazoparib 220-fold (EC50 without TMZ: 812 nM, Fig 3d; with TMZ: 3.73 nM, Fig 6d); TMZ enhanced the killing of BRCA2^{+/+} cells 160-fold (EC50 without TMZ 386 nM; with TMZ 2.39 nM; Fig 6h, upper panels); in contrast, TMZ enhanced the killing of the BRCA2^{-/-} cells only 6-fold (EC50 without TMZ 0.88 nM; with TMZ 0.14 nM; Fig 6h, lower panels).

Therefore, combining PARP depletion with PARP inhibition helps us distinguish the mechanisms by which PARP inhibitors induce lethality of HR-deficient and HR-proficient cells.

"Furthermore, the authors included analysis of PARP trapping without explaining how it was done in the Methods and without providing raw data. Results in Fig. 3e are surprising as PARP trapping requires induction of DNA damage (with TMZ, MMS, IR...)."

In the revised version of the manuscript we provide a detailed description of the method and raw data (Ext Data Fig 4c). TMZ and other DNA damaging agents enhance PARP trapping, as shown by comparing the data of Figs 3d and 6d, but trapping can also occur in the absence of exogenous DNA damaging agents, as we demonstrate in our manuscript (Fig. 3d) and as reported also by Kim et al. (Fig. 3A; PMID: 32844745) and Demin et al. (Fig. 2A; PMID: 34102106). We note that trapping in the absence of exogenous DNA damaging agents requires longer treatment with PARP inhibitors (a few hours instead of minutes), which matches the exposure in cancer patients treated with PARP inhibitors.

"Based on extensive biochemical analysis, Rudolph and Luger (PMID: 33531508, 35259019, 37531469) conclude that PARPi differ in catalytic inhibition potencies, with talazoparib being the most potent inhibitor and that PARP trapping in cells correlates with the inhibition of PARP enzymatic activity. Your data point to differential PARPi potency in inhibiting catalytic activity, which is responsible for differential response of HR-proficient and HR-deficient cells to PARPi. I would revert to the old version of the manuscript with old Fig. 3 and related text."

The data of Rudolph and Luger, cited above, have guided our thinking and the design of our experiments. Inhibition of PARP catalytic activity is a major determinant of trapping, as their data demonstrate, yet it also appears that there is a role for reverse allostery, as described by Zandarashvili et al (Science, 2020; PMID: 32241924) and as seen by us. The original version of our manuscript did not include measurements of PARP trapping in cells, so we had to adjust our conclusions to take into account the new trapping data.

"2) An RNH1 rescue experiment is an essential control and must be provided in the same manuscript."

This is now included in Ext Data Fig 2d-f and Ext Data Fig 3k,l.

"3) The experiment with CPT should be included in this manuscript."

This is now included in Ext Data Fig 7g,h.

"4) Experiments with the RNH1 catalytic mutant should be included in this manuscript."

This is now included in Ext Data Fig 2a-c and Ext Data Fig 3i,j.

We thank Referee #1 for the very helpful comments and critical reading of our manuscript.

Referee #2:

"I would like to thank the authors for considering the points raised. The revised version is improved and better incorporates/discusses the limitations of the study, although I may not have caught all the revision changes in the absence of clear pointers/highlights in the revised MS pdf."

We apologize for not highlighting the changes in the previous revision. In this version, the text with major changes is highlighted in blue. For clarity, minor stylistic changes are not marked.

"Regarding specific points raised in round 1, and the responses given by the authors in the rebuttal letter, can the authors please address the following open questions:

(1) From authors' reply: "If requested, we are happy to do the experiment showing that inhibiting transcription elongation also suppresses the synthetic lethality induced by PARP1 and PARP2 depletion."

Please include this experiment, as this critical point remains unresolved, and please relate the outcomes to the model presented."

We did the requested experiment monitoring synthetic lethality by a colony forming assay in DLD1 BRCA2+/+ and BRCA2-/- cells after depletion of PARP1 by siRNA. The cells were blocked at G1/S by thymidine and released into S phase. For the first 200 min after release into S phase, the cells were treated with or without DRB; then the cells were replated and allowed to form colonies over 14 days. DRB suppressed the synthetic lethality induced by PARP1 depletion (Fig. 5i,j). We did not attempt longer treatments with DRB, since this is not well tolerated by cells.

"(2) From authors' reply: "We agree with the Referee that similar phenotypes of TIMELESS/TIPIN and PARP1 inhibition/depletion do not constitute evidence of joint action. However, we present experiments that indicate joint action. The crystal structure of the TIMELESS-PARP1 complex (Xie et al., 2015, PMID: 26344098) identified residues R1081 of TIMELESS and D993 of PARP1 as critical for this interaction. In Fig. 4e,f, we show that a TIMELESS R1081G mutant is defective in protecting replication forks from TRCs. In Fig. 4h, we show that a PARP1 D993G mutant is also defective in protecting replication forks from TRCs. These results suggest that the physical interaction between TIMELESS and PARP1 is needed for the functions of both TIMELESS and PARP1 to protect the replisome from TRCs and, therefore, indicates joint action."

The experiments showing that TIMELESS-PARP1 interaction mutations are not as effective as WT proteins in preventing DNA damage (gH2AX and 53BP1) lend support to the notion of a meaningful protein-protein interaction in the context of DNA replication. Can this be related to TRCs as proposed in the MS? In this case, the residual damage observed upon complementation with the interaction mutations should be suppressed by inhibiting transcription as cells enter S phase. Is this the case?"

We did not do the exact experiment that the Referee requested. However, we have other evidence to address whether the protein-protein interaction between TIMELESS and PARP1 is relevant in the context of TRCs. We show an experiment using a proximity ligation assay for PCNA and RNA Pol II, in which the TRCs induced by depleting TIMELESS are rescued by ectopic expression of wt TIMELESS, but not by expression of a single amino acid substitution mutant of TIMELESS that is defective in its interaction with PARP1 (Fig. 4f). In several other experiments the damage induced by TIMELESS or PARP1 depletion was ameliorated by DRB treatment. Moreover, depletion of TIMELESS or PARP1 have similar effects on fork progression over transcribed genes. We, therefore, hope that the Referee will be convinced that the TIMELESS-PARP1 protein-protein interaction is important for the response of cells to TRCs.

"Textual issues:

(1) The authors might want to review the MS regarding referring to TRCs and TRC-dependent DNA damage. In the absence of assays showing such intermediates in the revised MS and given that TRCs are inferred only from rather indirect evidence, it would be more accurate to speak of transcription-dependent DNA damage."

We understand the point of the Referee. However, the proposed term does not include DNA replication as a factor involved in the induction of DNA damage. We feel that the evidence implicating DNA replication is quite strong. First, we did not observe induction of DNA damage in EdU-negative cells (Ext Data Fig 1i-k) or even in late S-phase cells (Ext Data Fig 3c,d). Second, in this revision, we show that overexpression of RNase H1 rescues the induction of DNA damage (Ext Data Fig 2d-f and 3k,l). Third, depletion of TIMELESS or PARP1 affects fork progression rates over transcribed genes (Fig 1h and 4j).

"(2) The notion that MiDAS is a bona fide readout of increased S-phase DNA damage from unresolved TRCs in the cellular assays in the MS is perhaps inaccurate. Since RO-3306 pleiotropically interferes with DNA synthesis (Cell Rep., 39, 110701 (2022)), may the observed EdU incorporation simply reflect replication stress and late replication, not necessarily DNA double-strand break-associated DNA synthesis?"

We agree that MiDAS can be the result of many aberrant processes, including delayed DNA replication. Therefore, the MiDAS data shown in Ext Data Fig 3f-h do not by themselves constitute proof of TRCs, but, nevertheless, support our argument, since the induction of MiDAS is rescued by DRB in our experiments.

"(3) The notion that yeast "Tof1 and Csm3 slow fork progression at highly transcribed loci (ref. 27)" serves as a blueprint for the TIMELESS-TIPIN functions suggested in the current MS reads like a bit of an overstatement. Ref. 27 describes accidental fork pausing mediated by Tof1 at sites where proteins are tightly bound to the DNA template. While this includes tRNA promoters, potentially in a transcription-independent manner, many other sites such as centromeres are also affected. (Similarly, refs. 25 and 26, which are referred to in the same context, do not actually deal with transcription-replication conflicts, but mechanisms to avoid transcription-replications collisions from occurring). The authors might consider rephrasing to better reflect the differences between yeast and the model proposed in the current MS."

We thank the Referee for bringing up this point. We agree that yeast Tof1 is implicated in mechanisms to avoid transcription-replication collisions from occurring and have modified the text in the revised manuscript. We also thank the Referee for all the other comments cited above that helped us improve our manuscript.

Referee #3:

"In this revised manuscript Petropoulos and colleagues make a strong case for the importance of TRCs as the source of DNA damage and synthetic lethality induced by PARP inhibition by PARPis in HR-deficient cells, with TIMELESS/TIPIN together with PARP1/2 acting to protect the replisome from TRCs. In contrast, PARP trapping seems to be less important for the efficiency of HRD cell killing by PARPis, but instead may play a significant role in clinical PARPi toxicity due to the impact it appears to have on HR-proficient cells.

This is a well-written paper, with an important message of interest to a broad audience.

The authors have addressed all concerns I raised in my original review.

I just have just on minor comment relating to the revised manuscript: The data presented in Fig. 3a-g is an important part of the argument that PARP inhibition is more important than PARP trapping. This would be further strengthened if panels e (PARP trapping) and g (DNA damage) showed data for the same cell line, i.e. panel e for HeLa and/or panel g for DLD BRCA2+/+."

Indeed, we have now performed the requested experiments to have PARP trapping and DNA damage response data for both HeLa and DLD1 BRCA2^{+/+} cells (new Fig 3d,f and Ext Data Fig 4d).

We thank the Referee for the positive remarks, as well as for all the comments on how to improve our manuscript (the original and revised version).

Reviewer Reports on the Second Revision:

Referees' comments:

Referee #1:

I would like to thank the authors for improving the manuscript further and indicating major changes. However, my point from before remains and I would urge the authors to rethink their results in Fig. 6h (and Ext. Data Fig. 8):

How can cells in which PARP1 is depleted be sensitive to PARP1 inhibitor if PARP1 is not there (BRCA2-/- lower panels)? As explained before, this can only happen through off-target effects. I cannot support publication of this manuscript until this point is clarified.

In addition, I have a few minor comments:

1) Fig. 1c, Ext. Data Fig. 1d,k: Please include statistics for comparisons between DRB-treated samples as it seems that DRB-treated siTIP and siTIM have higher levels of gH2AX, 53BP1 and RAD51 compared to DRB-treated siControl. The same applies to other figures. Please comment on these findings, which suggest that the effects of siTIP/siTIM or PARPi are not only mediated through TRCs.

2) Why was the experiment in Ext. Data Fig. 1h analysed differently compared to Fig. 1c? What do individual points in Ext. Data Fig. 1h indicate? Please clarify. Please refrain from different analysis/plotting of similar datasets. It confuses the reader. The same applies to Ext. Data Fig. 2f, 2d,e, Ext. Data Fig. 3b,j,l, 4f.

3) Fig. 6d: It should say Trapped PARP2 in the lower panel.

Referee #2:

The authors have substantially revised and improved the MS. The points raised previously have been adequately addressed with the exception of one remaining experimental issue that I hope can be easily addressed prior to publication.

The revision has not addressed whether residual DNA damage observed upon complementation of TIMELESS-depleted HeLa cells with a TIMELESS-PARP1 interaction mutant is suppressed by inhibiting transcription as cells enter S phase. I remain of the opinion that it should be demonstrated that in the set-up of Fig. 4e the DNA damage that is implied to arise by transcription should be demonstrated to subside upon transcription inhibition. By using the mainstay readout used throughout the MS, and if this experiment has the outcome the authors might expect, this will considerably strengthen the data for a meaningful protein-protein interaction between TIMELESS and PARP1 in the context of TRCs, and,

importantly, rule out any confounding effects that might be associated with the (over)expression of the TIMELESS mutant. The experiment seems very straightforward and should not be omitted.

Textual: Regarding the caveat of using the MiDAS assay as supporting evidence for TRC-linked DNA damage, I think there is a misunderstanding in the rebuttal letter that I'd like to clarify. The point I was trying to make is not that "many aberrant processes" result in MiDAS. Rather, the protocol employed and the use of RO-3306 in the MS very likely results in the authors observing replicative EdU incorporation late in the cell cycle upon removal of RO, but not mitotic repair synthesis. It would be more in line with the current literature to acknowledge this possibility, which, I believe, does not detract from the argument the authors make.

Referee 1's response to your response:

Thank you for explaining how the plots in Fig. 6 were generated. However, I disagree with the way the survival curves were plotted. Each curve should be normalized to its own DMSO control (not to the DMSO control of siCTRL). With the way you plotted the data, the information about the effect of BRCA2 KO is completely lost, and it's clear that BRCA2 loss has a huge effect on cell survival of PARP1-depleted cells. Moreover, it becomes clear that talazoparib has a very similar effect in WT and BRCA2 KO cells when PARP1 is depleted (not that, as you concluded, talazoparib is more effective in BRCA2 KO HR-deficient cells). This way the results make sense because the effect of talazoparib in PARP1-depleted cells is small in both WT and BRCA2 KO cells, which would be expected if PARP1 as the target for talazoparib inhibition is depleted.

Author Rebuttals to Second Revision:

Revision of Manuscript: 2022-12-19944B

January 21, 2024

Response to Referees' Comments

We would like to thank the Referees for the careful review of our manuscript and, more so, for providing feedback in a very timely manner. This has allowed us to perform the necessary experiments and other revisions needed and we hope that the manuscript is now acceptable for publication.

In terms of new experiments, in this revision, the experiment that is reported in Fig 4d, 4e is new and we think that the results address the major point raised by Referee #2.

We have also revised Fig 6h, as requested by Referee #1, and have made several other changes to address the minor points raised by Referees #1 and #2. We understand that Referee #3 had no comments in regard to this version of the manuscript.

The changes in the manuscript are highlighted by having the relevant text in blue colour. We thank again all three Referees for their advice and comments, which have led to a much improved manuscript, as compared to the original version.

Referee #1:

"I would like to thank the authors for improving the manuscript further and indicating major changes. However, my point from before remains and I would urge the authors to rethink their results in Fig. 6h (and Ext. Data Fig. 8):

How can cells in which PARP1 is depleted be sensitive to PARP1 inhibitor if PARP1 is not there (BRCA2^{-/-} lower panels)? As explained before, this can only happen through off-target effects. I cannot support publication of this manuscript until this point is clarified."

We show below the talazoparib dose-response curves from Fig 6h of the previous version that the Referee is referring to. For simplicity we only show the plots for the TMZ-untreated CTRL and PARP1 siRNA-transfected cells: BRCA2 KO (left) and BRCA2 WT (right).

The data were adjusted so that the survival of the cells for each siRNA condition in the absence of talazoparib would be 100%.

The graphs below show the non-adjusted CTG survival values (RLU, relative light units) as a function of talazoparib concentration for the CTRL and PARP1 siRNA-transfected cells: BRCA2 KO (left) and BRCA2 WT (right).

The non-adjusted data show that depleting PARP1 is lethal for the BRCA2 KO cells, but not for the BRCA2 WT cells. Of the BRCA2 KO cells, about 20% survive the depletion of PARP1 by siRNA. Considering that PARP1 is the most abundant nuclear protein after histones, it is likely that PARP1 is not completely depleted by siRNA in the surviving BRCA2 KO cells. These surviving cells with incomplete PARP1 depletion will be sensitive to talazoparib with the same EC50 as the siCTRL-transfected BRCA2 KO cells (which is what we saw), since PARP1 is the target in both cases.

The BRCA2 WT cells transfected with PARP1 siRNA are more resistant to talazoparib than the BRCA2 WT cells transfected with CTRL siRNA, which is consistent with talazoparib inducing lethality of BRCA2 WT cells by trapping PARP1 on chromatin. Thus, the responses of the BRCA2 KO and WT cells to talazoparib following PARP1 depletion reveals that different mechanisms are involved in the killing of these cells by talazoparib. If PARP1 trapping was required for the lethality of the BRCA2 KO cells by talazoparib, then depleting endogenous PARP1 by siRNA would have made the BRCA2 KO cells resistant to talazoparib, which is not the case.

We agree that showing the adjusted data, as we did in the previous version of the manuscript, is confusing. For this reason and in response to the comment of the Referee, we now show the non-adjusted data, expressed as 100% survival, in the revised Fig 6h.

An alternative interpretation of our data could be that talazoparib has an off-target effect that is responsible for the killing of PARP1 siRNA-transfected BRCA2 KO cells by talazoparib. We understand why the Referee proposed this alternative, since the non-adjusted data were not available. However, the off-target effect hypothesis cannot explain our data, since it implies the existence of a non-PARP target of talazoparib that confers synthetic lethality to BRCA2 KO cells, but not to BRCA2 WT cells (since talazoparib was very inefficient in killing BRCA2 WT cells). Which target would this be, whose inhibition leads to synthetic lethality only in the context of BRCA2 KO cells?

We had the opportunity to provide our response to Referee 1 (along the lines cited above) a couple of weeks ago and the Referee was kind enough to make the following comment:

“Thank you for explaining how the plots in Fig. 6 were generated. However, I disagree with the way the survival curves were plotted. Each curve should be normalized to its own DMSO control (not to the DMSO control of siCTRL). With the way you plotted the data, the information about the effect of BRCA2 KO is completely lost, and it’s clear that BRCA2 loss has a huge effect on cell survival of PARP1-depleted cells. Moreover, it becomes clear that talazoparib has a very similar effect in WT and BRCA2 KO cells when PARP1 is depleted (not that, as you concluded, talazoparib is more effective in BRCA2 KO HR-deficient cells). This way the results make sense because the effect of talazoparib in PARP1-depleted cells is small in both WT and BRCA2 KO cells, which would be expected if PARP1 as the target for talazoparib inhibition is depleted.”

We thank the Referee for stating that the results make sense with the new way of presenting the data. We also agree that the way we phrased our conclusion was not correct and, therefore, we have reworded our conclusion, as suggested by the Referee.

Additional minor comments of Referee #1:

“1) Fig. 1c, Ext. Data Fig. 1d,k: Please include statistics for comparisons between DRB-treated samples as it seems that DRB-treated siTIP and siTIM have higher levels of gH2AX, 53BP1 and RAD51 compared to DRB-treated siControl. The same applies to other figures. Please comment on these findings, which suggest that the effects of siTIP/siTIM or PARPi are not only mediated through TRCs.”

The graphs below show Fig 1c and Extended Data Fig 1d, 1k with the statistics comparisons between DRB-treated samples, as the Referee requested. Some differences between the DRB-treated siTIP and siTIM samples and the DRB-treated siCTRL samples are statistically significant (red lines and asterisks). These differences could indicate that part of the effects of siTIP or siTIM are mediated via mechanisms other than TRCs. In fact, some of these differences are also seen in non-replicating EdU-negative cells, when unsynchronized cells were examined (see the lower panels of Extended Data Fig 1k, shown below). An alternative explanation is that transcription elongation complexes, that were already ongoing at the time DRB was added to the cells, are responsible for TRCs and a DNA damage response. In the revised manuscript we mention that some of the differences between the DRB-treated siTIP and siTIM and the DRB-treated siCTRL samples are statistically significant and provide the explanations cited above (see legend of Fig 1).

Fig. 1c

Extended Data Fig. 1k

Extended Data Fig. 1d

-Statistics for -DRB vs +DRB treated samples are shown in black
-Statistics for +DRB vs +DRB treated samples are shown in red

“2) Why was the experiment in Ext. Data Fig. 1h analysed differently compared to Fig. 1c? What do individual points in Ext. Data Fig. 1h indicate? Please clarify. Please refrain from different analysis/plotting of similar datasets. It confuses the reader. The same applies to Ext. Data Fig. 2f, 2d,e, Ext. Data Fig. 3b,j,l, 4f.”

Fig 1c plots the percentages of cells with DNA damage. Ext Data Fig 1h plots the number of 53BP1 foci per cell. Whenever, we plot percentage of cells in a population we use the format of Fig 1c. Whenever, we plot data per cell, eg. number of foci per cell or mean intensity of immunofluorescence signal per cell, we plot the data as shown in Ext Data Fig 1h. The individual points in Ext Data Fig 1 h correspond to the cells that are in the top 10 or bottom 10 percentiles of the cell population. We now explain this in the figure legends.

“3) Fig. 6d: It should say Trapped PARP2 in the lower panel.”

Thank you. You are right. We corrected this.

We take the opportunity to thank very much Referee #1 for her/his input throughout the revision process.

Referee #2:

“The authors have substantially revised and improved the MS. The points raised previously have been adequately addressed with the exception of one remaining experimental issue that I hope can be easily addressed prior to publication.

The revision has not addressed whether residual DNA damage observed upon complementation of TIMELESS-depleted HeLa cells with a TIMELESS-PARP1 interaction mutant is suppressed by inhibiting transcription as cells enter S phase. I remain of the opinion that it should be demonstrated that in the set-up of Fig. 4e the DNA damage that is implied to arise by transcription should be demonstrated to subside upon transcription inhibition. By using the mainstay readout used throughout the MS, and if this experiment has the outcome the authors might expect, this will considerably strengthen the data for a meaningful protein-protein interaction between TIMELESS and PARP1 in the context of TRCs, and, importantly, rule out any confounding effects that might be associated with the (over)expression of the TIMELESS mutant. The experiment seems very straightforward and should not be omitted.”

We performed the experiment as the Referee suggested. In the new experiment we reduced the amount of plasmid transfected so that the levels of ectopically expressed TIMELESS are physiological (Fig 4d). The DNA damage-induction data are again very clean, as the wt TIMELESS protein suppresses the induction of DNA damage, while the mutant TIMELESS does not (Fig 4e). Moreover, in support of our model, DRB treatment reduces the levels of DNA damage observed in cells expressing the mutant TIMELESS protein (Fig 4e).

“Textual: Regarding the caveat of using the MiDAS assay as supporting evidence for TRC-linked DNA damage, I think there is a misunderstanding in the rebuttal letter that I'd like to clarify. The point I was trying to make is not that "many aberrant processes" result in MiDAS. Rather, the protocol employed and the use of RO-3306 in the MS very likely results in the authors observing replicative EdU incorporation late in the cell cycle upon removal of RO, but not mitotic repair synthesis. It would be more in line with the current literature to acknowledge this possibility, which, I believe, does not detract from the argument the authors make.”

To score MiDAS, we counted EdU foci in cells that were in mitosis (the cells were observed by microscopy to identify the mitotic cells by the presence of condensed chromosomes; the number of EdU foci decorating these chromosomes was then counted). While it is possible that the cells might have incorporated EdU in very late S phase, just before they entered mitosis, our prior experience with MiDAS (Macheret et al., Cell Research, 2000), suggests that cells released from a thymidine block in the presence of RO-3306, but in the absence of aphidicolin, will have completed S phase at the 10.5 h timepoint, when RO-3306 was washed away. Thus, under the conditions of the experiments reported here, we are most likely monitoring MiDAS. We would be grateful to review literature, that we may have missed, focusing more on this topic.

We very much want to thank Referee #2 for the very rapid review of our manuscript which allowed us to revise Fig 4e in a timely manner, as well as for all the comments during the review of this and the previous versions.

We believe that our manuscript has improved dramatically since the original version, thanks to the input we received from all the Referees, and we are grateful to them.

Reviewer Reports on the Third Revision:

Referee's comments:

Referee #1:

The changes have not been highlighted in the text and the text hasn't been modified according to correctly plotted CFUs!

The following changes are required for me to accept this manuscript:

1) abstract:

Remove:

In contrast to HR-deficient cells, the sensitivity of HR-proficient cells to PARP inhibitors was due to PARP trapping and could be ameliorated by depleting PARP1 by siRNA. We propose that inhibition of PARP1 enzymatic activity suffices for treatment efficacy in HR-deficient settings, whereas PARP trapping enhances toxicity.

Replace with:

Our results suggest that inhibition of PARP1 enzymatic activity suffices for treatment efficacy in HR-deficient settings.

2) results:

Remove:

However, it still remains to be determined whether PARP trapping
293 contributes to the efficacy of PARP inhibitors and whether it affects the selectivity of these inhibitors for HR-deficient over HR-proficient cells.

Replace with:

To address this further, we determined dose-response curves by which the four PARP inhibitors induced lethality of DLD1 BRCA2^{-/-} and BRCA2^{+/+} cells (Fig.

6a,b). For the HR-deficient BRCA2^{-/-} cells, the dose-response lethality curves (Fig. 6b) matched the curves for inhibition of PARP1 cellular enzymatic activity (Fig. 3b), but not the curves for PARP1 or PARP2 trapping (Fig. 3d and Extended Data Fig. 4d). The inverse was observed when we plotted the dose-response curves for the HR-proficient BRCA2^{+/+} cells; here, the dose-response curves for induction of lethality (Fig. 6b) matched the curves for PARP1 and PARP2 trapping (Fig. 3d and Extended Data Fig. 4d), rather than the curves for inhibition of PARP1 cellular enzymatic activity (Fig. 3b).

Remove:

If enhanced PARP trapping decreases the selectivity of PARP inhibitors for HR-deficient over HR-proficient cells, then less PARP trapping should increase selectivity. To explore this hypothesis, we determined dose-response lethality curves for talazoparib-treated DLD1 BRCA2^{-/-} and DLD1 BRCA2^{+/+} cells, in which endogenous PARP1 was depleted by siRNA (Fig. 6g). Depletion of PARP1

rendered the HR-proficient cells significantly more resistant to talazoparib (Fig. 6h). Similar results were observed when talazoparib was co-administered with TMZ to enhance PARP trapping (Fig. 6h). Finally, consistent with lethality being linked to a persistent DNA damage response, depletion of PARP1 suppressed the DNA damage response in talazoparib-treated HR-proficient, but not HR-deficient, cells (Extended Data Fig. 9c-e).

Replace with:

Depletion of PARP1 by siRNA severely reduced the viability of HR-deficient BRCA2^{-/-} cells (Fig. 6g,h), suggesting that the loss of PARP1 enzymatic activity is sufficient to induce lethality in HR-deficient cells. PARP1-depleted BRCA2^{+/+} cells were highly resistant to talazoparib, in accordance with its on-target inhibition (Fig. 6h). Finally, consistent with lethality being linked to a persistent DNA damage response, depletion of PARP1 suppressed the DNA damage response in talazoparib-treated HR-proficient, but not HR-deficient cells (Extended Data Fig. 9c-e).

3) discussion:

Remove:

Combining PARP inhibitors with siRNA-mediated depletion of PARP1 led to cell lethality of HR-deficient cells, even though these conditions should diminish PARP trapping. Finally, depletion of PARP1 with or without depletion of PARP2, in the absence of PARP inhibitors, was sufficient to lead to lethality of HR-deficient cells.

Replace with:

Finally, depletion of PARP1 with or without depletion of PARP2 was sufficient to lead to lethality of HR-deficient cells.

Remove:

Our data suggest that the lethality of HR-proficient cells was mediated principally by PARP trapping. Our panel of four PARP inhibitors showed a very good correlation between lethality and trapping potential. Administering TMZ, which leads to increased PARP trapping, augmented significantly the lethality of PARP inhibitors for HR-proficient cells.

Author Rebuttals to Third Revision:
Revision of Manuscript: 2022-12-19944C
February 9, 2024

Response to Referees' Comments

Referee #1:

"The changes have not been highlighted in the text and the text hasn't been modified according to correctly plotted CFUs!

The following changes are required for me to accept this manuscript:

1) abstract:

Remove:

In contrast to HR-deficient cells, the sensitivity of HR-proficient cells to PARP inhibitors was due to PARP trapping and could be ameliorated by depleting PARP1 by siRNA. We propose that inhibition of PARP1 enzymatic activity suffices for treatment efficacy in HR-deficient settings, whereas PARP trapping enhances toxicity.

Replace with:

Our results suggest that inhibition of PARP1 enzymatic activity suffices for treatment efficacy in HR-deficient settings.

2) results:

Remove:

However, it still remains to be determined whether PARP trapping contributes to the efficacy of PARP inhibitors and whether it affects the selectivity of these inhibitors for HR-deficient over HR-proficient cells.

Replace with:

To address this further, we determined dose-response curves by which the four PARP inhibitors induced lethality of DLD1 BRCA2^{-/-} and BRCA2^{+/+} cells (Fig. 6a,b). For the HR-deficient BRCA2^{-/-} cells, the dose-response lethality curves (Fig. 6b) matched the curves for inhibition of PARP1 cellular enzymatic activity (Fig. 3b), but not the curves for PARP1 or PARP2 trapping (Fig. 3d and Extended Data Fig. 4d). The inverse was observed when we plotted the dose-response curves for the HR-proficient BRCA2^{+/+} cells; here, the dose-response curves for induction of lethality (Fig. 6b) matched the curves for PARP1 and PARP2 trapping (Fig. 3d and Extended Data Fig. 4d), rather than the curves for inhibition of PARP1 cellular enzymatic activity (Fig. 3b).

Remove:

If enhanced PARP trapping decreases the selectivity of PARP inhibitors for HR-deficient over HR-proficient cells, then less PARP trapping should increase selectivity. To explore this hypothesis, we determined dose-response lethality curves for talazoparib-treated DLD1 BRCA2^{-/-} and DLD1 BRCA2^{+/+} cells, in which endogenous PARP1 was depleted by siRNA (Fig. 6g). Depletion of PARP1 rendered the HR-proficient cells significantly more resistant to talazoparib (Fig. 6h). Similar results were observed when talazoparib was co-

administered with TMZ to enhance PARP trapping (Fig. 6h). Finally, consistent with lethality being linked to a persistent DNA damage response, depletion of PARP1 suppressed the DNA damage response in talazoparib-treated HR-proficient, but not HR-deficient, cells (Extended Data Fig. 9c-e).

Replace with:

Depletion of PARP1 by siRNA severely reduced the viability of HR-deficient BRCA2^{-/-} cells (Fig. 6g,h), suggesting that the loss of PARP1 enzymatic activity is sufficient to induce lethality in HR-deficient cells. PARP1-depleted BRCA2^{+/+} cells were highly resistant to talazoparib, in accordance with its on-target inhibition (Fig. 6h). Finally, consistent with lethality being linked to a persistent DNA damage response, depletion of PARP1 suppressed the DNA damage response in talazoparib-treated HR-proficient, but not HR-deficient cells (Extended Data Fig. 9c-e).

3) discussion:

Remove:

Combining PARP inhibitors with siRNA-mediated depletion of PARP1 led to cell lethality of HR-deficient cells, even though these conditions should diminish PARP trapping. Finally, depletion of PARP1 with or without depletion of PARP2, in the absence of PARP inhibitors, was sufficient to lead to lethality of HR-deficient cells.

Replace with:

Finally, depletion of PARP1 with or without depletion of PARP2 was sufficient to lead to lethality of HR-deficient cells.

Remove:

Our data suggest that the lethality of HR-proficient cells was mediated principally by PARP trapping. Our panel of four PARP inhibitors showed a very good correlation between lethality and trapping potential. Administering TMZ, which leads to increased PARP trapping, augmented significantly the lethality of PARP inhibitors for HR-proficient cells.”

We have made the changes requested by the Referee and thank her/him for the many helpful comments during the review of our manuscript.

We understand that Referees #2 and #3 did not have any comments at this round of revision.